# PETS: A Principled Framework Towards Optimal Trajectory Allocation for Efficient Test-Time Self-Consistency

**Zhangyi Liu** [* 1]  **Huaizhi Qu** [* 2]  **Xiaowei Yin** [* 2]  **He Sun** [3]  **Yanjun Han** [4]  **Tianlong Chen** [2]  **Zhun Deng** [2]

## Abstract

Test-time scaling can improve model performance by aggregating stochastic reasoning trajectories. However, achieving sample-efficient test-time self-consistency under a limited budget remains an open challenge. We introduce PETS (**P**rincipled and **E**fficient **T**est-Time **S**elf-Consistency), which initiates a principled study of trajectory allocation through an optimization framework. Central to our approach is the *self-consistency rate*, a new measure defined as agreement with the infinite-budget majority vote. This formulation makes sample-efficient test-time allocation theoretically grounded and amenable to rigorous analysis. We study both offline and online settings. In the offline regime, where all questions are known in advance, we connect trajectory allocation to crowdsourcing, a classic and well-developed area, by modeling reasoning traces as workers. This perspective allows us to leverage rich existing theory, yielding theoretical guarantees and an efficient majority-voting-based allocation algorithm. In the online streaming regime, where questions arrive sequentially and allocations must be made on the fly, we propose a novel method inspired by the offline framework. Our approach adapts budgets to question difficulty while preserving strong theoretical guarantees and computational efficiency. Experiments show that PETS consistently outperforms uniform allocation. On GPQA, PETS achieves perfect self-consistency in both settings while reducing the sampling budget by up to $75\%$ (offline) and $55\%$ (online) relative to uniform allocation.

---

[*]Equal contribution [1]Stanford University [2]UNC at Chapel Hill [3]Yale University [4]New York University. Correspondence to: Zhun Deng <zhundeng@cs.unc.edu>.

*Proceedings of the 43rd International Conference on Machine Learning*, Seoul, South Korea. PMLR 306, 2026. Copyright 2026 by the author(s).

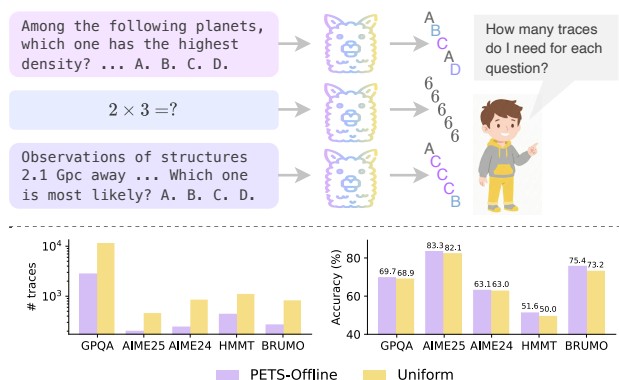

*Figure 1.* In this paper, we study how to allocate an LLM's sampling budget across questions to best match the full-budget outcome under self-consistency. Our results show that PETS substantially reduces the required budget while maintaining accuracy.

## 1. Introduction

Test-time scaling methods can substantially enhance LLMs' reasoning performance (Muennighoff et al., 2025; Huang et al., 2025b; Guo et al., 2025; Snell et al., 2024; Fu et al., 2025a). Self-consistency (Wang et al., 2022), adopted in Gemini's Deep Think mode (Thang Luong & Edward Lockhart), is a simple and effective approach but can be computationally expensive due to the need to sample many reasoning traces per query. A natural remedy is to allocate different sampling budgets across questions (Figure 1). Most existing methods rely on heuristic signals, such as trace-level confidence (Fu et al., 2025a) or LLM-predicted difficulty (Wang et al., 2025a), which typically lack theoretical guarantees and may use the budget inefficiently. Another line of work analyzes test-time scaling with external reward models or verifiers (Snell et al., 2024; Huang et al., 2025a; Zuo & Zhu, 2025). However, reward models can be mis-specified in practice, especially on complex or out-of-distribution queries, leading to unreliable guidance. They also introduce additional training and deployment costs.

These limitations motivate the need for theoretically principled test-time scaling methods that do not rely on auxiliary supervision, but instead leverage the intrinsic structure of self-consistency itself. Under this lens, we propose PETS, a **P**rincipled and **E**fficient **T**est-Time **S**elf-Consistency frame-

work for trajectory allocation. Our approach is inspired by the principle that sampling effort should be distributed according to the *relative difficulty* of different questions, rather than allocated uniformly. Central to PETS is the optimization of **self-consistency rate**, defined as the probability that majority voting with a finite budget $B$ matches the population majority. Because questions at different difficulty levels $\boldsymbol{\theta}$ exhibit distinct convergence behaviors, they require different numbers of traces to attain high self-consistency. Our goal is therefore to allocate traces across questions so as to maximize the aggregate self-consistency rate of the question set under a fixed budget.

We study two settings: ❶ *Offline batch setting*, where the entire question set is available for optimization. In this regime, we connect trajectory allocation to crowdsourcing by modeling reasoning traces as workers. We adopt and extend a Bayesian adaptive trajectory allocation algorithm (Section 3) that iteratively allocates additional trajectories while maintaining a posterior over each question's difficulty level. ❷ *Online streaming setting*, where questions arrive sequentially and allocation must be made without access to the full dataset. Inspired by the offline case. We propose a one-shot allocation strategy (Section 4), which can be cast into a supervised learning case as in the batch setting with the help of additional samples as the training set to estimate the distribution of question difficulties. We solve a constrained allocation problem to obtain a budget allocation plan over difficulty levels, and use a *provably optimal* procedure to instantiate per-question budgets upon arrival.

Experiments show that PETS significantly improves the efficiency of self-consistency while maintaining accuracy, demonstrating its strong practicability. As shown in Figure 1, compared to naive uniform allocation, it requires substantially fewer trajectories to reach full self-consistency, achieves higher self-consistency under the same budget, and ultimately translates these gains into improved accuracy.

**Our contribution.** To summarize, we develop theoretically principled test-time scaling methods that leverage the intrinsic structure of self-consistency. Specifically,

- We introduce *self-consistency rate*, a new performance measure defined as agreement with the infinite-budget majority vote. This formulation provides a principled target for finite-budget inference and enables rigorous theoretical analysis of sample-efficient allocation.
- In the offline setting, we make the first connection between self-consistency trajectory allocation and optimal budget allocation in crowdsourcing. This allows us to employ a rich body of existing tools to maximize the expected gain in self-consistency rate per allocation.
- In the online setting, we propose a novel one-shot allocation strategy that estimates the difficulty distribution from a small training set and solves a constrained op-

timization problem to determine per-question budgets for allocation upon arrival.
- Extensive experiments show that PETS consistently reduces the traces needed to reach full self-consistency in both unweighted and weighted cases, improves self-consistency under fixed budgets, and yields higher accuracy than uniform baselines.

## 2. Setup

We consider a collection of multiple-choice (or fill-in-the-blank) questions $\{q_i\}_{i=1}^N$. For each question $q_i$, we assume access to a stochastic reasoning procedure. Each sampled reasoning trace produces a random answer $Y_i$ sampled from answer set $\mathcal{Y}$. Without loss of generality, we use $\mathcal{Y} = \{1, \ldots, M\}$ to denote the collection of possible answers. Reasoning traces are assumed to be i.i.d. conditioned on $q_i$.

**Majority voting and related concepts.** Given a finite integer budget $B \in \mathbb{N}_+$, for each question $q_i$, $B$ reasoning traces $T_i^{(1)}, T_i^{(2)}, \cdots, T_i^{(B)}$ are i.i.d. drawn and lead to $B$ answers $Y_i^{(1)}, Y_i^{(2)}, \cdots, Y_i^{(B)}$. For each answer in $y \in \mathcal{Y}$, the corresponding vote count is

$$V_i(y; B) = \sum_{j=1}^{B} \mathbf{1}\{Y_i^{(j)} = y\}.$$

The final answer based on majority voting is denoted as

$$Y_i^{\mathrm{Maj}}(B) = \arg\max_{y \in \mathcal{Y}} V_i(y; B).$$

Similarly, for each question $q_i$, we define weighted majority voting by considering the weighted vote count for weight vector $\boldsymbol{w}_i = (w_i^{(1)}, w_i^{(2)}, \cdots, w_i^{(B)})^\top \in \mathbb{R}_+^B$ of all $j \in [B]$.

$$V_i^{\mathrm{W}}(y; \boldsymbol{w}_i, B) = \sum_{j=1}^{B} w_i^{(j)} \mathbf{1}\{Y_i^{(j)} = y\}.$$

Correspondingly, we can define the final answer based on weighted majority voting as:

$$Y_i^{\mathrm{WMaj}}(B) = \arg\max_{y \in \mathcal{Y}} V_i^{\mathrm{W}}(y; \boldsymbol{w}_i, B).$$

In this paper, we mainly follow Fu et al. (2025a) so that for each question $q_i$, the corresponding weight $w_i^{(j)}$ depends only on $T_i^{(j)}$, so that the final weight vector takes the form $\boldsymbol{w}_i = (w(T_i^{(1)}), w(T_i^{(2)}), \cdots, w(T_i^{(B)}))^\top$.

We make a natural assumption[1] that the trace and answer pairs $(T_i^{(1)}, Y_i^{(1)}), (T_i^{(2)}, Y_i^{(2)}), \ldots, (T_i^{(B)}, Y_i^{(B)})$ are i.i.d. drawn from an unknown joint distribution $\mathcal{D}_i$. The marginal

---

[1]Traces are independently generated from the same question $q_i$ without fixing a random seed.

distribution of $Y_i^{(j)} \in \{1, \ldots, M\}$ is given by a pmf $\boldsymbol{\theta}_i = (\theta_{i,1}, \theta_{i,2}, \cdots, \theta_{i,M})^\top$, where $\theta_{i,y} = \mathbb{P}(Y_i = y)$ are unknown parameters. The pmf $\boldsymbol{\theta}$ characterizes the uncertainty in LLM's answer to the question $q_i$, and can thus represent the difficulty of each problem.[2]

We further define $y_i^\infty = \arg\max_{y \in \mathcal{Y}} \theta_{i,y}$ and $y_i^{W,\infty} = \arg\max_{y \in \mathcal{Y}} \mathbb{E}_{(T_i, Y_i) \sim \mathcal{D}_i}[w(T_i)\mathbf{1}\{Y_i = y\}]$, which correspond to the population (weighted) majority label in the infinite sample limit for question $q_i$. $y_i^\infty$ and $y_i^{W,\infty}$ formalize the fundamental limit of test-time scaling. With an infinite sampling budget, aggregating infinitely many reasoning trajectories yields a well-defined consensus prediction given by the infinite-budget (weighted) majority vote.

Given the above content, we can define the *self-consistency rate* of the answer obtained by majority voting and the weighted version for question $q_i$ under budget $B$ as

$$\text{SC}(q_i; B) = \mathbb{P}(Y_i^{\text{Maj}}(B) = y_i^\infty | \boldsymbol{\theta}_i) \tag{1}$$

and

$$\text{SC}^W(q_i; B) = \mathbb{P}(Y_i^{\text{WMaj}}(B) = y_i^{W,\infty} | \mathcal{D}_i). \tag{2}$$

Intuitively, self-consistency measures how likely a budget-$B$ (weighted) vote recovers the population (weighted) majority label, providing a principled target for finite-budget inference. By defining self-consistency rate, we can ***rigorously study how test-time samples approximate infinite-budget behavior and design sample-efficient allocation strategies under realistic computational constraints***. Moreover, the rate at which it improves with $B$ provides a natural notion of question difficulty. Both quantities are monotonically increasing in $B$, and their growth is governed by $\boldsymbol{\theta}_i$ (or $\mathcal{D}_i$ in the weighted case).

**Optimal trajectory budget allocation problem.** Our general goal is to design budget allocation strategies that maximize self-consistency rate under a limited reasoning total budget in various settings. Loosely speaking, given a set of questions $\{q_i\}_{i=1}^N$, our aim is to find a policy $\pi$ to allocate budget for each question, i.e., number of reasoning traces for each question.

$$\max_\pi \sum_{i=1}^N \mathbb{E}_{\mathcal{D}_i \sim \mathcal{D}_i^{\text{meta}}(\pi)} \big[ \text{SC}^W(q_i; B_\pi(q_i)) \big],$$
$$\text{s.t.} \sum_{i=1}^N B_\pi(q_i) \leq B_{\text{total}}. \tag{3}$$

Here, the joint distribution of $(T_i, Y_i)$ can be more complex than a fixed distribution $\mathcal{D}_i$ as it might be dependent on

the policy. For instance, in a Bayesian perspective, we can consider a meta distribution (distribution of distributions) and consider $\mathcal{D}_i$ is drawn from $\mathcal{D}_i^{\text{meta}}(\pi)$. In Section 3, $\mathcal{D}_i^{\text{meta}}(\pi)$ can be a posterior distribution over parameters like $\boldsymbol{\theta}_i$ based on $\pi$. Details are specified in later sections.

We study this problem under two different information settings that can both be unified under our above formulation. ❶ *Offline batch setting*: the full question set $\{q_i\}_{i=1}^N$ is known in advance, so the policy can allocate and adapt budgets globally across questions during execution. The outcome is summarized by a final per-question budget $B_\pi(q_i)$ for each $q_i$. ❷ *Online streaming setting*: Questions $\{q_1, q_2, \ldots, q_N\}$ arrive sequentially as i.i.d. draws from a latent distribution $\mathcal{D}$ and is not observable in advance. When $q_t$ arrives, the policy must choose a budget $B_t = B_\pi(q_t)$ immediately, without seeing future questions.

## 3. Offline PETS in the Batch Setting

In this section, we study the problem of optimal trajectory allocation in the offline setting. Here, allocation proceeds sequentially, and the availability of all questions enables us to gradually assess their relative difficulties throughout the process. Building on our notion of the self-consistency rate, we establish an interesting connection to the fruitful and well-developed literature on crowdsourcing. Table 1 summarizes the correspondence.

Specifically, under standard majority voting, the trajectory allocation problem is closely related to optimal budget allocation in crowdsourcing. Each reasoning answer $Y_i^{(b)}$ can be viewed as a noisy worker label for question $q_i$, while allocating additional traces corresponds to assigning more labeling effort to an item under a global budget constraint. As a result, the offline trajectory allocation problem can be formulated as a learning-while-allocating problem, and we can directly adopt the Bayesian framework of Chen et al. (2013). To sum up, the learner maintains a posterior over the answer distribution of each question, analogous to the posterior over item labels in crowdsourcing, and sequentially decides which question to sample next.

Beyond standard majority voting, we extend the Bayesian allocation framework to confidence-weighted aggregation, a strategy that has recently gained popularity in test-time scaling and self-consistency methods, e.g., (Fu et al., 2025a). We remark that the weighted majority voting formulation in (Fu et al., 2025a) differs from the worker-reliability weighting scheme considered in (Chen et al., 2013), and thus requires additional care when extending the crowdsourcing-based allocation framework.

**Bayesian Setup for Trajectory Allocation.** We formalize offline allocation as a Bayesian decision problem. For each question $q_i$, each sampled trace yields an answer

---

[2]If the maximum coordinate $\arg\max_{y \in \mathcal{Y}} \theta_y$ is much larger than the remaining parts, then it shows that this problem is relatively easier. Figure 5 in Appendix C.3 shows how $\boldsymbol{\theta}$ affects the self-consistency rate under different budget $B$.

*Table 1.* Connection to crowdsourcing.

| Crowdsourcing | Test-time self-consistency |
|---|---|
| Item $i$, Worker label $Y_i^{(b)}$ | Question $q_i$, Trace answer $Y_i^{(b)}$ |
| Adaptive worker assignment to items | Adaptive trajectory sampling for questions |
| Posterior over item labels (and worker reliabilities) | Posterior over answer distribution (and confidence weights) |
| Worker-specific reliability | Trace confidence without persistent Worker identity |
| Goal: infer the true label | Goal: infer the population majority answer $y_i^{\infty}$ (or $y_i^{W,\infty}$) |

$Y_i \in \mathcal{Y}$ and a confidence score $C_i = w(T_i) \in \mathbb{R}_+$. Let $\boldsymbol{\theta}_i \in \Delta^{M-1}$ denote the (unknown) answer distribution, and let $\boldsymbol{\mu}_i = (\mu_{i,1}, \ldots, \mu_{i,M}) \in \mathbb{R}^M$ be the (unknown) class-conditional confidence means (we fix $\sigma^2 = 1$ for simplicity). We make a natural assumption that answers follow a categorical distribution, and confidence is Gaussian with mean depending on the sampled answer. The joint likelihood for a single trace-induced pair $(Y_i, C_i)$ factorizes as

$$p(y, c \mid \boldsymbol{\theta}_i, \boldsymbol{\mu}_i) = p(y \mid \boldsymbol{\theta}_i)\, p(c \mid y, \boldsymbol{\mu}_i),$$

Therefore, the weighted population-optimal label is

$$y_i^{W,\infty} = \arg\max_{m \in \mathcal{Y}} \theta_{i,m} \mu_{i,m}.$$

The weighted self-consistency rate (2) can equivalently writes in the following form

$$\mathrm{SC}^W(q_i; B) = \mathbb{P}\Big(Y_i^{\mathrm{WMaj}}(B) = y_i^{W,\infty} \,\Big|\, \boldsymbol{\theta}_i, \boldsymbol{\mu}_i\Big), \quad (4)$$

Since latent parameters $(\boldsymbol{\theta}_i, \boldsymbol{\mu}_i)$ are unknown and ground-truth labels are unavailable, a direct quantification of the objective (3) is unclear. We thus adopt a Bayesian framework that maintains a posterior over the latent parameters, for us to quantify self-consistency under posterior belief.

**Modeling as an MDP process.** We model offline trajectory allocation as a finite-horizon Bayesian Markov Decision Process (MDP), following the crowdsourcing formulation of Chen et al. (2013), with adaptations to accommodate confidence-weighted traces. At stage $t$, the state $S^t$ summarizes the posterior beliefs over $(\boldsymbol{\theta}_i, \boldsymbol{\mu}_i)$ for each question via their respective posterior hyperparameters.

The action $i_t \in [N]$ selects a question to which one additional trace is allocated. After allocating a trace to question $i_t$, we observe $(y_t, c_t)$, which induces a conjugate Bayesian update of the posterior parameters in $S^t$. The process has a fixed horizon $H$, corresponding to the total trace budget

$B_{\text{total}}$. The terminal reward is defined as the sum of posterior self-consistency across questions.

$$\{\hat{y}_i\}_{i=1}^N = \arg\max_{\{\hat{y}_i\}} \sum_{i=1}^N \mathbb{P}(\hat{y}_i = y_i^{W,\infty} | S^H) \quad (5)$$

which measures the posterior probability of recovering the (weighted) population-majority answer for each question. Please refer to Appendix B.1 for further details on MDP formulation in our paper.

**Lemma 3.1** (Bayes-optimal terminal decision). *Given the terminal belief $S^H$, the Bayes-optimal decision for each question is therefore*

$$\hat{y}_i \in \arg\max_{m \in [M]} \mathbb{P}\Big(y_i^{W,\infty} = m \mid S_i^H\Big), \quad (6)$$

The posterior probability $\mathbb{P}(y_i^{W,\infty} = m \mid S_i^H)$ quantifies the posterior belief that class $m$ is the population majority label, which can be estimated via Monte Carlo sampling.

Under the terminal decision rule (6), we seek an allocation policy that maximizes the expected terminal utility:

$$V(S^0) = \mathbb{E}_\pi \Big[ \sum_{i=1}^N \max_{m \in [M]} \Pr\Big(y_i^{W,\infty} = m \mid S_i^H\Big) \Big], \quad (7)$$

where the expectation is taken over all sample paths induced by policy $\pi = (i_0, \ldots, i_{H-1})$, which sequentially selects a question to allocate one additional budget at each time step.

**Approximation of dynamic programming.** Exact dynamic programming of Equation (7) is intractable due to the exponentially growing belief space. Our PETS-Offline therefore similarly adopt the *Optimistic Knowledge Gradient (OKG)* heuristic (Chen et al., 2013), which selects the question with the largest optimistic one-step improvement in the terminal utility of Equation (7). We provide the details and full algorithm (Algorithm 2) in Appendix B.2.

## 4. Online PETS in the Streaming Setting

The key idea of online allocation is to assign different numbers of samples to questions with different answer distribution parameters $\boldsymbol{\theta}$ (i.e., difficulty vectors) in one shot upon seeing the question. Unlike the offline setting in Section 3, where $\boldsymbol{\theta}$ and voting weights can be estimated during the allocation process, the main challenge in the online regime is the lack of information about how the current question's difficulty compares to that of future questions, given the online nature. In this setting, we need to have access to *a prior distribution over question difficulties via additional training data that are assumed to be drawn from the same distribution*. Consequently, when a new question arrives, its

budget could be determined immediately based solely on its estimated difficulty label and the prior distribution.

To simplify the problem, we restrict our attention to the setting where we know that there will be $N$ questions in the upcoming estimation period, although we do not know their exact content or arrival times. This assumption is common in practice, since model deployers can often estimate the query volume over a given time window, for example, based on historical usage statistics, service-level forecasts, or system capacity planning.

**Mathematical formulation.** There are $N$ questions $\{q_1, q_2, \ldots, q_N\}$ arriving sequentially. Each question $q_t$ is associated with a difficulty vector $\boldsymbol{\theta}_t = \boldsymbol{\theta}(q_t)$ as defined in Section 2. With total budget $B_{\text{total}}$, upon observing $\boldsymbol{\theta}_t$ for question $t$, an allocation policy $\pi$ assigns a budget $B_t = B_\pi(q_t) = B_\pi(\boldsymbol{\theta}_t)$ without access to future questions or any intermediate feedback from other questions. Therefore, in the online setting, $\pi$ can only depend on the realized prior distribution $\mathcal{D}$ and the current difficulty label $\boldsymbol{\theta}_t$.

## 4.1. Execution Protocol

We here describe the execution protocol of PETS-Online. At a high level, the procedure consists of estimating ❶ the distribution of problem parameters, ❷ the mapping from each incoming question to its corresponding parameter grid, and ❸ solving a budget allocation optimization problem based on these estimates to obtain the final allocation plan. This gives a two-stage architecture: a training-time stage that builds a small library of difficulty grids and their representative self-consistency curves, and a test-time stage that maps each streaming question to one grid and allocates its budget in one shot.

**Estimation of parameters and sample distribution.** We estimate question difficulty in two steps. First, because the original $(M-1)$-dimensional difficulty parameter $\boldsymbol{\theta}_i$ can be high-dimensional in the multi-choice case, we reduce it to a two-dimensional surrogate $(a_i, b_i)$ via a Gaussian-probit approximation:

$$g_{a_i, b_i}(B) := \Phi(a_i \sqrt{B} + b_i),$$

where $\Phi$ is the standard normal CDF. This surrogate approximates the self-consistency curve $\text{SC}(\boldsymbol{\theta}_i; B)$ over different budgets $B$. The Gaussian approximation is motivated by applying a normal approximation to multinomial vote margins; in practice, $(a_i, b_i)$ is fitted by regression from a large pool of sampled answers for the training question. We then discretize the reduced space into $K$ difficulty grids with prototype parameters $\{\hat{\boldsymbol{\theta}}^j\}_{j=1}^K$; in the multi-choice implementation, $\hat{\boldsymbol{\theta}}^j$ is represented by a prototype probit curve $(\hat{a}_j, \hat{b}_j)$.

Second, we use a lightweight warm-up procedure to assign each incoming question to a grid, after which the corresponding budget is allocated in one shot. For a question $q$, we first draw 4 warm-up responses and let

$$C_q^{(4)} = (c_1, c_2, c_3, c_4)$$

be the sorted option-count vector in descending order, padded with zeros if fewer than four distinct options appear. These patterns induce a deterministic grid assignment

$$T_q = g(C_q^{(4)}) \in \{1, \ldots, K\},$$

This construction is independent of the total number of answer options, since at most four distinct options can appear in four warm-up samples. See Appendix C.3 and Appendix C.4 for more details.

**Optimization framework.** After discretization, the self-consistency rate in (3) for a question $q_i$ under budget $B$ can be approximated as[3]

$$\text{SC}(\boldsymbol{\theta}^{\text{buc}}(q_i); B) = \mathbb{P}\Big(Y_i^{\text{Maj}}(B) = y_i^\infty \mid \boldsymbol{\theta}^{\text{buc}}(q_i)\Big). \quad (8)$$

An online streaming policy $\pi$ is therefore fully specified by a vector of integer budgets $\{B_1, \ldots, B_K\}$, where any question assigned to grid $j$ receives budget $B_j$. Over a horizon of $N$ streaming questions with total budget $B_{\text{total}}$, the optimal allocation problem can be rewritten as

$$\max_{\{B_j\}_{j=1}^K} \sum_{j=1}^K \hat{p}_j \, \text{SC}(\hat{\boldsymbol{\theta}}^j; B_j) \quad \text{s.t.} \quad \sum_{j=1}^K \hat{p}_j B_j \leq \bar{B}_{\text{total}}, \quad (9)$$

where $\bar{B}_{\text{total}} := B_{\text{total}}/N$ is the average budget per round, $\hat{p}_j$ is the estimated distribution, and $\hat{\boldsymbol{\theta}}^j$ is the representation parameter of the grid $j$. Finally, we note that although (9) is a static optimization problem, our final policy is a dynamic streaming policy which calls (9) at every round; we refer to Appendix C.1 for more details.

## 4.2. Optimal Budget Allocation

In this section, we present an efficient greedy algorithm that solves the integer program (9). We start with the binary-choice case ($M = 2$) in Algorithm 1, and extend it to the multi-choice case in Appendix C.3. For a binary-choice question $q_i$, the choice set contains only two choices $\mathcal{Y} = \{0, 1\}$, and the difficulty vector reduces to a scalar parameter $\theta = \mathbb{P}[Y_i = 1] \in [0, 1]$. Under $B$ i.i.d. samples and majority voting (with random tie-breaking when $B$ is

---

[3]A weighted version can be defined analogously. Since weights are difficult to estimate in the online setting, in experiments, we use the unweighted version for policy optimization and apply weighted majority voting only at inference time.

even), we obtain

$$\mathbb{P}[Y_i^{\text{Maj}}(B) = 1 \mid \theta]$$
$$= \begin{cases} \sum_{b=(B+1)/2}^{B} \binom{B}{b} \theta^b (1-\theta)^{B-b}, & B \text{ odd} \\ \sum_{b=B/2+1}^{B} \binom{B}{b} \theta^b (1-\theta)^{B-b} \\ \quad + \frac{1}{2} \binom{B}{B/2} \theta^{B/2} (1-\theta)^{B/2}, & B \text{ even} \end{cases} \quad (10)$$

and $\mathbb{P}[Y_i^{\text{Maj}}(B) = 0 \mid \theta] = 1 - \mathbb{P}[Y_i^{\text{Maj}}(B) = 1 \mid \theta]$. The corresponding self-consistency rate (1) is then

$$\text{SC}(\theta; B) = \max\{\mathbb{P}[Y_i^{\text{Maj}}(B) = 1 \mid \theta], \mathbb{P}[Y_i^{\text{Maj}}(B) = 0 \mid \theta]\}, \quad (11)$$

where $\text{SC}(\theta; 0) = \frac{1}{2}$. We define the marginal gain $R(\theta, n)$ at budget level $n$ for a question with difficulty parameter $\theta$ as the increase in self-consistency rate resulting from allocating one additional unit of budget:

$$R(\theta, n) := \text{SC}(\theta; n+1) - \text{SC}(\theta; n).$$

We propose a greedy yet optimal algorithm (Algorithm 1) that repeatedly allocates budget to the grid with the largest current marginal improvement.

---

**Algorithm 1** Greedy Budget Allocation Algorithm

---

**Require:** Grid $(\theta^j)_{j=1}^K$, probabilities $(p_j)_{j=1}^K$,
    average per-round budget $\bar{B}_{\text{total}}$
 1: Initialize current budget $B_j \leftarrow 0$ for all $j \in [K]$.
 2: Initialize current marginal gain $\delta_j \leftarrow R(\theta^j, 0)$ for all $j \in [K]$.
 3: Initialize total used budget $T \leftarrow 0$.
 4: **while** $T \leq \bar{B}_{\text{total}}$ **do**
 5: $\quad j^\star \leftarrow \arg\max_j \delta_j$
 6: $\quad$ **if** $B_{j^\star} = 0$ **then**
 7: $\quad\quad B_{j^\star} \leftarrow 1, T \leftarrow T + p_{j^\star}$
 8: $\quad$ **else**
 9: $\quad\quad B_{j^\star} \leftarrow B_{j^\star} + 2, T \leftarrow T + 2p_{j^\star}$
10: $\quad$ **end if**
11: $\quad$ Update $\delta_{j^\star} \leftarrow R(\theta_{j^\star}, B_{j^\star})$
12: **end while**

---

**Theorem 4.1.** *Algorithm 1 outputs an optimal solution to the discretized online allocation problem (9) in expectation, with a randomized rounding rule.*[4]

### 4.2.1. CONNECTION WITH THE OFFLINE CASE

Although the offline and online cases rely on different allocation procedures, their budget proportions become nearly identical as the budget grows. As shown in Figure 2, increasing the average per-question budget leads the two allocations to converge to similar proportions. Further discussion and theoretical support are provided in Appendix D.

---

[4]This is because the remaining budget may not suffice for another iteration. See Appendix C.1 for details.

## 5. Experiment

In this section, we conduct evaluation of PETS on widely used knowledge and reasoning benchmarks, including GPQA-Diamond (Rein et al., 2024), AIME 24 (hug, 2025), AIME 25 (mat, 2025), Brumo 25 (mat, 2026a), and HMMT Feb 25 (mat, 2026b). We consider popular reasoning LLMs: Qwen3-4B-Thinking and Qwen3-30B-A3B-Thinking (Yang et al., 2025), gpt-oss-20b and gpt-oss-120b (Agarwal et al., 2025), and QwenLong-L1.5-30B-A3B (Shen et al., 2025). Following our offline budget allocation (Section 3) and online budget allocation (Section 4), we evaluate PETS under the offline and online scenarios.

For each dataset, we sample 128 responses per question from each model. To define a maximum allocation budget of 64 traces per question as the finite proxy for the infinite sampling limit, we uniformly subsample 64 responses from the 128. We repeat this subsampling process 30 times and report the mean performance.

We also include confidence weighted version in our experiment, where we follow Fu et al. (2025b) to compute tail trace confidence $C_i = w(T_i)$ as the weight for weighted majority voting at aggregation time (In PETS-offline we also involve it at sampling time as shown in Section 3).

We evaluate self-consistency rate as defined in Equation 1 and Equation 2. Our primary metric is the number of traces required to reach full self-consistency $(= 1)$, where the majority answer matches the population majority answer $y_i^\infty$ (or $y_i^{W,\infty}$ in the weighted version), approximated using 64 traces. We also report accuracy on the dataset. More details and full results are provided in Appendix E.

In both offline and online settings, we compare against two baselines: uniform budget allocation across questions (Uniform) and the confidence-guided allocation (Conf. guided) method of DeepConf (Fu et al., 2025a).

### 5.1. PETS for Offline Budget Allocation

We first evaluate PETS-Offline, where all questions are available to the budget-allocation optimizer. Following Section 2, we consider both unweighted and confidence-weighted settings, and only compare methods within the same setting. We report self-consistency and accuracy at the trace count where PETS-Offline first reaches full self-consistency $(= 1)$. Results are shown in Table 2 and Figure 3.

Across models and datasets, ❶ PETS-Offline consistently reduces the number of traces required to reach full self-consistency by up to 75% compared to uniform allocation, while achieving higher self-consistency at any budget (Figure 3). ❷ When the population majority aligns with the true answer, improved self-consistency directly translates into higher accuracy. ❸ Confidence-weighted majority voting

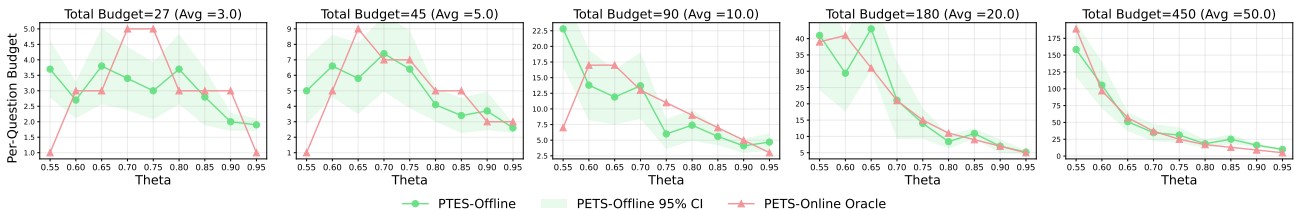

*Figure 2.* Budget allocation plan of the offline and online settings on 9 simulated binary choice questions, $\mathcal{Y} = \{1, 2\}$. Each question is associated with a $\theta = \max(\theta_1, \theta_2)$, and larger theta indicates easier questions.

*Table 2.* PETS-Offline Results. We compare PETS against confidence-guided and uniform sample budget allocation. "(conf)" denotes the trace confidence-weighted variant. # traces reports the number of sampled traces required to reach full consistency (consistency = 1); Con. and Acc. report each method's achieved consistency and accuracy evaluated at the trace count where PETS-Off. attains consistency 1.

| Dataset | Method | Qwen3-4B | | | Qwen3-30B | | | Qwen-Long | | | GPT-20B | | | GPT-120B | | |
|---|---|---|---|---|---|---|---|---|---|---|---|---|---|---|---|---|
| | | # traces ↓ | Con. ↑ | Acc. ↑ | # traces | Con. | Acc. | # traces | Con. | Acc. | # traces | Con. | Acc. | # traces | Con. | Acc. |
| GPQA | PETS-Off. | **2780** | **1.00** | 69.7 | **2607** | **1.00** | 71.8 | **2513** | **1.00** | 76.3 | **3180** | **1.00** | 75.7 | **2580** | **1.00** | 82.5 |
| | PETS-Off. (conf) | 3667 | 0.99 | 70.3 | 3687 | 0.99 | 72.1 | 2887 | 0.99 | 77.0 | 3693 | 0.99 | 76.2 | 3540 | 0.99 | 82.1 |
| | Conf. guided | 10652 | 0.96 | 68.5 | 9702 | 0.96 | 71.2 | 8158 | 0.96 | 76.1 | 10058 | 0.96 | 75.4 | 8474 | 0.97 | 81.6 |
| | Uniform | 11013 | 0.96 | 68.9 | 10367 | 0.96 | 71.5 | 9973 | 0.96 | 76.5 | 10853 | 0.96 | 75.1 | 10393 | 0.97 | 81.5 |
| | Uniform (conf) | 11453 | 0.95 | 69.9 | 11607 | 0.95 | 72.2 | 11400 | 0.95 | 76.4 | 11193 | 0.96 | 75.6 | 10500 | 0.96 | 80.8 |
| AIME25 | PETS-Off. | **212** | **1.00** | 83.3 | **190** | **1.00** | 88.6 | **152** | **1.00** | 90.0 | **181** | **1.00** | 90.0 | 212 | **1.00** | 93.9 |
| | PETS-Off. (conf) | 257 | 0.98 | **84.0** | 223 | 0.97 | 86.7 | 203 | 0.97 | **90.4** | 251 | 0.97 | 89.7 | **211** | 0.98 | 93.3 |
| | Conf. guided | 402 | 0.95 | 81.3 | 312 | 0.95 | 87.5 | 234 | 1.00 | 90.0 | 408 | 0.97 | 89.2 | 732 | 0.95 | 91.3 |
| | Uniform | 470 | 0.94 | 82.1 | 545 | 0.96 | 86.1 | 288 | 0.97 | 89.3 | 410 | 0.95 | 90.2 | 681 | 0.95 | 91.3 |
| | Uniform (conf) | 610 | 0.91 | 83.3 | 763 | 0.93 | 85.6 | 752 | 0.94 | 88.9 | 618 | 0.94 | 89.0 | 911 | 0.94 | 91.9 |
| AIME24 | PETS-Off. | **259** | **1.00** | 63.1 | 135 | **1.00** | 70.0 | 83 | **1.00** | 70.0 | **200** | **1.00** | 73.3 | 184 | **1.00** | 74.4 |
| | PETS-Off. (conf) | 369 | 0.96 | **66.0** | 120 | **1.00** | 69.9 | 91 | 0.99 | 69.9 | 232 | 0.97 | 72.6 | 218 | 0.99 | 73.7 |
| | Conf. guided | 648 | 0.95 | 62.9 | 138 | 0.99 | 70.0 | 84 | 1.00 | 70.0 | 546 | 0.98 | 72.5 | 378 | 0.96 | 73.3 |
| | Uniform | 861 | 0.94 | 63.0 | 202 | 0.98 | 69.6 | 96 | 0.99 | 69.9 | 665 | 0.96 | 71.9 | 448 | 0.97 | 73.7 |
| | Uniform (conf) | 942 | 0.92 | 63.4 | 179 | 0.97 | 68.6 | 115 | 0.98 | 69.7 | 1206 | 0.94 | 70.8 | 535 | 0.95 | 71.9 |
| HMMT | PETS-Off. | **464** | **1.00** | 51.6 | 381 | **1.00** | 64.7 | 363 | **1.00** | 71.7 | 329 | **1.00** | 83.3 | 329 | **1.00** | 83.3 |
| | PETS-Off. (conf) | 579 | 0.96 | **52.0** | **361** | 0.97 | 65.2 | 371 | 0.97 | **73.2** | 324 | 0.99 | 83.1 | 324 | 0.99 | 83.1 |
| | Conf. guided | 1206 | 0.92 | 48.3 | 894 | 0.95 | 58.1 | 984 | 0.91 | 64.6 | 732 | 0.94 | 80.5 | 882 | 0.93 | 81.1 |
| | Uniform | 1133 | 0.90 | 50.0 | 1219 | 0.91 | 64.7 | 1227 | 0.91 | 68.8 | 916 | 0.94 | **83.6** | 916 | 0.94 | **83.6** |
| | Uniform (conf) | 1383 | 0.87 | 50.6 | 1089 | 0.91 | 65.1 | 1165 | 0.90 | 71.7 | 902 | 0.93 | 82.6 | 902 | 0.93 | 82.6 |
| BRUMO | PETS-Off. | **280** | **1.00** | 75.4 | **148** | **1.00** | 86.7 | 149 | **1.00** | 86.7 | 253 | **1.00** | 92.1 | 253 | **1.00** | 92.1 |
| | PETS-Off. (conf) | 292 | 0.98 | **77.3** | 162 | 0.98 | 86.1 | 150 | 0.98 | 87.8 | 217 | 0.98 | 92.4 | 217 | 0.98 | **92.4** |
| | Conf. guided | 654 | 0.94 | 71.3 | 252 | 0.95 | 80.8 | 348 | 0.98 | 85.0 | 336 | 0.96 | **93.8** | 552 | 0.95 | 90.5 |
| | Uniform | 826 | 0.94 | 73.2 | 237 | 0.96 | 83.7 | 320 | 0.97 | 87.3 | 776 | 0.94 | 90.1 | 776 | 0.94 | 90.1 |
| | Uniform (conf) | 589 | 0.94 | 75.7 | 310 | 0.95 | 83.6 | 421 | 0.97 | 87.2 | 574 | 0.95 | 90.7 | 574 | 0.95 | 90.7 |

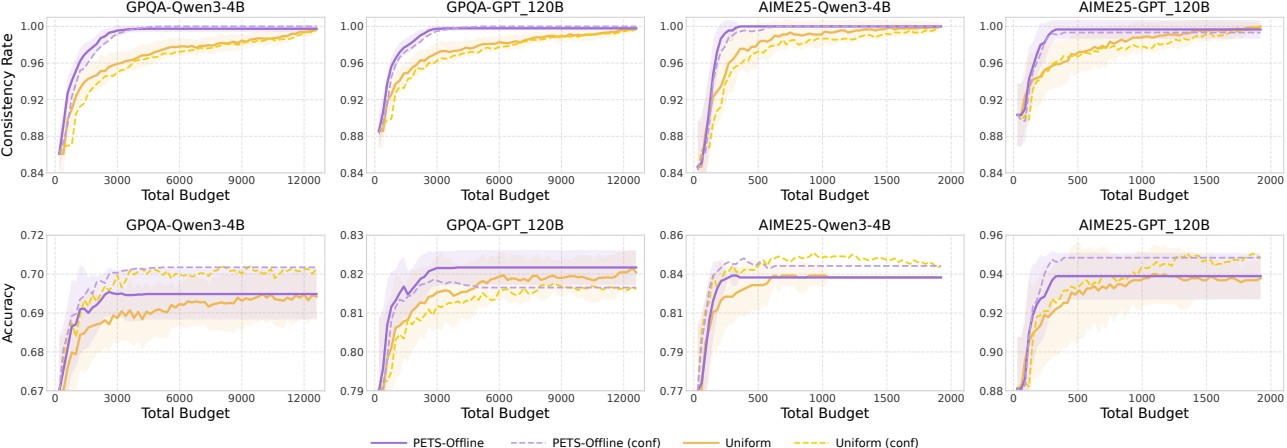

*Figure 3.* Budget allocation curve in the offline setting. "(conf)" denotes the trace confidence-weighted variant. Consistency is computed within each matched-variant comparison group: PETS-Offline vs. Uniform, and PETS-Offline (conf) vs. Uniform (conf).

*Table 3.* PETS-Online Results. We compare PETS against confidence-guided and uniform sample budget allocation. "(conf)" denotes the trace confidence-weighted variant. # traces reports the number of sampled traces required to reach full consistency (consistency = 1); Con. and Acc. report each method's achieved consistency and accuracy evaluated at the trace count where PETS-On. attains consistency 1.

| Dataset | Method | Qwen3-4B | | | Qwen3-30B | | | Qwen-Long | | | GPT-20B | | | GPT-120B | | |
|---|---|---|---|---|---|---|---|---|---|---|---|---|---|---|---|---|
| | | # traces ↓ | Con. ↑ | Acc. ↑ | # traces | Con. | Acc. | # traces | Con. | Acc. | # traces | Con. | Acc. | # traces | Con. | Acc. |
| GPQA | PETS-On. | **4662** | **1.00** | 68.8 | **3826** | **1.00** | 71.6 | 4852 | **1.00** | 76.3 | **6826** | **1.00** | 75.9 | **4172** | **1.00** | **82.4** |
| | PETS-On. (conf) | 4888 | **1.00** | **69.6** | 4357 | 0.99 | **71.8** | **4146** | **1.00** | **76.9** | 7269 | 0.99 | 76.1 | 5094 | 0.99 | 81.6 |
| | Conf. guided | 7623 | 0.97 | 68.2 | 7361 | 0.98 | 70.5 | 7782 | 0.98 | 75.4 | 9355 | 0.98 | 74.3 | 7162 | 0.97 | 80.5 |
| | Uniform | 9520 | 0.97 | 68.1 | 8456 | 0.97 | 71.2 | 9195 | 0.97 | 76.0 | 9699 | 0.98 | 76.1 | 8109 | 0.98 | 81.8 |
| | Uniform (conf) | 9856 | 0.97 | 69.4 | 9257 | 0.96 | **72.0** | 9526 | 0.97 | 76.4 | 9895 | 0.98 | **76.3** | 9593 | 0.97 | 81.4 |
| AIME25 | PETS-On. | **194** | **1.00** | 75.0 | 504 | **1.00** | **83.5** | 218 | **1.00** | 85.0 | **195** | **1.00** | 85.0 | 454 | **1.00** | **91.7** |
| | PETS-On. (conf) | 287 | 0.95 | **76.3** | 535 | 0.97 | 82.5 | 681 | 0.96 | 85.7 | 330 | 0.96 | 84.8 | 493 | 0.98 | **92.3** |
| | Conf. guided | 232 | 0.96 | 75.0 | 272 | 0.96 | 82.5 | **146** | 0.97 | **86.0** | 385 | 0.94 | 84.5 | 232 | 0.96 | 89.5 |
| | Uniform | 290 | 0.95 | 74.8 | 415 | 0.95 | 81.0 | 229 | 0.95 | 84.7 | 221 | 0.96 | **86.0** | 416 | 0.95 | 89.2 |
| | Uniform (conf) | 452 | 0.92 | **76.3** | 486 | 0.94 | 81.2 | 495 | 0.93 | 83.5 | 349 | 0.93 | 85.2 | 459 | 0.94 | 89.8 |
| AIME24 | PETS-On. | **170** | **1.00** | 56.2 | 81 | **1.00** | 65.0 | 83 | **1.00** | 65.0 | 130 | **1.00** | 70.0 | 107 | **1.00** | 69.8 |
| | PETS-On. (conf) | 185 | 0.96 | **60.0** | 81 | **1.00** | 64.8 | 86 | 0.99 | 64.8 | **104** | 0.99 | 68.8 | **92** | **1.00** | 69.5 |
| | Conf. guided | 292 | 0.96 | 57.0 | **80** | 0.99 | 64.5 | **80** | **1.00** | 65.0 | 106 | 0.98 | 68.0 | 170 | 0.98 | 68.5 |
| | Uniform | 393 | 0.94 | 56.8 | 83 | **1.00** | 64.7 | 82 | **1.00** | 65.0 | 237 | 0.98 | 67.7 | 205 | 0.97 | 67.0 |
| | Uniform (conf) | 405 | 0.94 | 58.8 | 83 | **1.00** | 64.7 | 101 | 0.99 | 64.8 | 171 | 0.98 | 68.0 | 144 | 0.98 | 67.7 |
| HMMT | PETS-On. | **531** | **1.00** | 48.5 | 825 | **1.00** | 56.2 | 872 | **1.00** | 67.0 | 248 | **1.00** | 85.0 | **221** | **1.00** | 85.0 |
| | PETS-On. (conf) | 615 | 0.93 | 46.8 | 761 | 0.95 | 56.8 | 1061 | 0.95 | **68.8** | 233 | 0.97 | **85.5** | 263 | 0.96 | **86.5** |
| | Conf. guided | 727 | 0.91 | 43.0 | 623 | 0.94 | 52.5 | **666** | 0.99 | 64.5 | 436 | 0.91 | 81.5 | 382 | 0.93 | 80.0 |
| | Uniform | 792 | 0.89 | 46.2 | 813 | 0.93 | 56.2 | 803 | 0.96 | 66.7 | 475 | 0.92 | 83.3 | 309 | 0.95 | 84.8 |
| | Uniform (conf) | 931 | 0.89 | 45.7 | 786 | 0.92 | 56.7 | 959 | 0.95 | 67.8 | 409 | 0.92 | 83.2 | 458 | 0.91 | 85.0 |
| BRUMO | PETS-On. | 601 | **1.00** | 63.2 | 125 | **1.00** | 80.0 | 130 | **1.00** | 80.2 | 213 | **1.00** | 95.0 | 383 | **1.00** | 88.8 |
| | PETS-On. (conf) | 494 | 0.97 | 64.7 | **121** | 0.99 | 79.5 | 138 | 0.99 | **81.5** | 162 | 0.99 | 94.7 | **241** | 0.99 | 89.7 |
| | Conf. guided | **398** | 0.93 | 58.0 | 202 | 0.96 | 73.5 | 142 | 0.95 | 76.5 | 368 | 0.89 | 88.0 | 424 | 0.92 | 85.5 |
| | Uniform | 613 | 0.94 | 61.5 | 172 | 0.96 | 77.7 | 167 | 0.97 | 80.7 | 245 | 0.95 | 91.7 | 521 | 0.92 | 86.3 |
| | Uniform (conf) | 519 | 0.93 | **65.5** | 159 | 0.96 | 77.7 | 189 | 0.98 | 81.3 | 184 | 0.95 | 91.7 | 353 | 0.94 | 87.2 |

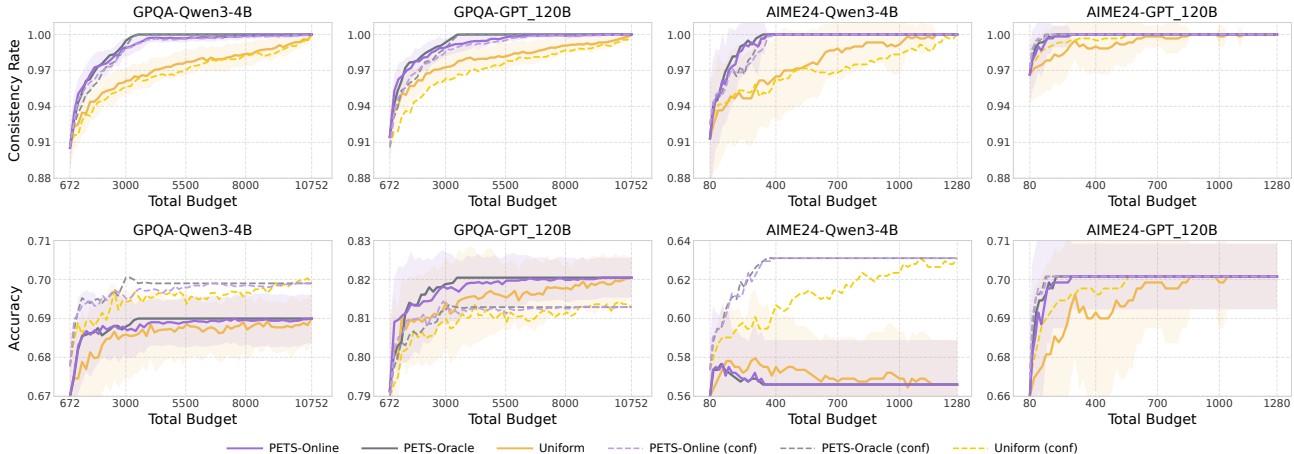

*Figure 4.* Budget allocation curve in the online setting. "(conf)" denotes the trace confidence-weighted variant. Consistency is computed within each matched-variant comparison group: PETS-Online vs. Uniform, and PETS-Online (conf) vs. Uniform (conf). Oracle variant assumes access to the latent parameter $\theta$, while in the online setting, $\theta$ is learnt from a training dataset.

further shifts the population majority towards the ground truth, thus resulting in improved accuracy.

## 5.2. PETS for Online Budget Allocation

We next evaluate PETS in the online case, where questions arrive sequentially, and allocation decisions are irreversible. Besides Uniform, we also compare PETS-Online

with PETS-Oracle, which can access the latent difficulty $\theta$ and thus serves as an upper bound without estimation error.

As in the offline setting, we also consider both unweighted and confidence-weighted versions. PETS-Online allocates budget on the fly using only observed traces. Results are shown in Table 3 and Figure 4.

Consistent with offline results, ❶ PETS-Online substantially

*Table 4.* Effect of Oracle KMeans cluster number $k$ on Oracle accuracy. The Oracle uses KMeans clustering on ground-truth $(a, b)$ parameters to partition questions into $k$ difficulty bins. Predictor and Baseline results are unaffected by this parameter.

| Dataset | Model | $k=2$ | $k=3$ | $k=5$ | $k=8$ |
|---------|-------|-------|-------|-------|-------|
| GPQA | Qwen3-4B | 0.6905 | 0.6905 | **0.6964** | 0.6905 |
| | GPT-120B | 0.8036 | 0.8036 | 0.8214 | **0.8274** |
| AIME 2024 | Qwen3-4B | **0.9500** | 0.9000 | 0.9000 | 0.9000 |
| | GPT-120B | 1.0000 | 1.0000 | 1.0000 | 1.0000 |

*Table 5.* Effect of Predictor difficulty bin number $k$ on Predictor accuracy. The Predictor assigns each question to one of $k$ bins based on the number of distinct answers among $K_0=4$ warm-up.

| Dataset | Model | $k=2$ | $k=3$ | $k=4$ | $k=5$ |
|---------|-------|-------|-------|-------|-------|
| GPQA | Qwen3-4B | **0.6845** | **0.6845** | **0.6845** | 0.6786 |
| | GPT-120B | **0.8214** | 0.8095 | 0.8095 | 0.8095 |
| AIME 2024 | Qwen3-4B | **0.9500** | 0.9000 | 0.9000 | 0.9000 |
| | GPT-120B | 1.0000 | 1.0000 | 1.0000 | 1.0000 |

reduces the traces required to reach full self-consistency, ❷ leading to improved accuracy. ❸ Confidence-weighted self-consistency further boosts accuracy by reshaping the population majority. ❹ PETS-Online closely matches PETS-Oracle in both self-consistency and accuracy, indicating that the estimation procedure in Appendix C.4 is effective.

We further report Pass@1 and MV@128 accuracy as lower and upper reference bounds, respectively, for all methods. The results are shown in Figures 16 and 17.

## 5.3. Additional Results

In this section, we provide two ablations for the online difficulty discretization. Tables 4 and 5 show that both the oracle allocation, which clusters ground-truth $(a, b)$ parameters, and the practical predictor, which uses only $K_0=4$ warm-up samples, are not highly sensitive to the number of difficulty bins. Accuracy varies only modestly across choices of $k$, and in several AIME 2024 settings remains unchanged, suggesting that coarse difficulty grouping is sufficient for the online allocation policy.

We also report wall-clock allocation overhead in Table 6. The online predictor adds only a few seconds of preprocessing and budget assignment on the evaluated datasets, while the offline OKG allocator is substantially more expensive due to sequential Monte Carlo allocation. This supports the intended use of PETS-Online as a practical streaming policy: it preserves the adaptive allocation behavior of PETS while keeping the decision-time overhead small.

## 6. Related Work

**Test-time scaling and efficient reasoning.** Test-time scaling improves reasoning either by extending chain-of-thought

*Table 6.* Wall-clock runtime (seconds) of different budget allocation methods. **Online Predictor** trains bucket statistics on 30 questions and then assigns each test question a budget via $K_0=4$ warm-up samples. **Online Uniform** applies a fixed per-question budget with no training. **Offline OKG** performs sequential Monte Carlo allocation ($B=16$ per question, $n_{\text{samples}}=500$).

| Dataset | Model | Online Predictor | Online Uniform | Offline OKG |
|---------|-------|------------------|----------------|-------------|
| GPQA | Qwen3-4B | 4.73 | $< 0.01$ | 212.2 |
| | GPT-120B | 4.68 | $< 0.01$ | 204.1 |
| AIME 2024 | Qwen3-4B | 1.55 | 0.08 | 36.0 |
| | GPT-120B | 1.19 | 0.03 | 39.0 |

trajectories or by sampling multiple trajectories in parallel (Snell et al., 2024; Welleck et al., 2024; Wei et al., 2022). Self-consistency and Best-of-$N$ aggregate such samples by voting or selection (Wang et al., 2022; Brown et al., 2024). Because naive scaling is costly, efficient-reasoning methods shorten traces, impose token budgets, prune trajectories, or filter samples using confidence and difficulty estimates rather than directly optimizing allocation (Muennighoff et al., 2025; Chen et al., 2024a; Fu et al., 2025a; Wang et al., 2025a).

**Crowdsourcing and adaptive allocation.** Our formulation is also connected to crowdsourcing and budgeted labeling: Dawid–Skene models noisy workers and item difficulty (Dawid & Skene, 1979), while adaptive assignment and optimistic knowledge gradient methods allocate labels by expected value of information (Sheng et al., 2008; Chen et al., 2013). Related work also studies aggregation, adaptive stopping, and compute allocation across queries, tokens, or rollouts (Komiyama et al., 2025; Chen et al., 2024b; Zuo & Zhu, 2025; Lin et al., 2025; Wang et al., 2025b). Unlike reward- or judge-based selection, which can suffer from reward hacking or miscalibration (Huang et al., 2025a; Gao et al., 2024), PETS optimizes self-consistency against an infinite-compute voting oracle.

## 7. Conclusion

We present PETS, a principled framework that improves the efficiency of parallel sampling with self-consistency without sacrificing performance. Across challenging benchmarks and state-of-the-art reasoning models, PETS consistently increases self-consistency and accuracy while substantially reducing the number of required reasoning trajectories. Our experiments reveal that while majority voting can approximate the population majority and improve accuracy when the population majority matches the true answer, but when the majority is systematically wrong, additional sampling provides little benefit, revealing a limitation of allocation-only approaches. Finally, while PETS-Online estimates the latent question difficulty $\theta$ via a short warm-up phase, an important next step is to train models to predict $\theta$ directly from the question prior to generation.

## Acknowledgement

Z. Deng, Y. Han, and T. Chen are grateful for the kind support of Renaissance Philanthropy AI4Math fund.

## Impact Statement

This paper presents work whose goal is to advance the field of LLM reasoning. There are many potential societal consequences of our work, none of which we feel must be specifically highlighted here.

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

# A. Additional Related Work

**Test-Time Scaling.** Since the emergence of long-reasoning models such as GPT-o1 (Jaech et al., 2024) and DeepSeek-R1 (Guo et al., 2025), test-time scaling (Snell et al., 2024; Welleck et al., 2024; Chen et al., 2025) has gained traction as a way to improve performance by allocating substantially more computation (e.g., reasoning tokens) at inference time. One line of work scales the length of chain-of-thought (CoT) trajectory by extending the reasoning process (Wei et al., 2022); representative examples include GPT-o1 (Jaech et al., 2024), DeepSeek-R1 (Guo et al., 2025), Kimi K2 (Team et al., 2025), Qwen3 (Yang et al., 2025), gpt-oss (Agarwal et al., 2025), and Gemini 3 Pro (gem). Another line of work scales via parallel sampling of multiple trajectories, leveraging self-consistency (Wang et al., 2022) or Best-of-$N$ selection (Brown et al., 2024) and aggregating outputs (e.g., by majority vote). In this paper, PETS focuses on the intersection of these two lines and considers the efficient selection of multiple long reasoning trajectories.

**Efficient Reasoning.** Although test-time scaling can improve performance, its increased inference cost has motivated works on efficient reasoning. Along the line of CoT extension, Muennighoff et al. (2025) proposes budget forcing to control test-time compute by terminating the model's reasoning once a preset budget is reached. Other approaches aim to elicit shorter yet effective reasoning by fine-tuning with condensed CoT traces (Chen et al., 2024a; Luo et al., 2025; Hou et al., 2025; Zhang et al., 2025). Along the parallel sampling direction, several methods introduce more efficient variants of self-consistency that reduce the number of trajectories while maintaining accuracy (Li et al., 2024; Wan et al., 2025; Fu et al., 2024). Most closely related to our work, DeepConf (Fu et al., 2025a) leverages per-trajectory confidence to filter low-confidence traces and terminate generation when confidence falls below a threshold.

**Crowdsourcing.** Crowdsourcing has been extensively studied as a budgeted labeling problem, with classical models such as Dawid–Skene capturing item difficulty and annotation noise (Dawid & Skene, 1979). Building on these probabilistic foundations, a line of work has focused on adaptive task assignment and budget allocation, where items are sequentially selected for labeling based on expected benefit (Sheng et al., 2008). In particular, Chen et al. (2013) formulate crowdsourcing as a Bayesian decision process and propose the optimistic knowledge gradient (OKG) policy, providing a principled approach to optimal budget allocation under majority-style aggregation. A central ingredient in these approaches is uncertainty quantification (Deng et al., 2025; Zollo et al., 2024b;a; Deng et al., 2023): reliable estimates of uncertainty determine which items require additional labels, which workers or responses should be trusted, and when the current aggregate label is sufficiently confident. In this sense, crowdsourcing systems depend on uncertainty quantification to trade off exploration, such as collecting more annotations for ambiguous items, against exploitation, such as allocating budget to decisions with high expected utility. Conversely, crowdsourcing can be viewed as a mechanism for reducing epistemic uncertainty by strategically acquiring diverse and informative human judgments. However, these methods are primarily designed for human annotators, and few works have extended them to the setting of test-time consistency in large language models.

**Computational Resources Allocation.** A common family of approaches generates multiple candidate answers and then selects or aggregates them. Komiyama et al. (2025) analyze majority-vote best-of-$N$ in the asymptotic regime $N \to \infty$, and propose an adaptive generation/early-stopping rule based on answer agreement, including extensions to weighted multi-LLM ensembles. Complementarily, Chen et al. (2024b) provide simple two-stage aggregation schemes (knockout and league) with provable failure-probability decay as test-time compute grows, using the LLM itself as a black-box generator and pairwise comparator.

When selection relies on an imperfect reward/judge model, scaling can backfire. Huang et al. (2025a) formalize inference-time alignment through Best-of-$N$ sampling and show that large $N$ can induce reward hacking; they propose a pessimistic alternative with scaling-monotonic guarantees. Beyond per-question scaling, allocating compute across queries is crucial under a global budget. Zuo & Zhu (2025) cast Test-time scaling allocation as a bandit learning problem that adapts compute online to query difficulty and solvability. Other work targets different compute axes: Lin et al. (2025) study budget tokens within a single reasoning process via planning and uncertainty-aware scheduling, while Wang et al. (2025b) study efficient test-time scaling for search by allocating rollouts at the direction level to avoid candidate-count bias.

Finally, several Bayesian works address reliability and evaluation. Yao et al. (2024) treat multiple LLMs as annotators and extend Dawid–Skene for uncertainty-aware truth aggregation. Gao et al. (2024) calibrate win-rate estimates from LLM evaluators, mitigating bias in LLM-as-judge comparisons. In contrast to reward-based TTS, our work focuses on budget allocation to optimize self-consistency defined against an infinite-compute (weighted) voting oracle, enabling principled allocation without depending on a potentially misspecified reward model.

# B. Appendix of Offline Case (Section 3)

## B.1. Bayesian MDP

We place independent priors on $\boldsymbol{\theta}_i$ and $\boldsymbol{\mu}_i$:

$$\boldsymbol{\theta}_i \sim \mathrm{Dir}(\boldsymbol{\alpha}_i^0), \qquad \boldsymbol{\alpha}_i^0 \in \mathbb{R}_+^M, \qquad \text{and} \qquad \mu_{i,m} \sim \mathcal{N}(\beta_{i,m}^0, v_{i,m}^0), \ v_{i,m}^0 > 0, \ m \in [M],$$

with $\boldsymbol{\theta}_i \perp \boldsymbol{\mu}_i$ a priori. The Dirichlet parameters $\alpha_{i,m}^0$ can be viewed as pseudo-counts for class $m$. The Gaussian parameters $(\beta_{i,m}^0, v_{i,m}^0)$ specify the prior mean and variance of the class-conditional confidence mean $\mu_{i,m}$.

At stage $t$, let $S^t$ denote the global belief state, which collects posterior parameters for all questions:

$$S^t \triangleq \big(\{\boldsymbol{\alpha}_i^t\}_{i=1}^N, \ \{\boldsymbol{\beta}_i^t\}_{i=1}^N, \ \{\boldsymbol{v}_i^t\}_{i=1}^N\big), \qquad \boldsymbol{\beta}_i^t = (\beta_{i,1}^t, \ldots, \beta_{i,M}^t), \ \boldsymbol{v}_i^t = (v_{i,1}^t, \ldots, v_{i,M}^t).$$

In the fully observed offline setting, each observation reveals $(Y_i, C_i)$ for the sampled trace, so $\boldsymbol{\alpha}_i^t$ admits a Dirichlet-multinomial conjugate update and $(\boldsymbol{\beta}_i^t, \boldsymbol{v}_i^t)$ admit Gaussian conjugate updates conditioned on $Y_i$ .

At each state $S^t$, we would select a question $i_t$ and observe $(Y_t, C_t) = (y_t, c_t)$, the Dirichlet parameters update via the conjugate Categorical–Dirichlet rule

$$\boldsymbol{\alpha}_i^{t+1} = \boldsymbol{\alpha}_i^t + \boldsymbol{\delta}_{y_t}, \qquad \boldsymbol{\alpha}_j^{t+1} = \boldsymbol{\alpha}_j^t \ \forall j \neq i,$$

where $\boldsymbol{\delta}_m \in \mathbb{R}^M$ is the one-hot vector with a $1$ at entry $m$.

For the confidence model, only the component $\mu_{i,y_t}$ is updated. Recall that under state $S^t$, we maintain

$$\mu_{i,m} \mid S^t \sim \mathcal{N}(\beta_{i,m}^t, v_{i,m}^t), \qquad m \in [M],$$

and we assume the observation model $C_t \mid (Y_t = m, \mu_{i,m}) \sim \mathcal{N}(\mu_{i,m}, 1)$. The Normal–Normal conjugate update yields

$$v_{i,y_t}^{t+1} = \left(\frac{1}{v_{i,y_t}^t} + 1\right)^{-1}, \qquad \beta_{i,y_t}^{t+1} = v_{i,y_t}^{t+1}\left(\frac{\beta_{i,y_t}^t}{v_{i,y_t}^t} + c_t\right),$$

while $(\beta_{i,m}^{t+1}, v_{i,m}^{t+1}) = (\beta_{i,m}^t, v_{i,m}^t)$ for all $m \neq y_t$, and $(\boldsymbol{\beta}_j^{t+1}, \boldsymbol{v}_j^{t+1}) = (\boldsymbol{\beta}_j^t, \boldsymbol{v}_j^t)$ for all $j \neq i$.

Given belief state $S^t$ and action $i_t = i$, the posterior distribution of the next label is

$$\Pr(Y_t = m \mid S^t, i_t = i) = \mathbb{E}[\theta_{i,m} \mid \boldsymbol{\alpha}_i^t] = \frac{\alpha_{i,m}^t}{\sum_{k=1}^M \alpha_{i,k}^t}.$$

Moreover, conditional on $Y_t = m$, the posterior predictive confidence is Gaussian:

$$C_t \mid (Y_t = m, S^t, i_t = i) \sim \mathcal{N}\big(\beta_{i,m}^t, \ 1 + v_{i,m}^t\big),$$

Together, the posterior predictive distribution and the conjugate update rules characterize the state transitions and transition probabilities of the resulting Bayesian MDP.

## B.2. Algorithm of PETS-Offline

We introduce the PETS-Offline algorithm here. Before doing that, we will give further explanation on $R^+(S_i^t)$, which is an optimistic approximation of $R(S_i^t)$, the expected one-step utility gain.

$$R(S_i^t) = \mathbb{E}_{Y_t, C_t}\big[U(S_i^{t+1}) - U(S_i^t) \mid S^t, \ i_t = i\big]$$

$$= \sum_{m=1}^M \mathbb{P}(Y_t = m \mid S^t, i_t = i) \, \mathbb{E}_{C_t}\big[U(S_i^{t+1}) - U(S_i^t) \mid S^t, \ i_t = i, \ Y_t = m\big]$$

Here the per-question terminal utility $U(S_i^t)$ under belief $S_i^t \triangleq (\boldsymbol{\alpha}_i^t, \boldsymbol{\beta}_i^t, \boldsymbol{v}_i^t)$ is defined as

$$U(S_i^t) \triangleq \max_{m \in [M]} \Pr\left(y_i^{W,\infty} = m \mid S_i^t\right),$$

In the optimistic version following (Chen et al., 2013), we use $R^+(S_i^t)$ to approximate $R(S_i^t)$ to avoid the complex computation of $\mathbb{P}(Y_t = m \mid S^t, i_t = i)$ and to achieve consistency of the algorithm as has been proven in (Chen et al., 2013).

In our setting where we have an additional parameters governing the posterior distribution of $C_t$, to relieve the computational burden, we make further simplification:

$$
\begin{aligned}
R_i^+(S^t) &\triangleq \max_{m \in [M]} \mathbb{E}\left[U(S_i^{t+1}) - U(S_i^t) \mid S^t, \, i_t = i, \, Y_t = m\right] \\
&= \max_{m \in [M]} \left(\mathbb{E}_{C_t \mid S^t, i_t = i, Y_t = m}\left[U(\mathrm{Upd}(S_i^t; m, C_t))\right] - U(S_i^t)\right) \\
&\approx \max_{m \in [M]} \left(U(\mathrm{Upd}(S_i^t; m, \beta_{i,m}^t)) - U(S_i^t)\right),
\end{aligned}
$$

Here $\mathrm{Upd}(S_i^t; m, c)$ denotes the posterior update obtained by hypothetically observing $(Y_t = m, C_t = c)$, and the approximation plugs in the predictive mean $\mathbb{E}[C_t \mid S^t, i_t = i, Y_t = m] = \beta_{i,m}^t$. This policy is myopic (and thus not globally optimal in general), but provides a principled and computationally efficient heuristic for adaptive budget allocation.

---

**Algorithm 2** Optimistic Knowledge Gradient (confidence-weighted Bayesian setting)

---

**Require:** Priors $\{(\boldsymbol{\alpha}_i^0, \boldsymbol{\beta}_i^0, \boldsymbol{v}_i^0)\}_{i=1}^N$, total budget $T$, plug-in parameter $\kappa \geq 0$
 1: **for** $t = 0, 1, \ldots, T - 1$ **do**
 2:     Compute per-question optimistic utility gain for all $i \in [N]$:

$$R_i^+(S^t) \approx \max_{m \in [M]} \left(U(\mathrm{Upd}(S_i^t; m, \beta_{i,m}^t)) - U(S_i^t)\right)$$

 3:     Select $i_t = \arg\max_{i \in [N]} R_i^+(S^t)$.
 4:     Query the LLM on question $i_t$ and observe $(y_t, c_t)$ with $y_t \in [M], c_t \in \mathbb{R}_+$.
 5:     Dirichlet update (label): $\boldsymbol{\alpha}_{i_t}^{t+1} = \boldsymbol{\alpha}_{i_t}^t + \boldsymbol{\delta}_{y_t}$.
 6:     Normal update (confidence mean for class $y_t$):

$$v_{i_t, y_t}^{t+1} = \left(\frac{1}{v_{i_t, y_t}^t} + 1\right)^{-1}, \quad \beta_{i_t, y_t}^{t+1} = v_{i_t, y_t}^{t+1}\left(\frac{\beta_{i_t, y_t}^t}{v_{i_t, y_t}^t} + c_t\right).$$

 7:     Keep all other parameters unchanged: for $j \neq i_t$, set $(\boldsymbol{\alpha}_j^{t+1}, \boldsymbol{\beta}_j^{t+1}, \boldsymbol{v}_j^{t+1}) = (\boldsymbol{\alpha}_j^t, \boldsymbol{\beta}_j^t, \boldsymbol{v}_j^t)$, and for $m \neq y_t$, set $(\beta_{i_t, m}^{t+1}, v_{i_t, m}^{t+1}) = (\beta_{i_t, m}^t, v_{i_t, m}^t)$.
 8: **end for**
 9: **return** Terminal labels $\hat{y}_i \in \arg\max_{m \in [M]} \Pr(y_i^{W,\infty} = m \mid S_i^T)$ for all $i \in [N]$.

---

In the special case $C_t \equiv 1$, the unweighted variant of our algorithm was proved to be consistent by Chen et al. (2013).

Note that a key component in this algorithm is to compute $\mathbb{P}(y_i^{W,\infty} = m \mid S_i^t)$, which can be achieved via Monte Carlo sampling.

### B.3. Proof of lemma 3.1

*Proof of Lemma 3.1.* In the offline trajectory allocation setting, after allocating a total of $H$ traces, we output a terminal label $\hat{y}_i$ for each question $i \in [N]$. Given the terminal posterior (belief) state $S^H = \{\boldsymbol{\alpha}_i^H\}_{i=1}^N$, our goal is to maximize the

conditional expected self-consistency rate:

$$\{\hat{y}_i\}_{i=1}^N \in \arg \max_{\{\hat{y}_i\}_{i=1}^N} \sum_{i=1}^N \mathbb{P}\left(\hat{y}_i = y_i^{W,\infty} \mid \mathcal{F}^H\right)$$

$$= \arg \max_{\{\hat{y}_i\}_{i=1}^N} \sum_{i=1}^N \sum_{m=1}^M \mathbb{I}(\hat{y}_i = m)\, \mathbb{P}\left(y_i^{W,\infty} = m \mid S_i^H\right), \tag{12}$$

where $\{\mathcal{F}^t\}_{t=0}^H$ denotes the filtration generated by the sample path $(i_0, y_{i_0}, c_{i_0} \ldots, i_{t-1}, y_{i_{t-1}}, c_{i_{t-1}})$, and the second equality follows because conditioning on $\mathcal{F}^H$ fixes the terminal posterior $S_i^H$ for each $i$.

Now observe that the objective in (12) is separable across questions. Indeed, for each fixed $i$, the decision variable $\hat{y}_i$ only appears in the $i$-th summand:

$$\sum_{m=1}^M \mathbb{I}(\hat{y}_i = m)\, \mathbb{P}\left(y_i^\infty = m \mid S_i^H\right).$$

Therefore, maximizing the total sum is equivalent to maximizing each term independently. For any question $i$, we have

$$\hat{y}_i \in \arg \max_{\hat{y} \in [M]} \sum_{m=1}^M \mathbb{I}(\hat{y} = m)\, \mathbb{P}\left(y_i^{W,\infty} = m \mid S_i^H\right)$$

$$= \arg \max_{m \in [M]} \mathbb{P}\left(y_i^{W,\infty} = m \mid S_i^H\right),$$

which proves the claim. $\qquad\square$

## C. Appendix of Online Case (Section 4)

### C.1. Details of Algorithm 1

**Streaming Policy.** Recall that the discretized formulation (9) yields a static grid-level allocation vector $(B_1, \ldots, B_K)$ under the average budget constraint $\sum_{j=1}^K \hat{p}_j B_j \leq B_{\text{total}}/N$. In the actual online streaming setting, however, the policy must remain feasible under a remaining-budget constraint that evolves with time. We therefore implement the online policy as a state-dependent dynamic re-planning rule.

At time $t$, define the remaining total budget

$$R_t \in \mathbb{Z}_{\geq 0}, \qquad R_1 := B_{\text{total}},$$

and the remaining horizon

$$H_t := N - t + 1.$$

Equivalently, the remaining per-question average budget is

$$\bar{B}_t := \frac{R_t}{H_t}.$$

Upon observing $q_t$, the policy computes a grid-level budget vector by solving (approximately) the discretized allocation problem under the current average budget $\bar{B}_t$:

$$(B_1^{(t)}, \ldots, B_K^{(t)}) \leftarrow \texttt{GreedyAlloc}\left(\{\hat{p}_j, \hat{\boldsymbol{\theta}}^j\}_{j=1}^K,\ \bar{B}_t\right), \tag{13}$$

where `GreedyAlloc` is Algorithm 1 (run with capacity $\bar{B}_t$) and returns an integer vector satisfying

$$\sum_{j=1}^K \hat{p}_j B_j^{(t)} \leq \bar{B}_t.$$

The actual budget spent on the current question is then $b_t := B_{j_t}^{(t)}$. After spending $b_t$ on question $q_t$, the remaining budget updates as

$$R_{t+1} := R_t - b_t.$$

By construction, the policy always respects the global budget constraint $\sum_{t=1}^{N} b_t \leq B_{\text{total}}$: indeed, $R_t$ is nonnegative and decreases by exactly the spent amount each step.

The rule (13) makes the policy dynamic: even though (9) is written with a fixed average constraint, the online execution continuously recomputes the target grid budgets using the remaining budget-to-horizon ratio $\bar{B}_t$. This re-planning step corrects for randomness and prevents early overspending, while retaining the same structure as in Algorithm 1.

**Random Rounding Method.** Our greedy allocation procedure increases the grid-level budgets in discrete steps. In particular, for grid $j$ one increment corresponds to adding two additional trials, which consumes expected budget $2\hat{p}_j$ (since a $\hat{p}_j$ fraction of incoming questions fall into grid $j$). As a result, near the end of the budget, the remaining expected budget may be insufficient to execute the next full 2-trial increment for the current best grid. Let $\delta$ denote the remaining expected budget (per-question average budget under the gridized constraint) at the moment when the algorithm is about to take one more 2-trial increment for the currently selected grid $i^\star$. Suppose that

$$\delta \in (0, 2\hat{p}_{i^\star}),$$

so a full increment for grid $i^\star$ would violate feasibility. Let $B = (B_1, \ldots, B_K)$ be the current integer allocation vector. Define $B^+$ as the neighboring allocation obtained by applying one additional greedy increment to grid $i^\star$:

$$B_{i^\star}^+ = B_{i^\star} + 2, \qquad B_j^+ = B_j \; (j \neq i^\star).$$

We then output a randomized allocation $\tilde{B}$ as follows:

- With probability $\rho := \delta/(2\hat{p}_{i^\star})$, set $\tilde{B} = B^+$.

- With probability $1 - \rho$, set $\tilde{B} = B$.

By construction,

$$\mathbb{E}\left[\sum_{j=1}^{K} \hat{p}_j \, \tilde{B}_j\right] = \sum_{j=1}^{K} \hat{p}_j \, B_j \; + \; \rho \cdot 2\hat{p}_{i^\star} = \sum_{j=1}^{K} \hat{p}_j \, B_j \; + \; \delta,$$

so the randomized rounding consumes exactly the remaining expected budget and keeps the expected budget constraint tight (hence feasible).

## C.2. Proof of Optimality of Algorithm 1 (Theorem 4.1)

We first establish a diminishing-returns property of $\text{SC}(\theta; B)$ when $\theta$ is a scalar, which directly motivates the optimality of our greedy allocation rule in Algorithm 1.

**Lemma C.1.** *For any $\theta \in [0, 1]$ and $n \in \mathbb{N}_+$, the marginal gain $R(\theta, n)$ is zero for odd $n$. Meanwhile, it is a nonincreasing function of $n$ over even integers, and strictly decreasing when $\theta \neq \frac{1}{2}$.*

*Proof of Lemma C.1.* We focus on the binary case where each call returns an answer in $\{0, 1\}$ and the ground-truth label is 1, since the function $\text{SC}(\theta; n)$ enjoys a symmetry around $\theta = \frac{1}{2}$:

$$\text{SC}(\theta; n) = \text{SC}(1 - \theta; n), \qquad \forall \theta \in [0, 1], \; B \in \mathbb{N}. \tag{14}$$

Indeed, flipping the answer of 0 and 1 for each call transforms the binomial count $X \sim \text{Bin}(n, \theta)$ into $B - X \sim \text{Bin}(B, 1 - \theta)$, while the majority rule with random tie-breaking is invariant under this relabeling. Consequently, by (14),

$$R(\theta, n) = R(1 - \theta, n), \qquad \forall \theta \in [0, 1], \; B \in \mathbb{N}.$$

Therefore, it suffices to analyze the case $\theta \geq \frac{1}{2}$ (replace $\theta$ by $1 - \theta$ if $\theta < \frac{1}{2}$).

Finally, the boundary case $\theta = \frac{1}{2}$ is trivial: when $\theta = \frac{1}{2}$, we have $X \sim \text{Bin}(n, \frac{1}{2})$ which is symmetric around $n/2$, hence

$$\text{SC}\left(\tfrac{1}{2}; n\right) = \mathbb{P}\left(X > \tfrac{n}{2}\right) + \tfrac{1}{2}\mathbb{P}\left(X = \tfrac{n}{2}\right) = \tfrac{1}{2}, \qquad \forall n,$$

and thus

$$R\left(\tfrac{1}{2}, n\right) \equiv 0.$$

which means, any extra budgets on questions with $\theta = \frac{1}{2}$ bring no increment. In the remainder of the proof, we may assume $\theta > \frac{1}{2}$.

If $n$ is odd, then set $n = 2m - 1$, then there are two cases which will change the majority vote result:

- When there are $m$ correct answers in $2m - 1$ samples, so the majority is correct. if the $2m$-th is wrong. then total becomes $m$ correct and $m$ wrong, which is a tie. With random tie breaking, correctness drops from 1 to 0.5. So the "drop amount" is 0.5.
  Probability of the borderline configuration: :

  $$\mathbb{P}[\text{Choose which m of the first } (2m - 1) \text{ are correct}] = \binom{2m - 1}{m}\theta^m(1 - \theta)^{m-1}.$$

  Then :
  $$\mathbb{P}[\text{the } 2m \text{ -th sample is wrong}] = 1 - \theta.$$

  while the $2m$-th is wrong. Then the decreasing probability is

  $$\frac{1}{2}\binom{2m - 1}{m}\theta^m(1 - \theta)^{m-1} \cdot (1 - \theta).$$

- When there are $m - 1$ correct answers in $2m - 1$ samples, and the $2m$-th is correct. Then similarly, increasing probability is

  $$\frac{1}{2}\binom{2m - 1}{m - 1}\theta^{m-1}(1 - \theta)^m \cdot \theta.$$

Thus $R(\theta, 2m - 1) = 0$ for odd $n = 2m - 1$. Similarly, for even $n$, there are also two cases which will increasing the correct probability, which is $m$ correct answers in $2m$ samples and the $2m + 1$-th is also correct or wrong, thus

$$R(\theta, 2m) = \frac{1}{2}\theta\binom{2m}{m}[\theta(1 - \theta)]^m - \frac{1}{2}(1 - \theta)\binom{2m}{m}[\theta(1 - \theta)]^m = \left(\theta - \frac{1}{2}\right)\binom{2m}{m}[\theta(1 - \theta)]^m.$$

Furthermore, since

$$\frac{R(\theta, 2m + 2)}{R(\theta, 2m)} = \frac{\binom{2m+2}{m+1}}{\binom{2m}{m}} \cdot \theta(1 - \theta) = \left(4 - \frac{2}{m + 1}\right)\theta(1 - \theta) < 1.$$

Then $R(\theta, 2m)$ is strictly decreasing for any $m$ when $\theta \neq \frac{1}{2}$. $\qquad \square$

*Proof of Theorem 4.1.* Recall the discretized (grided) optimization problem:

$$\max_{\{B_j \in \mathbb{Z}_{\geq 0}\}} \sum_{j=1}^{K} \hat{p}_j \text{SC}(\theta_j; B_j) \qquad \text{s.t.} \quad \sum_{j=1}^{K} \hat{p}_j B_j \leq B_{\text{total}}/N, \tag{15}$$

where $\hat{p}_j$ is the estimated probability mass of grid $j$ (i.e., the fraction of problems whose difficulty falls into grid $j$).

If we increase $B_j$ by one (i.e. add one extra trial to grid $j$), then the objective in (15) increases by

$$\hat{p}_j R(\theta_j, B_j),$$

and the expected-budget constraint increases by $p_j$. Therefore, the marginal gain per unit expected cost equals

$$\frac{\hat{p}_j R(\theta_j, B_j)}{\hat{p}_j} = R(\theta_j, B_j).$$

This shows that the grid probability $\hat{p}_j$ appears in both the objective and the expected-budget constraint, and cancels out when comparing actions by marginal gain per expected-cost unit. Hence, choosing the next increment by maximizing $R(\theta_j, B_j)$ is exactly greedy with respect to the correct marginal reward per expected-cost unit.

By Lemma C.1, for every $\theta > 1/2$ we have

$$R(\theta, 2m-1) = 0 \quad \text{and} \quad R(\theta, 0) > R(\theta, 2) > \cdots > R(\theta, 2m) > \cdots,$$

i.e., the nontrivial marginal gains occur only at even budgets and form a nonincreasing sequence. In particular, since $R(\theta, 2m-1) = 0$, we have

$$\text{SC}(\theta; 2m) = \text{SC}(\theta; 2m-1),$$

so allocating the $2m$-th trial does not improve the objective compared with $2m-1$. Equivalently, for any $m \geq 0$,

$$\text{SC}(\theta; 2m+1) - \text{SC}(\theta; 2m-1) = \big(\text{SC}(\theta; 2m+1) - \text{SC}(\theta; 2m)\big) = R(\theta, 2m).$$

Thus, when we think in effective increments, adding two trials to a grid (from $2m-1$ to $2m+1$) yields exactly the marginal reward $R(\theta, 2m)$, and these effective marginal rewards decrease with $m$.

Consider the multiset of all effective marginal rewards

$$\mathcal{H} := \big\{ R(\theta_j, 2m) : j \in [K], m = 0, 1, 2, \ldots \big\}.$$

Any feasible allocation $\{B_j\}$ in (15) corresponds to choosing a prefix of the sequence

$$R(\theta_j, 0), R(\theta_j, 2), R(\theta_j, 4), \ldots$$

because one cannot obtain the $(m+1)$-th effective gain for grid $i$ without also taking the first $m$ effective gains. Moreover, by Lemma C.1, each such sequence is nonincreasing.

Algorithm 1 maintains the active set

$$S = \{(j, B_j)\}_{i=1}^K$$

and repeatedly selects the grid with the largest currently available effective marginal reward, i.e., it chooses $j^\star \in \arg\max_j R(\theta_j, B_j)$ and then increases $B_{j^\star}$ by 2. Because each grid's effective marginal rewards form a nonincreasing sequence, this procedure is exactly the process which picks the globally largest remaining element from a collection of nonincreasing lists. Hence after $t$ effective steps, the greedy algorithm has selected the $t$ largest elements in $\mathcal{H}$ that are feasible under the prefix constraints, which maximizes the accumulated improvement in the objective among all allocations spending the same expected budget.

If the remaining budget is insufficient to complete the next 2-trial effective step for the current best grid, we can randomize the last step: suppose the remaining expected budget is $\delta \in (0, 2p_{j^\star})$. We perform the next 2 trials for grid $j^\star$ with probability $\delta/(2p_{j^\star})$ and do nothing otherwise. This keeps the expected budget exactly feasible and achieves the optimal convex combination of the two neighboring integer allocations. Therefore, Algorithm 1 attains the optimum of (15). $\qquad\square$

### C.3. 2-parameter Approximation in Multi-Choice Case

In the multi-choice setting, the difficulty parameter $\boldsymbol{\theta}$ is high-dimensional, making direct optimization costly. We therefore use a Gaussian surrogate family $\{\Phi(a\sqrt{n} + b)\}_{a>0}$ to approximate the self-consistency rate $\text{SC}(\boldsymbol{\theta}; n)$. This enables a direct extension of Algorithm 1: replace $\text{SC}(\boldsymbol{\theta}; n)$ with $\Phi(a\sqrt{n} + b)$ for a suitable pair $(a, b)$.

**Proposition C.2** (Gaussian-probit approximation with $1/\sqrt{n}$ rate). *Fix $M \geq 2$ and $\boldsymbol{\theta} \in \Delta^{M-1}$ with $\theta_1 > \theta_2 \geq \cdots \geq \theta_M$. Let $d := M - 1$. Define the margin vector $\Delta \in \mathbb{R}_+^d$ by*

$$\Delta_j := \theta_1 - \theta_{j+1} \qquad (j = 1, \ldots, d),$$

*and define the covariance matrix $\Sigma \in \mathbb{R}^{d \times d}$ of $V_1$ as in (17) below. Let*

$$\sigma_{\min} := \min_{1 \leq j \leq d} \sqrt{\Sigma_{jj}}, \qquad a := \min_{1 \leq j \leq d} \frac{\Delta_j}{\sqrt{\Sigma_{jj}}}.$$

*Then for all $n \geq 1$, the majority-vote success probability satisfies*

$$\left| \mathrm{SC}(\boldsymbol{\theta}; n) - \Phi(a\sqrt{n}) \right| \leq \frac{C(\boldsymbol{\theta}, M)}{\sqrt{n}},$$

*where one explicit admissible constant is*

$$C(\boldsymbol{\theta}, M) := C_{\mathrm{BE}} d^{1/4} \rho(\boldsymbol{\theta}) + \frac{d}{\sqrt{2\pi}\sigma_{\min}} + \frac{(d-1)\phi(a)}{a},$$

*with*

$$\rho(\boldsymbol{\theta}) := \mathbb{E}\left\| \Sigma^{-1/2}(V_1 - \Delta) \right\|_2^3, \qquad \phi(x) := \frac{1}{\sqrt{2\pi}} e^{-x^2/2}.$$

*and $C_{\mathrm{BE}}$ is the Berry-Esseen approximation constant(Raič, 2019). Moreover, since $\|V_1 - \Delta\|_2 \leq 2\sqrt{d}$ a.s., we have the fully explicit bound*

$$\rho(\boldsymbol{\theta}) \leq \frac{8d^{3/2}}{\lambda_{\min}(\Sigma)^{3/2}},$$

*hence*

$$C(\boldsymbol{\theta}, M) \leq \frac{8C_{\mathrm{BE}} d^{7/4}}{\lambda_{\min}(\Sigma)^{3/2}} + \frac{d}{\sqrt{2\pi}\sigma_{\min}} + \frac{(d-1)\phi(a)}{a}.$$

*Proof of Proposition C.2.* Let $A \in \{1, \ldots, M\}$ with $\mathbb{P}(A = i) = \theta_i$. After $n$ i.i.d. draws, let

$$X = (X_1, \ldots, X_M) \sim \mathrm{Multinomial}(n, \boldsymbol{\theta}), \qquad \mathrm{SC}(\boldsymbol{\theta}; n) = p_n := \mathbb{P}(\arg\max_i X_i = 1),$$

with uniform random tie-breaking.

Define the margin vector

$$D := (D_2, \ldots, D_M) \in \mathbb{Z}^d, \qquad D_j := X_1 - X_j \ (j = 2, \ldots, M).$$

Let $\mathcal{E}_n := \{\arg\max_i X_i = 1\}$. Then deterministically

$$\{D \in [1, \infty)^d\} \subseteq \mathcal{E}_n \subseteq \{D \in \mathbb{R}_+^d\}.$$

Hence

$$\mathbb{P}(D \in [1, \infty)^d) \leq p_n \leq \mathbb{P}(D \in \mathbb{R}_+^d). \tag{16}$$

Let $A_1, \ldots, A_n$ be the i.i.d. samples. Define for each round $b$ the vector

$$V_b = (V_{b,2}, \ldots, V_{b,M}) \in \{-1, 0, 1\}^d, \qquad V_{b,j} := \mathbb{I}(A_b = 1) - \mathbb{I}(A_b = j).$$

Then

$$D = \sum_{b=1}^{n} V_b, \qquad \mathbb{E}[V_b] = \Delta := (\theta_1 - \theta_2, \ldots, \theta_1 - \theta_M) \in \mathbb{R}_+^d.$$

A direct computation gives the covariance matrix $\Sigma = \mathbb{Cov}(V_1) \in \mathbb{R}^{d \times d}$:

$$\Sigma_{jj} = \theta_1 + \theta_{j+1} - (\theta_1 - \theta_{j+1})^2, \qquad \Sigma_{jk} = \theta_1 - (\theta_1 - \theta_{j+1})(\theta_1 - \theta_{k+1}) \quad (j \neq k). \tag{17}$$

Note that $\Delta_j \leq \theta_1 \leq 1$ implies $\Sigma_{jk} \geq \theta_1 - \theta_1^2 = \theta_1(1 - \theta_1) \geq 0$, so $\Sigma$ has nonnegative off-diagonal entries. Since $\Sigma = \mathbb{Cov}(V_1)$, it is always positive semidefinite. To prove $\Sigma \succ 0$, it suffices to show that for any $u \in \mathbb{R}^d$,

$$u^\top \Sigma u = \mathbb{Var}(u^\top V_1) = 0 \implies u = 0.$$

Note that $V_1$ takes only the following values:

$$V_1 = \begin{cases} \mathbf{1}_d, & A = 1, \\ -e_j, & A = j + 1, \quad j = 1, \ldots, d, \end{cases}$$

where $\mathbf{1}_d$ is the all-ones vector in $\mathbb{R}^d$ and $e_j$ is the $j$-th standard basis vector. Hence

$$u^\top V_1 = \begin{cases} \sum_{i=1}^d u_i, & A = 1, \\ -u_j, & A = j + 1, \quad j = 1, \ldots, d. \end{cases}$$

If $\mathbb{V}\mathrm{ar}(u^\top V_1) = 0$, then $u^\top V_1$ must be almost surely constant. Since $\theta_1 > 0$ and $\theta_{j+1} > 0$ for all $j$, all events $\{A = 1\}$ and $\{A = j + 1\}$ occur with positive probability, so we must have

$$\sum_{i=1}^d u_i = -u_j, \qquad \forall j = 1, \ldots, d.$$

In particular, the right-hand side does not depend on $j$, implying $u_1 = \cdots = u_d =: c$. Substituting into the above identities yields $dc = -c$, i.e., $(d+1)c = 0$, hence $c = 0$ and therefore $u = 0$.

Thus $\mathbb{V}\mathrm{ar}(u^\top V_1) = 0$ implies $u = 0$, which proves $\Sigma \succ 0$.

Let $G \sim \mathcal{N}(n\Delta, n\Sigma)$. By Bentkus' convex-set Berry–Esseen bound(Raič, 2019), for any convex set $A \subset \mathbb{R}^d$,

$$\left| \mathbb{P}(D \in A) - \mathbb{P}(G \in A) \right| \leq \frac{C_{\mathrm{BE}} d^{1/4}}{\sqrt{n}} \mathbb{E} \left\| \Sigma^{-1/2}(V_1 - \Delta) \right\|_2^3.$$

Both $A_0 := \mathbb{R}_+^d$ and $A_1 := [1, \infty)^d$ are convex, hence

$$\left| \mathbb{P}(D \in A_k) - \mathbb{P}(G \in A_k) \right| \leq \frac{C_{\mathrm{BE}} d^{1/4}}{\sqrt{n}} \rho(\boldsymbol{\theta}), \qquad k \in \{0, 1\}. \tag{18}$$

Write $G = n\Delta + \sqrt{n}\, Z$ with $Z \sim \mathcal{N}(0, \Sigma)$. Then

$$0 \leq \mathbb{P}(G \in A_0) - \mathbb{P}(G \in A_1) \leq \sum_{j=1}^d \mathbb{P}(0 \leq G_j \leq 1) = \sum_{j=1}^d \mathbb{P}\left( -\sqrt{n}\,\Delta_j \leq Z_j \leq \frac{1}{\sqrt{n}} - \sqrt{n}\,\Delta_j \right).$$

Since $Z_j \sim \mathcal{N}(0, \Sigma_{jj})$ has density bounded by $1/\sqrt{2\pi\Sigma_{jj}}$,

$$\mathbb{P}\left( -\sqrt{n}\,\Delta_j \leq Z_j \leq \frac{1}{\sqrt{n}} - \sqrt{n}\,\Delta_j \right) \leq \frac{1}{\sqrt{n}} \cdot \frac{1}{\sqrt{2\pi\Sigma_{jj}}} \leq \frac{1}{\sqrt{n}} \cdot \frac{1}{\sqrt{2\pi}\,\sigma_{\min}}.$$

Therefore,

$$0 \leq \mathbb{P}(G \in A_0) - \mathbb{P}(G \in A_1) \leq \frac{d}{\sqrt{n}} \cdot \frac{1}{\sqrt{2\pi}\,\sigma_{\min}}. \tag{19}$$

Let $Z' = (Z_1', \ldots, Z_d')$ be the coordinate-wise standardized version of $Z$:

$$Z_j' := -\frac{Z_j}{\sqrt{\Sigma_{jj}}}, \qquad c_j := \frac{\Delta_j}{\sqrt{\Sigma_{jj}}} > 0, \qquad a := \min_{1 \leq j \leq d} c_j.$$

Then

$$\mathbb{P}(G \in A_0) = \mathbb{P}(Z \geq -\sqrt{n}\,\Delta) = \mathbb{P}(Z' \leq \sqrt{n}\, c) =: g(\sqrt{n}).$$

Let $j^\star \in \arg\min_j c_j$ so that $c_{j^\star} = a$. Since the event $\{Z' \leq \sqrt{n}\, c\}$ implies $Z_{j^\star}' \leq a\sqrt{n}$, we have the upper bound

$$g(\sqrt{n}) \leq \mathbb{P}(Z_{j^\star}' \leq a\sqrt{n}) = \Phi(a\sqrt{n}).$$

For the lower bound, by Gaussian association (nonnegative correlations) the decreasing events (Esary et al., 1967). $\{Z_j' \leq a\sqrt{n}\}$ satisfy

$$\mathbb{P}\left( \bigcap_{j=1}^d \{Z_j' \leq a\sqrt{n}\} \right) \geq \prod_{j=1}^d \mathbb{P}(Z_j' \leq a\sqrt{n}) = \Phi(a\sqrt{n})^d.$$

Moreover $\bigcap_j \{Z'_j \leq a\sqrt{n}\} \subseteq \{Z' \leq \sqrt{n}\,c\}$, hence

$$g(\sqrt{n}) \geq \Phi(a\sqrt{n})^d.$$

Therefore,

$$0 \leq \Phi(a\sqrt{n}) - g(\sqrt{n}) \leq \Phi(a\sqrt{n}) - \Phi(a\sqrt{n})^d. \tag{20}$$

Let $p := \Phi(a\sqrt{n}) \in (1/2, 1)$ and note that for $x \in [0,1]$, $1 - x^{d-1} \leq (d-1)(1-x)$ (Bernoulli inequality). Then

$$p - p^d = p\left(1 - p^{d-1}\right) \leq (d-1)p(1-p) \leq (d-1)(1-p).$$

Using Mills' ratio $1 - \Phi(x) \leq \phi(x)/x$ for $x > 0$, we obtain

$$0 \leq \Phi(a\sqrt{n}) - g(\sqrt{n}) \leq (d-1)\left(1 - \Phi(a\sqrt{n})\right) \leq (d-1)\frac{\phi(a\sqrt{n})}{a\sqrt{n}} \leq \frac{(d-1)\phi(a)}{a\sqrt{n}},$$

where the last step uses $\phi(a\sqrt{n}) \leq \phi(a)$ for $n \geq 1$.

From (16), (18), and (19),

$$\left|p_n - \mathbb{P}(G \in A_0)\right| \leq \frac{C_{\mathrm{BE}}d^{1/4}}{\sqrt{n}}\rho(\boldsymbol{\theta}) + \frac{d}{\sqrt{n}} \cdot \frac{1}{\sqrt{2\pi}\sigma_{\min}}.$$

Adding Step 5 yields

$$\left|p_n - \Phi(a\sqrt{n})\right| \leq \frac{C_{\mathrm{BE}}d^{1/4}}{\sqrt{n}}\rho(\boldsymbol{\theta}) + \frac{d}{\sqrt{n}} \cdot \frac{1}{\sqrt{2\pi}\sigma_{\min}} + \frac{(d-1)\phi(a)}{a\sqrt{n}}.$$

This proves the claim with $b = 0$ and the stated explicit constant $C(\boldsymbol{\theta}, M)$. $\qquad\square$

*Remark* C.3. Indeed, in practice we use the regression method to decide parameter $a, b$ for given $\boldsymbol{\theta}$, such approximation has very great precision, see Figure 5 as a reference.

Recall the Gaussian surrogate family

$$g_{a,b}(k) := \Phi\left(a\sqrt{k} + b\right), \qquad a > 0, \ k \geq 0,$$

where $\Phi$ is the standard normal CDF and $\phi := \Phi'$ is the standard normal PDF.

**Concavity on the relevant budget range.** In our allocation procedure, each question (or difficulty grid) receives at least a small warm-up budget before the greedy stage. Let $k_{\min} \geq 1$ denote the smallest budget value that can occur in the greedy stage (e.g., $k_{\min} = 4$ if we warm up with 4 samples). The next lemma shows $g_{a,b}$ is concave for all $k \geq k_{\min}$ as long as $a\sqrt{k} + b$ is nonnegative on that range.

**Lemma C.4** (Concavity of $g_{a,b}$ for $k \geq k_{\min}$). *Fix $a > 0$ and $b \in \mathbb{R}$. If*

$$a\sqrt{k_{\min}} + b \geq 0, \tag{21}$$

*then the function $k \mapsto g_{a,b}(k)$ is concave on $[k_{\min}, \infty)$ (in the usual continuous sense). In particular, $g'_{a,b}(k)$ is nonincreasing on $[k_{\min}, \infty)$.*

*Proof.* For $k > 0$, let $t(k) := a\sqrt{k} + b$. By the chain rule,

$$g'_{a,b}(k) = \phi(t(k)) \cdot \frac{a}{2\sqrt{k}}.$$

Differentiating again (using $\phi'(x) = -x\phi(x)$) yields

$$\begin{aligned}
g''_{a,b}(k) &= \frac{d}{dk}\left(\phi(t(k)) \cdot \frac{a}{2\sqrt{k}}\right) \\
&= \phi'(t(k)) \cdot t'(k) \cdot \frac{a}{2\sqrt{k}} + \phi(t(k)) \cdot \frac{d}{dk}\left(\frac{a}{2\sqrt{k}}\right) \\
&= \left(-t(k)\phi(t(k))\right) \cdot \frac{a}{2\sqrt{k}} \cdot \frac{a}{2\sqrt{k}} + \phi(t(k)) \cdot \left(-\frac{a}{4k^{3/2}}\right) \\
&= -\phi(t(k))\left(\frac{a^2\, t(k)}{4k} + \frac{a}{4k^{3/2}}\right).
\end{aligned}$$

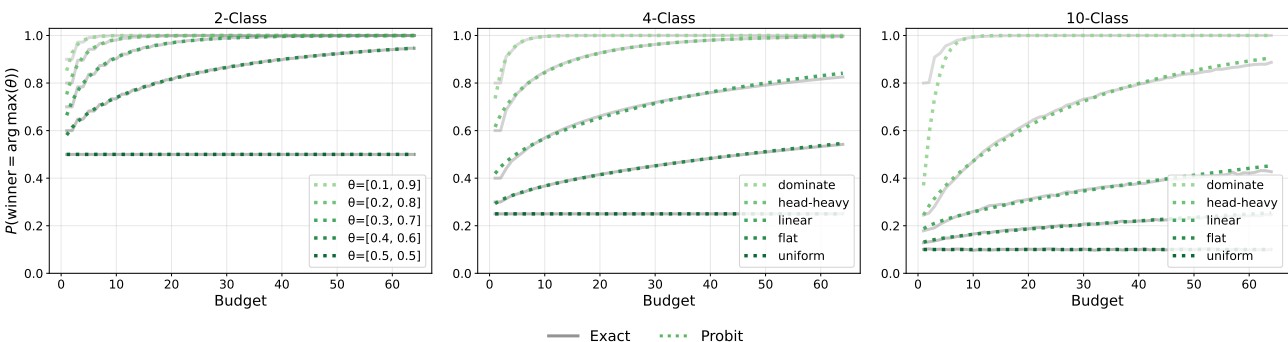

*Figure 5.* Probability that multinomial majority voting selects the true best option as a function of budget. We plot $\mathbb{P}\big(Y^{\mathrm{Maj}}(B) = \arg\max_{y \in \mathcal{Y}} \theta_y\big)$ versus BUDGET $\in \{1, \ldots, 64\}$ for different ground-truth preference vectors $\boldsymbol{\theta} = (\theta_1, \ldots, \theta_M)$. Gray curves (*Exact*) are computed from the true $\boldsymbol{\theta}$ (exact for $M = 2, 4$; Monte Carlo estimates for $M = 10$), while green dotted curves (*Probit*) are produced by fitting a two-parameter probit model and evaluating the fitted model across budgets. The fitted probit curves closely track the exact/MC curves in all regimes. Panels correspond to $M \in \{2, 4, 10\}$. For $M = 4$, we use: DOMINATE $[0.8, 0.1, 0.1, 0]$, HEAD-HEAVY $[0.6, 0.3, 0.05, 0.05]$, LINEAR $[0.4, 0.3, 0.2, 0.1]$, FLAT $[0.3, 0.25, 0.25, 0.2]$, and UNIFORM $[0.25, 0.25, 0.25, 0.25]$. For $M = 10$, we use: DOMINATE $[0.8, 0.15, 0.05, 0, \ldots, 0]$, HEAD-HEAVY $[0.25, 0.15, 0.10, 0.0714, \ldots, 0.0714]$, LINEAR $[10, 9, \ldots, 1]/55$, FLAT normalize($[10, 9, \ldots, 1]^{0.4}$), and UNIFORM $[0.1, \ldots, 0.1]$.

Now assume (21). Then for all $k \geq k_{\min}$ we have $t(k) = a\sqrt{k} + b \geq 0$. Since $\phi(t(k)) > 0$ and the bracketed term is nonnegative, we conclude $g''_{a,b}(k) \leq 0$ for all $k \geq k_{\min}$, i.e., $g_{a,b}$ is concave on $[k_{\min}, \infty)$. $\square$

In our multi-choice extension, we approximate $\mathrm{SC}(\boldsymbol{\theta}_i; k)$ by

$$\mathrm{SC}(\boldsymbol{\theta}; k) \approx= \Phi(a_i\sqrt{k} + b_i),$$

which has diminishing returns on the greedy range. Therefore, replacing $\mathrm{SC}(\boldsymbol{\theta}_i; k)$ by this Gaussian family preserves the key structural property needed by Algorithm 1, and the same greedy allocation rule applies verbatim in the multi-choice case.

## C.4. Warm-up griding and Difficulty Estimation Details

In practice, we use a short warm-up to map each question to a coarse difficulty grid, and then apply a precomputed one-shot budget for that grid. For each question $q$, we first draw 4 responses and let $C_q^{(4)} = (c_1, c_2, c_3, c_4)$ be the sorted option counts. Up to label permutation, $C_q^{(4)}$ takes one of five patterns, inducing five grids $T_q \in \{1, \ldots, 5\}$ via a deterministic rule $T_q = g(C_q^{(4)})$. Using **training** questions with large i.i.d. answer pools, we estimate the grid proportions $\{\hat{p}_j\}_{j=1}^5$ and fit a representative surrogate curve $(\hat{a}_j, \hat{b}_j)$ for each grid. At **test** time, we assign $q$ to $T_q$ using the 4 warm-up samples and then allocate $B_{T_q}$ additional samples in a single shot. We provide the estimation details below.

Given a question $q$, we draw 4 i.i.d. LLM responses and count how many times each answer option is selected. Let

$$C_q^{(4)} = (c_1, c_2, c_3, c_4)$$

denote the sorted (descending) counts of answer options among the first four responses, padding with zeros if fewer than four distinct options appear. Note that this definition is valid regardless of the total number of answer options: with only four samples, at most four options can appear, so the count pattern space is unchanged for multiple-choice with $\geq 4$ options and also extends to other discrete-answer settings.

By symmetry among answer labels, there are exactly five possible patterns:

$$\Big\{(4, 0, 0, 0), (3, 1, 0, 0), (2, 2, 0, 0), (2, 1, 1, 0), (1, 1, 1, 1)\Big\}.$$

We define a deterministic grid mapping

$$T_q = g(C_q^{(4)}) \in \{1, \ldots, 5\},$$

e.g., $g$ can map the patterns above to grids in the listed order (any fixed one-to-one mapping works as long as it is used consistently).

For each training question $q$, we assume access to a large pool of i.i.d. answers generated from

$$y_q \sim \mathrm{Cat}(\boldsymbol{\theta}_q).$$

To account for the randomness of using only four warm-up samples, we repeatedly subsample four answers (without replacement from the pool, or with replacement if the pool is large) and record the induced grid:

$$T_q^{(r)} = g\left(C_{q,r}^{(4)}\right), \qquad r = 1, \dots, R.$$

This yields an empirical, question-specific grid distribution

$$\hat{p}_q(j) = \frac{1}{R} \sum_{r=1}^{R} \mathbf{1}\{T_q^{(r)} = j\}, \qquad j \in \{1, \dots, 5\}.$$

Averaging over training questions produces the empirical grid proportions

$$\hat{p}_j = \frac{1}{|\mathcal{Q}_{\mathrm{train}}|} \sum_{q \in \mathcal{Q}_{\mathrm{train}}} \hat{p}_q(j), \qquad j \in \{1, \dots, 5\}.$$

For each training question $q$, we fit its two-parameter difficulty curve $(\hat{a}_q, \hat{b}_q)$ using the procedure in Section C.3 based on the full answer pool (so that the fit is stable).

To obtain a representative curve for each grid, we aggregate the per-question fits within that grid. To reduce sensitivity to borderline cases, we recommend using the soft grid weights $\hat{p}_q(j)$:

$$(\hat{a}_j, \hat{b}_j) = \arg \min_{(a,b)} \sum_{q \in \mathcal{Q}_{\mathrm{train}}} \hat{p}_q(j)\, \ell_q(a, b),$$

where $\ell_q(a, b) = \mathbb{E}_n \left| \mathrm{SC}(\boldsymbol{\theta}(q); n) - \Phi(a\sqrt{n} + b) \right|$ is the fitting loss in Section C.3. A simpler alternative is a weighted average in parameter space,

$$\hat{a}_j = \frac{\sum_q \hat{p}_q(j)\hat{a}_q}{\sum_q \hat{p}_q(j)}, \qquad \hat{b}_j = \frac{\sum_q \hat{p}_q(j)\hat{b}_q}{\sum_q \hat{p}_q(j)},$$

which we found to work well when the per-question fits are already accurate.

## D. An Asymptotic Perspective on Online vs. Offline PETS

In this section, we prove that as the total budget $B \to \infty$, the budget allocated to each question type converges to a fixed proportion for both the offline (Section 3) and the online (Section 4) settings. Moreover, the limiting proportions in the offline and online cases are very close, which highlights the consistency between the two methods.

### D.1. Convergence of Budget Proportions for the Online Case

We first show that as $B \to \infty$, every problem type will be sampled infinitely many times.

**Lemma D.1.** *For a binary-choice problem set $\mathcal{P}$ with difficulty labels $\{\theta_i\}_{i \in [N]}$ and corresponding probabilities $\{p_i\}_{i \in [N]}$ (where $p_i > 0$), suppose we allocate budgets using Algorithm 1. Then, as the total budget $B \to \infty$, the sample count $B_i(B)$ for every $i$ diverges to infinity.*

*Proof of Lemma D.1.* Since difficulty labels $\theta$ and $1-\theta$ lead to the same behavior in our algorithm, without loss of generality we assume $\theta_i \in (1/2, 1)$. For simplicity, assume each problem type has been sampled once initially. Then, following Algorithm 1, the marginal accuracy gain from allocating two additional budget units at level $2m - 1$ is

$$R(2m-1; \theta_i) = p_i \left( \theta_i - \frac{1}{2} \right) \binom{2m}{m} [\theta_i(1 - \theta_i)]^m, \qquad m \in \mathbb{N}, \tag{22}$$

where $\lambda_i$ is the probability mass of the problem type with label $\theta_i$. At round $t$, the algorithm selects

$$i_t \in \arg\max_i R(2m_i(t) - 1; \theta_i), \qquad m_{i_t}(t + 1) = m_{i_t}(t) + 1, \quad m_j(t + 1) = m_j(t) \ (j \neq i_t),$$

with the initialization $m_i(0) = 1$. Define $\lambda_i := \theta_i(1 - \theta_i) \in (0, 1/4)$. Then

$$\frac{R(2m + 1; \theta_i)}{R(2m - 1; \theta_i)} = \frac{\binom{2m+2}{m+1}}{\binom{2m}{m}} \cdot \lambda_i = \left(4 - \frac{2}{m+1}\right)\lambda_i < 4\lambda_i < 1.$$

Therefore, as $m$ increases, the marginal gain for each arm strictly decreases.

Now suppose there exists some type $j$ that is sampled only finitely many times as $B \to \infty$. Then there exists a constant $M$ such that $m_j(t) \equiv M$ for all sufficiently large $t$, and thus $R(m_j(t); \theta_j) \equiv R(M; \theta_j) > 0$ remains constant. On the other hand, at least one arm $i$ must be selected infinitely often; for that arm, we have $m_i(t) \to \infty$ and hence $R(m_i(t); \theta_i) \to 0$, which contradicts the greedy choice rule (since eventually $R(M; \theta_j)$ would dominate). Therefore, every arm must be sampled infinitely often, i.e., $m_i(t) \to \infty$ for all $i$. $\qquad \square$

Next, we derive the convergence of sampling proportions and identify the limiting ratios.

**Proposition D.2.** *Under the same conditions as Lemma D.1, the ratio $m_i(B)/B$ converges to a constant for each $i$ as $B \to \infty$. Moreover,*

$$\frac{m_i(B)}{B} \propto \frac{1}{p(\theta_i)}.$$

*where $p(\theta) = -\log(4\theta(1 - \theta)) > 0$.*

*Proof of Proposition D.6.* Define, for each difficulty label,

$$\Phi_i(2m - 1) := -\log R(2m - 1; \theta_i). \tag{23}$$

For each update of arm $i$, the corresponding increment in $\Phi_i$ is

$$\Delta_i(2m - 1) = \Phi_i(2m + 1) - \Phi_i(2m - 1) = -\log\left(\left(4 - \frac{2}{m+1}\right)p_i\right).$$

Since for $m \geq 1$ we have $4 - \frac{2}{m+1} \in [3, 4)$, it follows that for each fixed $i$,

$$0 < -\log(4p_i) \leq \Delta_i(m) \leq -\log(3p_i) < \infty.$$

Let

$$\Delta_{\max} := \max_i \sup_{m \geq 1} \Delta_i(m) = \max_i\left(-\log(3p_i)\right) < \infty.$$

Define the minimum and maximum potentials at time $t$ as

$$L(t) := \min_i \Phi_i\big(m_i(t)\big), \qquad U(t) := \max_i \Phi_i\big(m_i(t)\big).$$

At time $t$, the algorithm samples the arm attaining the minimum potential, say $i_t$, and increases its potential from $L(t)$ to at most $L(t) + \Delta_{\max}$, while leaving all other arms unchanged. Hence,

$$U(t + 1) \leq \max\{U(t), \ L(t) + \Delta_{\max}\}.$$

Consequently,

$$U(t + 1) - L(t + 1) \leq \max\{U(0) - L(0), \ \Delta_{\max}\}.$$

By induction, letting $C := \max\{U(0) - L(0), \Delta_{\max}\}$, we obtain

$$\big|\Phi_i\big(m_i(t)\big) - \Phi_j\big(m_j(t)\big)\big| \leq C, \qquad \forall i, j. \tag{24}$$

Next, we derive a first-order approximation for $\Phi_i(m)$. By Stirling's formula,

$$R(m; \theta_i) = \lambda_i \left(\theta_i - \frac{1}{2}\right) \binom{2m}{m} p_i^m \tag{25}$$

$$= \lambda_i \left(\theta_i - \frac{1}{2}\right) \frac{4^m}{\sqrt{\pi m}} \left(1 + O\left(\frac{1}{m}\right)\right) p_i^m. \tag{26}$$

Therefore,

$$\Phi_i(m) = -\log R(m; \theta_i) = -m \log(4p_i) + \frac{1}{2} \log m + b_i + o(1), \tag{27}$$

where $b_i := -\log\left(\frac{\lambda_i\left(\theta_i - \frac{1}{2}\right)}{\sqrt{\pi}}\right)$ is a constant, and $o(1) \to 0$ as $m \to \infty$.

Combining (24) and (27), we obtain

$$-m_i(t) \log(4p_i) + \frac{1}{2} \log m_i(t) + b_i = -m_j(t) \log(4p_j) + \frac{1}{2} \log m_j(t) + b_j + O(1).$$

Moving the logarithmic terms to the right-hand side yields

$$m_i(t) \log(4p_i) = m_j(t) \log(4p_j) + O(\log t),$$

since $m_j(t) \leq t$ and all arms are sampled infinitely often asymptotically. Fixing $m_1(t)$, this implies

$$m_i(t) = \frac{\log(4p_1)}{\log(4p_i)} m_1(t) + o(t).$$

Hence,

$$t + N = \sum_i m_i(t) = m_1(t) \sum_i \frac{\log(4p_1)}{\log(4p_i)} + o(t).$$

Finally,

$$\frac{m_i(t)}{m_j(t)} = \frac{\log(4p_j)}{\log(4p_i)} + o(1) \quad \Longrightarrow \quad \lim_{t \to \infty} \frac{m_i(t)}{m_j(t)} = \frac{\log(4p_j)}{\log(4p_i)} = \frac{\log\left(4\theta_j(1 - \theta_j)\right)}{\log\left(4\theta_i(1 - \theta_i)\right)}. \tag{28}$$

$\square$

### D.2. Convergence of Budget Proportions for Offline Case

Let $X \sim \text{Beta}(a, b)$ with parameters $a, b \in \mathbb{N}_+$. Its density is

$$f_{a,b}(x) = \frac{1}{B(a, b)} x^{a-1}(1 - x)^{b-1}, \qquad x \in (0, 1),$$

where the Beta function is

$$B(a, b) = \int_0^1 t^{a-1}(1 - t)^{b-1} \, dt = \frac{\Gamma(a)\Gamma(b)}{\Gamma(a + b)} = \frac{(a - 1)!(b - 1)!}{(a + b - 1)!}.$$

We define the tail probability at $1/2$:

$$P_{a,b} := \mathbb{P}(X \geq 1/2).$$

We first obtain a closed form for $\mathbb{P}(X \geq x)$ when $a, b$ are integers:

**Lemma D.3.** *For any integer $a, b \geq 1$ and any $x \in [0, 1]$,*

$$\mathbb{P}(X \geq x) = \sum_{k=0}^{a-1} \binom{a + b - 1}{k} x^k (1 - x)^{a+b-1-k}.$$

*Equivalently,*

$$\mathbb{P}(X \leq x) = \sum_{k=a}^{a+b-1} \binom{a + b - 1}{k} x^k (1 - x)^{a+b-1-k}.$$

*Proof.* Define the polynomial

$$Q(x) := \sum_{k=0}^{a-1} \binom{a+b-1}{k} x^k (1-x)^{a+b-1-k}, \qquad x \in [0, 1].$$

We will show that $Q(x) = \mathbb{P}(X \geq x)$ by verifying that $Q$ has the same derivative as the Beta tail probability and the same boundary value at $x = 1$.

**(i) Differentiate $Q(x)$.** For each term $x^k (1-x)^{a+b-1-k}$ we have

$$\frac{d}{dx}\left[x^k(1-x)^{a+b-1-k}\right] = kx^{k-1}(1-x)^{a+b-1-k} - (a+b-1-k)x^k(1-x)^{a+b-2-k}.$$

Hence

$$Q'(x) = \sum_{k=0}^{a-1} \binom{a+b-1}{k}\left[kx^{k-1}(1-x)^{a+b-1-k} - (a+b-1-k)x^k(1-x)^{a+b-2-k}\right].$$

The $k = 0$ term in the first part is zero, so re-index the first sum by $j = k - 1$:

$$\sum_{k=1}^{a-1} \binom{a+b-1}{k} k\, x^{k-1}(1-x)^{a+b-1-k} = \sum_{j=0}^{a-2} \binom{a+b-1}{j+1}(j+1)\, x^j(1-x)^{a+b-2-j}.$$

Also rewrite the second part as

$$\sum_{k=0}^{a-1} \binom{a+b-1}{k}(a+b-1-k)x^k(1-x)^{a+b-2-k}.$$

Therefore

$$Q'(x) = \sum_{j=0}^{a-2} \binom{a+b-1}{j+1}(j+1)x^j(1-x)^{a+b-2-j}$$

$$- \sum_{k=0}^{a-1} \binom{a+b-1}{k}(a+b-1-k)x^k(1-x)^{a+b-2-k}$$

$$= (a+b-1)\sum_{j=0}^{a-2} \binom{a+b-2}{j}x^j(1-x)^{a+b-2-j} - (a+b-1)\sum_{k=0}^{a-1} \binom{a+b-2}{k}x^k(1-x)^{a+b-2-k}.$$

These two sums cancel term-by-term for $k = 0, 1, \ldots, a - 2$, leaving only the $k = a - 1$ term from the second sum:

$$Q'(x) = -(a+b-1)\binom{a+b-2}{a-1}x^{a-1}(1-x)^{b-1} = -\frac{(a+b-1)!}{(a-1)!(b-1)!}x^{a-1}(1-x)^{b-1}.$$

**(ii) Differentiate the Beta tail probability.** Define

$$T(x) := \mathbb{P}(X \geq x) = \int_x^1 \frac{1}{B(a,b)}t^{a-1}(1-t)^{b-1}\, dt.$$

By the fundamental theorem of calculus,

$$T'(x) = -\frac{1}{B(a,b)}x^{a-1}(1-x)^{b-1} = -\frac{(a+b-1)!}{(a-1)!(b-1)!}x^{a-1}(1-x)^{b-1},$$

which matches $Q'(x)$.

We have

$$T(1) = \mathbb{P}(X \geq 1) = 0.$$

Also,

$$Q(1) = \sum_{k=0}^{a-1} \binom{a+b-1}{k} 1^k (1-1)^{a+b-1-k} = 0,$$

Since $Q'(x) = T'(x)$ on $[0, 1]$ and $Q(1) = T(1)$, we conclude $Q(x) = T(x)$ for all $x \in [0, 1]$, i.e.

$$\mathbb{P}(X \geq x) = Q(x) = \sum_{k=0}^{a-1} \binom{a+b-1}{k} x^k (1-x)^{a+b-1-k}.$$

This proves the lemma. $\qquad\qquad\qquad\qquad\qquad\qquad\qquad\qquad\qquad\qquad\qquad\qquad\qquad\quad\square$

Then, we can derive the explicit increment of self-consistency of a problem from state $(m, n)$ to $(m + 1, n)$(that is, add one positive answer based on $m$ positive and $n$ negative answer) if the true answer is positive.

**Lemma D.4.** *We have*

$$\Delta_{m,n} := P_{m+1,n} - P_{m,n} = \frac{1}{2^{m+n}} \binom{m+n-1}{n-1}.$$

*where*

$$P_{m,n} = \mathbb{P}(X \geq 1/2), X \sim \text{Beta}(m, n).$$

*Proof.* Apply Lemma D.3 with $x = \frac{1}{2}$:

$$P_{a,b} = \sum_{k=0}^{a-1} \binom{a+b-1}{k} \left(\frac{1}{2}\right)^k \left(\frac{1}{2}\right)^{a+b-1-k} = 2^{-(a+b-1)} \sum_{k=0}^{a-1} \binom{a+b-1}{k}.$$

Let $m, n \in \{1, \ldots, N\}$

$$\Delta_{m,n} := P_{m+1,n} - P_{m,n} = \mathbb{P}(\text{Beta}(m + 1, n) \geq 1/2) - \mathbb{P}(\text{Beta}(m, n) \geq 1/2).$$

We write

$$P_{m+1,n} = 2^{-(m+n)} \sum_{k=0}^{m} \binom{m+n}{k}, \qquad P_{m,n} = 2^{-(m+n-1)} \sum_{k=0}^{m-1} \binom{m+n-1}{k}.$$

Let $A := m + n$. Then

$$P_{m+1,n} = 2^{-A} \sum_{k=0}^{m} \binom{A}{k}, \qquad P_{m,n} = 2^{-(A-1)} \sum_{k=0}^{m-1} \binom{A-1}{k}.$$

Hence

$$\Delta_{m,n} = 2^{-A} \sum_{k=0}^{m} \binom{A}{k} - 2^{-(A-1)} \sum_{k=0}^{m-1} \binom{A-1}{k}.$$

Now apply Pascal's identity $\binom{A}{k} = \binom{A-1}{k} + \binom{A-1}{k-1}$:

$$\sum_{k=0}^{m} \binom{A}{k} = \sum_{k=0}^{m} \binom{A-1}{k} + \sum_{k=0}^{m} \binom{A-1}{k-1}$$

$$= \binom{A-1}{m} + 2 \sum_{k=0}^{m-1} \binom{A-1}{k}.$$

Therefore

$$\Delta_{m,n} = 2^{-A}\left(\binom{A-1}{m} + 2\sum_{k=0}^{m-1}\binom{A-1}{k}\right) - 2^{-(A-1)}\sum_{k=0}^{m-1}\binom{A-1}{k}$$

$$= 2^{-A}\binom{A-1}{m} + \left(2^{-A}\cdot 2 - 2^{-(A-1)}\right)\sum_{k=0}^{m-1}\binom{A-1}{k}$$

$$= 2^{-A}\binom{A-1}{m},$$

Finally, substituting back $A = m + n$ yields

$$\Delta_{m,n} = 2^{-(m+n)}\binom{m+n-1}{m} = \frac{1}{2^{m+n}}\binom{m+n-1}{n-1}.$$

This is strictly positive because the binomial coefficient is positive. □

*Remark* D.5 (A simple probabilistic interpretation). Lemma D.3 can be rewritten as a Binomial CDF identity: if $Y \sim$ Binomial$(a + b - 1, 1 - x)$, then

$$\mathbb{P}(X \geq x) = \mathbb{P}(Y \geq b).$$

In particular, at $x = 1/2$,

$$P_{a,b} = 2^{-(a+b-1)}\sum_{k=0}^{a-1}\binom{a+b-1}{k} = \mathbb{P}\big(\text{Binomial}(a+b-1, 1/2) \leq a - 1\big).$$

Next, we derive the convergence of sampling proportions and identify the limiting ratios.

**Proposition D.6.** *Consider the offline setting that the entire set of question $\{q_i\}_{i=1}^N$ is available upfront, and the underlying difficulty label of problem $q_i$ is $\theta_i$. Using equal weight Offline PETS with* Beta$(1, 1)$, *the ratio $m_i(B)/B$ converges to a constant for each $i$ as $B \to \infty$. Moreover,*

$$\frac{m_i(B)}{B} \propto \frac{1}{p(\theta_i)}.$$

*Define the KL divergence to $\frac{1}{2}$:*

$$\mathsf{KL}(q\|\tfrac{1}{2}) := q\ln(2q) + (1-q)\ln\big(2(1-q)\big), \qquad q \in (0, 1),$$

*then $p(\theta) = \mathsf{KL}(\theta\|\tfrac{1}{2})$.*

*Proof.* Let $\theta \in (0, 1)$ be the success probability. Draw i.i.d. Bernoulli random variables

$$Y_1, Y_2, \ldots \overset{\text{i.i.d.}}{\sim} \text{Bernoulli}(\theta), \qquad \mathbb{P}(Y_i = 1) = \theta, \ \mathbb{P}(Y_i = 0) = 1 - \theta.$$

For each sample size $T \geq 1$, define the success and failure counts

$$m_T := \sum_{i=1}^{T} Y_i, \qquad n_T := T - m_T = \sum_{i=1}^{T}(1 - Y_i).$$

Clearly, $m_T + n_T = T$ and $(m_T, n_T)$ is the empirical outcome of a binomial experiment with parameter $\theta$, by Strong Law of Large Numbers,

$$\frac{m_T}{n_T} = \frac{m_T/T}{1 - m_T/T} \longrightarrow \frac{\theta}{1 - \theta} \qquad \text{almost surely.}$$

Then along the binomial sample path $(m_T, n_T)$, by Lemma D.4

$$\Delta_{m_T, n_T} = \frac{1}{2^T}\binom{T-1}{m_T} = \frac{1}{2}\mathbb{P}\big(\text{Binomial}(T-1, \tfrac{1}{2}) = m_T\big),$$

and moreover, with $q_T := m_T/(T-1)$, follows from Stirling's formula applied to the binomial pmf $\mathbb{P}(\text{Binomial}(T-1, \frac{1}{2}) = m_T)$ , yielding

$$\mathbb{P}\big(\text{Binomial}(T-1, \tfrac{1}{2}) = m_T\big) \approx \frac{1}{\sqrt{2\pi(T-1)q_T(1-q_T)}} \exp\Big( -(T-1)\, \mathsf{KL}(q_T \| \tfrac{1}{2}) \Big).$$

Finally, since $q_T \to \theta$ almost surely, the exponent satisfies

$$\frac{1}{T} \log \Delta_{m_T, n_T} \;\longrightarrow\; -\mathsf{KL}(\theta \| \tfrac{1}{2}) \qquad \text{a.s.}$$

The remaining part is similar with online case in Section D.1. Which means, since the linear term of $\Delta_{m_T, n_T}$ (resp. $\log R(m; \theta)$) converge to $-\mathsf{KL}(\theta \| \frac{1}{2})$, then we have similar expression

$$m_i(t)\mathsf{KL}(\theta_i \| \tfrac{1}{2}) = m_j(t)\mathsf{KL}(\theta_j \| \tfrac{1}{2}) + O(\log t).$$

The rest derivation is similar, hence

$$\frac{m_i(B)}{B} \propto \frac{1}{\mathsf{KL}(\theta_i \| \frac{1}{2})}.$$

$\square$

### D.3. Discussion of the Asymptotic Behavior for Online vs. Offline Cases

As shown in Sections D.2 and D.1, the asymptotic limits induced by the two allocation paradigms are not exactly identical. This discrepancy mainly stems from the different statistical viewpoints adopted in each setting. In the offline case, the difficulty parameters are inferred during the allocation process from progressively collected samples, and thus optimal decision-making requires a Bayesian perspective: one must update the posterior distribution via a Bayesian framework, otherwise (as discussed in (Chen et al., 2013)) the optimality of the resulting allocation cannot be properly assessed. In contrast, the online streaming case assumes the answer distribution parameters are known *a priori* (or at least can be estimated with sufficient confidence), and therefore employs a frequentist-style update based on empirical frequencies. As a consequence, the two settings lead to slightly different criteria for determining the majority-voting outcome. Nevertheless, we observe that the resulting limiting behaviors are remarkably close in practice; see Figure 2 for an illustration.

## E. Appendix of Experiment

### E.1. Implementation Details in PETS-Offline.

**Confidence-weighted voting.** In the confidence-weighted setting, each reasoning trajectory is associated with a scalar confidence score extracted from the model output following Fu et al. (2025b). Specifically, we compute a *tail trace confidence* for each generated reasoning trace by averaging the token-level confidence scores over the last $2048$ tokens, and use this value as the weight in majority voting. To mitigate the influence of low-confidence traces, we further apply a confidence-based filtering strategy: only the top $70\%$ most confident traces are retained for voting, while the remaining $30\%$ are assigned zero weight. We refer to this scheme as *top-70% confidence-weighted majority voting*. The resulting weighted voting rule is used consistently throughout the paper for defining both self-consistency and the population majority label $y_i^\infty$. Note that we use confidence-weighted majority voting as a proxy for the Bayes-optimal terminal decision.

**Uniform allocation baseline.** Under uniform allocation (standard test-time scaling), we fix an average per-question budget $b = 1, \dots, 64$ and sample exactly $b$ trajectories for every question. When comparing against PETS-Offline, we evaluate Uniform at the per-question budget $b^\star$ that matches the trace count where PETS-Offline first reaches full self-consistency.

### E.2. Implementation Details in PETS-Online.

In the experiment, PETS-Online is evaluated against PETS-Oracle and Uniform allocation. We now introduce each method in detail.

**PETS-Online streaming allocation.** The PETS-Online assumes not having access to $\boldsymbol{\theta}_q$ and implements a fully deployable online streaming policy. Instead of observing the true difficulty, we use a short warm-up procedure to obtain a coarse difficulty proxy. Specifically, for each question we first sample four responses and compute the sorted option-count vector $C_q^{(4)}$. Up to permutation of answer labels, $C_q^{(4)}$ falls into one of five predefined patterns, which deterministically map the question to one of five difficulty grids.

Using training questions with large i.i.d. answer pools, we estimate (i) the empirical mass of each grid and (ii) a representative two-parameter difficulty curve for each grid by fitting a Gaussian-based approximation to the self-consistency curve. Based on these grid-level statistics, we pre-compute the optimal per-grid budgets using the greedy allocation algorithm 1 as in the oracle setting. At test time, each incoming question undergoes the same four-sample warm-up to determine its grid, after which the corresponding precomputed budget is allocated in a single shot. This procedure respects the streaming and one-shot constraints, requires no access to future questions, and uses only minimal online sampling to approximate question difficulty $\boldsymbol{\theta}$.

**PETS-Oracle online streaming allocation.** In the oracle online streaming experiment, we assume access to the true difficulty vector $\boldsymbol{\theta}_q$ of each arriving question $q$, i.e., the underlying answer distribution of the model under infinite sampling. This oracle setting is not deployable in practice and is used solely as an upper bound to assess the optimality of the proposed online allocation strategy. Using a held-out training set with large i.i.d. answer pools, we estimate the prior distribution over question difficulties by discretizing the continuous difficulty space into $K$ grids, each represented by a prototype $\boldsymbol{\theta}_j$ with prior mass $p_j$. Based only on the prior statistics $(p_j, \boldsymbol{\theta}_j)$, we solve the discretized optimization problem in Equation (9) using the greedy algorithm (Alg. 1) to obtain a fixed budget allocation $\{B_j\}_{j=1}^K$ for all grids. At test time, questions arrive sequentially in a streaming manner. For each incoming question, its true difficulty vector $\boldsymbol{\theta}_q$ is revealed, the corresponding grid index is identified, and the precomputed budget $B_i$ is allocated in a single shot. Predictions are obtained by (weighted) majority voting, and we evaluate self-consistency and accuracy under the same protocol as in the offline setting.

**Uniform online streaming allocation.** As a baseline for the online streaming setting, we consider a uniform allocation strategy that assigns the same fixed budget to every question, independent of its difficulty. Given an average per-question budget constraint $\bar{B} = B_{\text{total}}/T$, uniform allocation assigns $B_q \equiv \lfloor \bar{B} \rceil$ samples to each arriving question.

## E.3. Full Experiment Results

### E.3.1. FULL PETS-OFFLINE RESULTS

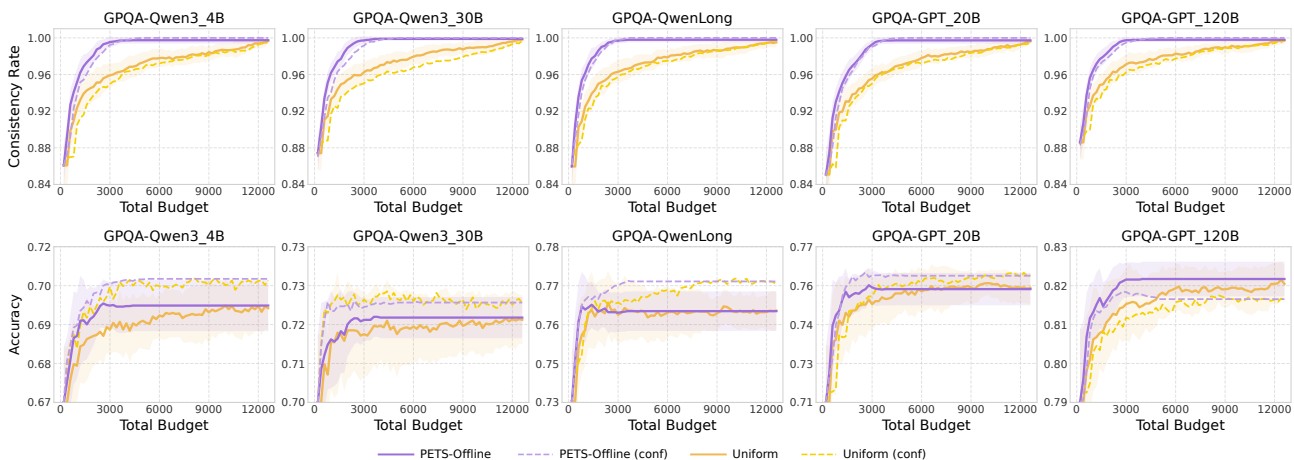

*Figure 6.* GPQA offline

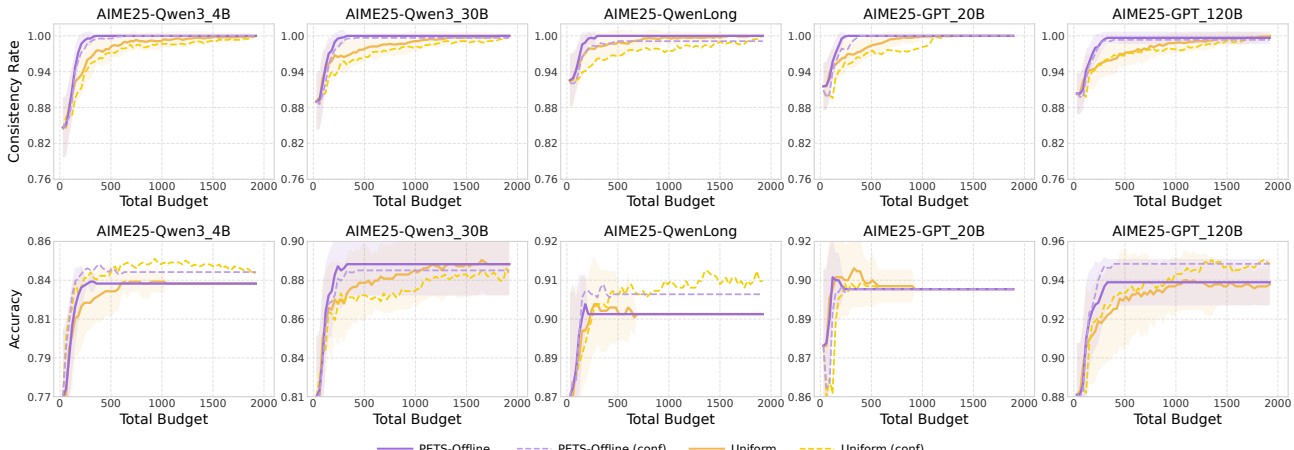

*Figure 7.* AIME 25 offline

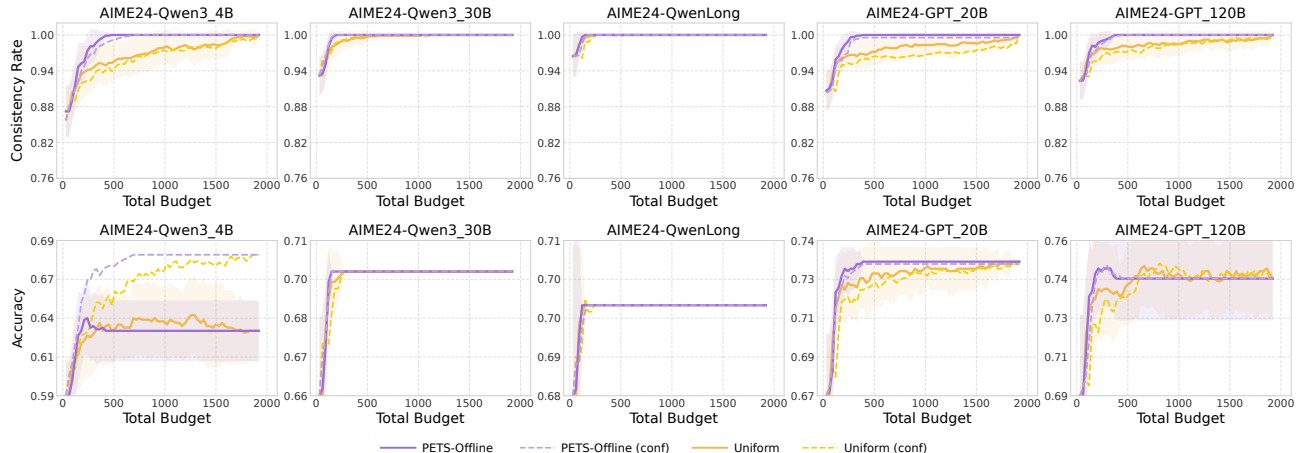

*Figure 8.* AIME 24 offline

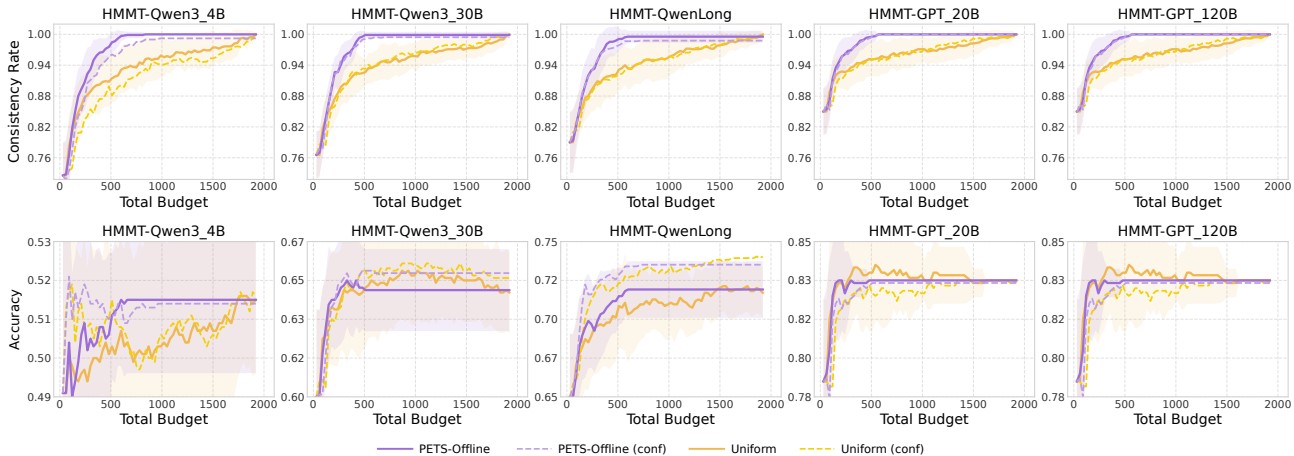

*Figure 9.* HMMT offline

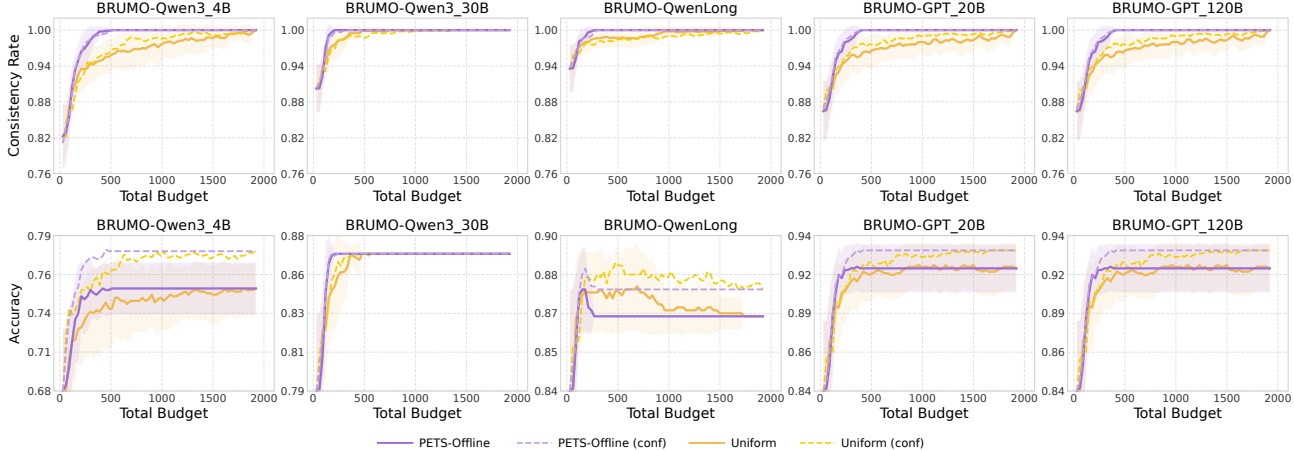

*Figure 10.* BRUMO offline

*Table 7.* PETS-Offline results. Results are formatted as mean (variance).

| Dataset | Method | Qwen3-4B | | | Qwen3-30B | | | Qwen-Long | | | GPT-20B | | | GPT-120B | | |
|---|---|---|---|---|---|---|---|---|---|---|---|---|---|---|---|---|
| | | #tok@con=1↓ | con@match↑ | acc@match↑ | #tok@con=1↓ | con@match↑ | acc@match↑ | #tok@con=1↓ | con@match↑ | acc@match↑ | #tok@con=1↓ | con@match↑ | acc@match↑ | #tok@con=1↓ | con@match↑ | acc@match↑ |
| GPQA | PETS-Offline | 2780 (515.55) | 0.997 (0.00) | 0.697 (0.01) | 2607 (547.05) | 0.999 (0.00) | 0.718 (0.01) | 2513 (529.63) | 0.998 (0.00) | 0.763 (0.01) | 3180 (365.21) | 0.998 (0.00) | 0.757 (0.01) | 2580 (493.68) | 0.998 (0.00) | 0.825 (0.01) |
| | PETS-Offline (conf) | 3667 (758.10) | 0.989 (0.01) | 0.703 (0.01) | 3687 (657.44) | 0.988 (0.01) | 0.721 (0.01) | 2887 (459.94) | 0.992 (0.01) | 0.770 (0.01) | 3693 (534.94) | 0.992 (0.01) | 0.762 (0.01) | 3540 (788.10) | 0.990 (0.01) | 0.821 (0.01) |
| | Uniform | 11013 (1679.85) | 0.957 (0.02) | 0.689 (0.01) | 10367 (2031.53) | 0.958 (0.01) | 0.715 (0.01) | 9973 (2531.31) | 0.958 (0.01) | 0.765 (0.01) | 10853 (2021.22) | 0.958 (0.01) | 0.751 (0.01) | 10393 (2148.61) | 0.965 (0.01) | 0.815 (0.01) |
| | Uniform (conf) | 11453 (1382.58) | 0.947 (0.01) | 0.699 (0.01) | 11607 (1154.88) | 0.949 (0.01) | 0.722 (0.01) | 11400 (1353.41) | 0.950 (0.02) | 0.764 (0.01) | 11193 (1460.74) | 0.956 (0.01) | 0.756 (0.01) | 10500 (1988.76) | 0.957 (0.02) | 0.808 (0.01) |
| AIME25 | PETS-Offline | 212 (47.23) | 1.000 (0.00) | 0.833 (0.00) | 190 (56.99) | 1.000 (0.00) | 0.886 (0.02) | 152 (60.48) | 1.000 (0.00) | 0.900 (0.00) | 181 (47.59) | 1.000 (0.00) | 0.900 (0.00) | 212 (62.50) | 0.997 (0.01) | 0.939 (0.01) |
| | PETS-Offline (conf) | 257 (81.37) | 0.979 (0.02) | 0.840 (0.02) | 223 (58.26) | 0.966 (0.03) | 0.867 (0.03) | 203 (82.13) | 0.970 (0.02) | 0.904 (0.02) | 251 (87.55) | 0.970 (0.03) | 0.897 (0.00) | 211 (65.67) | 0.976 (0.02) | 0.933 (0.02) |
| | Uniform | 470 (286.16) | 0.937 (0.03) | 0.821 (0.02) | 545 (462.35) | 0.957 (0.03) | 0.861 (0.02) | 288 (224.17) | 0.969 (0.02) | 0.893 (0.02) | 410 (244.78) | 0.953 (0.03) | 0.902 (0.02) | 681 (353.75) | 0.947 (0.03) | 0.913 (0.03) |
| | Uniform (conf) | 610 (395.44) | 0.913 (0.05) | 0.833 (0.03) | 763 (508.98) | 0.928 (0.04) | 0.856 (0.03) | 752 (564.94) | 0.942 (0.03) | 0.889 (0.02) | 618 (342.65) | 0.937 (0.04) | 0.890 (0.03) | 911 (542.15) | 0.943 (0.03) | 0.919 (0.03) |
| AIME24 | PETS-Offline | 259 (90.68) | 1.000 (0.00) | 0.631 (0.02) | 135 (31.27) | 1.000 (0.00) | 0.700 (0.00) | 83 (45.04) | 1.000 (0.00) | 0.700 (0.00) | 200 (70.17) | 1.000 (0.00) | 0.733 (0.00) | 184 (81.01) | 1.000 (0.00) | 0.744 (0.02) |
| | PETS-Offline (conf) | 369 (122.85) | 0.961 (0.03) | 0.660 (0.03) | 120 (35.23) | 0.996 (0.01) | 0.699 (0.01) | 91 (53.13) | 0.989 (0.02) | 0.699 (0.01) | 232 (72.18) | 0.969 (0.02) | 0.726 (0.01) | 218 (93.86) | 0.986 (0.02) | 0.737 (0.02) |
| | Uniform | 861 (531.96) | 0.938 (0.03) | 0.630 (0.03) | 202 (129.44) | 0.977 (0.02) | 0.699 (0.01) | 96 (62.23) | 0.990 (0.02) | 0.699 (0.01) | 665 (554.41) | 0.962 (0.02) | 0.719 (0.01) | 448 (483.41) | 0.967 (0.03) | 0.737 (0.03) |
| | Uniform (conf) | 942 (516.00) | 0.922 (0.04) | 0.634 (0.03) | 179 (115.38) | 0.973 (0.03) | 0.686 (0.02) | 115 (78.82) | 0.978 (0.03) | 0.697 (0.01) | 1206 (676.46) | 0.941 (0.02) | 0.708 (0.02) | 535 (501.94) | 0.948 (0.02) | 0.719 (0.03) |
| HMMT | PETS-Offline | 464 (136.40) | 1.000 (0.00) | 0.516 (0.02) | 381 (79.24) | 0.999 (0.01) | 0.647 (0.02) | 363 (97.88) | 0.996 (0.01) | 0.717 (0.02) | 329 (129.81) | 1.000 (0.00) | 0.833 (0.00) | 329 (129.81) | 1.000 (0.00) | 0.833 (0.00) |
| | PETS-Offline (conf) | 579 (198.24) | 0.963 (0.03) | 0.520 (0.03) | 361 (103.47) | 0.973 (0.03) | 0.652 (0.02) | 371 (126.91) | 0.971 (0.03) | 0.732 (0.02) | 324 (118.80) | 0.989 (0.02) | 0.831 (0.01) | 324 (118.80) | 0.989 (0.02) | 0.831 (0.01) |
| | Uniform | 1133 (489.01) | 0.904 (0.04) | 0.500 (0.03) | 1219 (483.32) | 0.913 (0.03) | 0.647 (0.03) | 1227 (463.11) | 0.911 (0.04) | 0.688 (0.03) | 916 (483.33) | 0.938 (0.03) | 0.856 (0.02) | 916 (483.33) | 0.938 (0.03) | 0.856 (0.02) |
| | Uniform (conf) | 1383 (472.53) | 0.872 (0.07) | 0.506 (0.04) | 1089 (473.89) | 0.913 (0.05) | 0.651 (0.03) | 1165 (419.04) | 0.901 (0.03) | 0.717 (0.03) | 902 (464.29) | 0.930 (0.04) | 0.826 (0.03) | 902 (464.29) | 0.930 (0.04) | 0.826 (0.03) |
| BRUMO | PETS-Offline | 280 (74.88) | 1.000 (0.00) | 0.754 (0.02) | 148 (28.33) | 1.000 (0.00) | 0.867 (0.00) | 149 (51.95) | 1.000 (0.00) | 0.867 (0.00) | 253 (75.44) | 1.000 (0.00) | 0.921 (0.02) | 253 (75.44) | 1.000 (0.00) | 0.921 (0.02) |
| | PETS-Offline (conf) | 292 (109.71) | 0.978 (0.03) | 0.773 (0.03) | 162 (53.20) | 0.981 (0.03) | 0.861 (0.02) | 150 (77.59) | 0.981 (0.02) | 0.878 (0.02) | 217 (77.06) | 0.984 (0.02) | 0.924 (0.02) | 217 (77.06) | 0.984 (0.02) | 0.924 (0.02) |
| | Uniform | 826 (481.21) | 0.936 (0.03) | 0.732 (0.03) | 237 (87.14) | 0.958 (0.03) | 0.837 (0.03) | 320 (310.71) | 0.974 (0.02) | 0.873 (0.02) | 776 (494.32) | 0.943 (0.02) | 0.901 (0.03) | 776 (494.32) | 0.943 (0.02) | 0.901 (0.03) |
| | Uniform (conf) | 589 (309.52) | 0.941 (0.04) | 0.757 (0.03) | 310 (256.06) | 0.949 (0.03) | 0.836 (0.03) | 421 (463.32) | 0.970 (0.02) | 0.872 (0.02) | 574 (430.99) | 0.952 (0.03) | 0.907 (0.03) | 574 (430.99) | 0.952 (0.03) | 0.907 (0.03) |

### E.3.2. FULL PETS-ONLINE RESULTS

Besides the full results, here we also presents the comparison with the oracle case of the online setting, which assumes access to the latent parameter $\theta$; in the realistic online setting, $\theta$ is unavailable and must be learned from a training dataset.

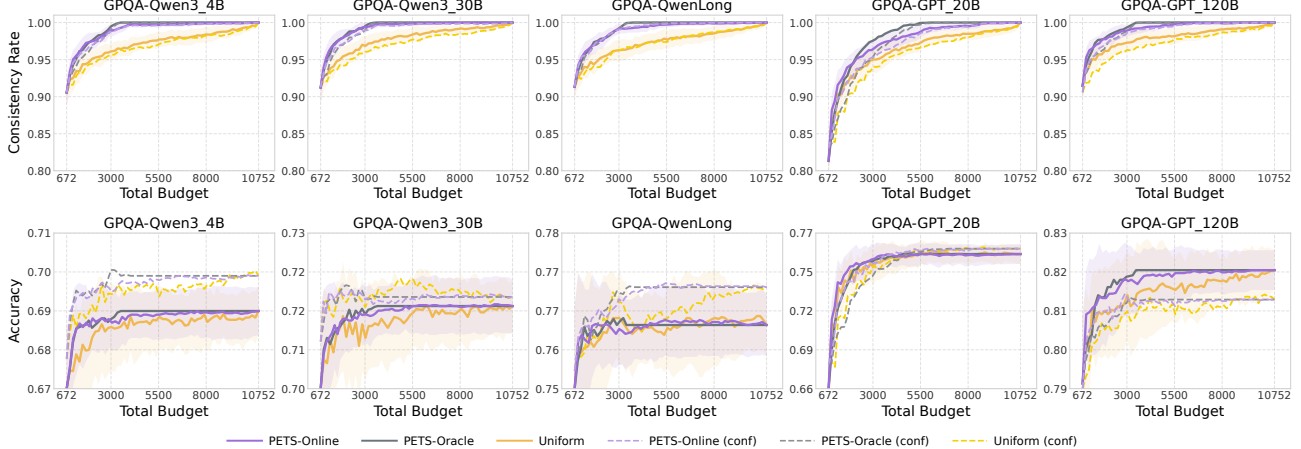

*Figure 11.* GPQA online

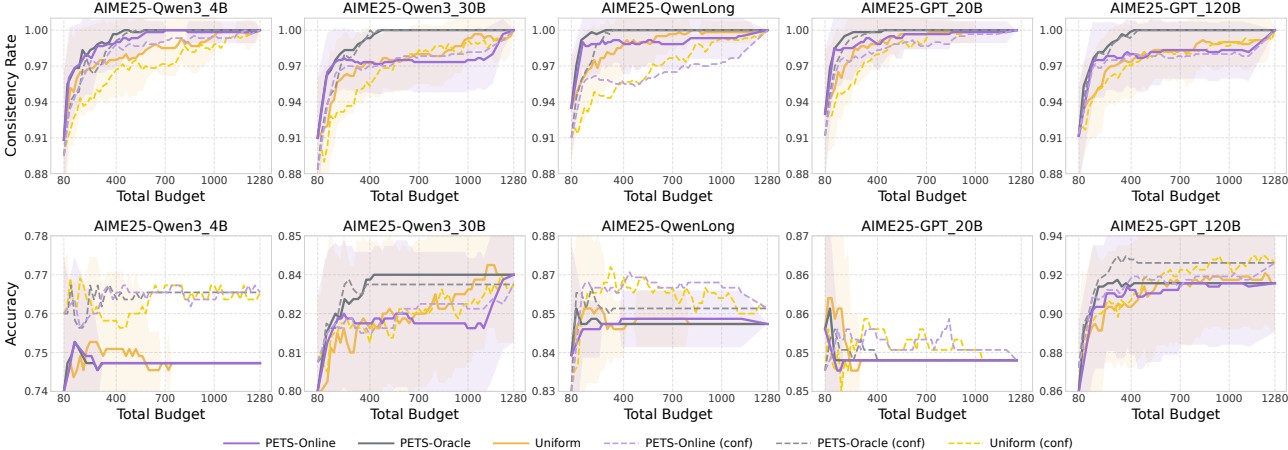

*Figure 12.* AIME 25 online

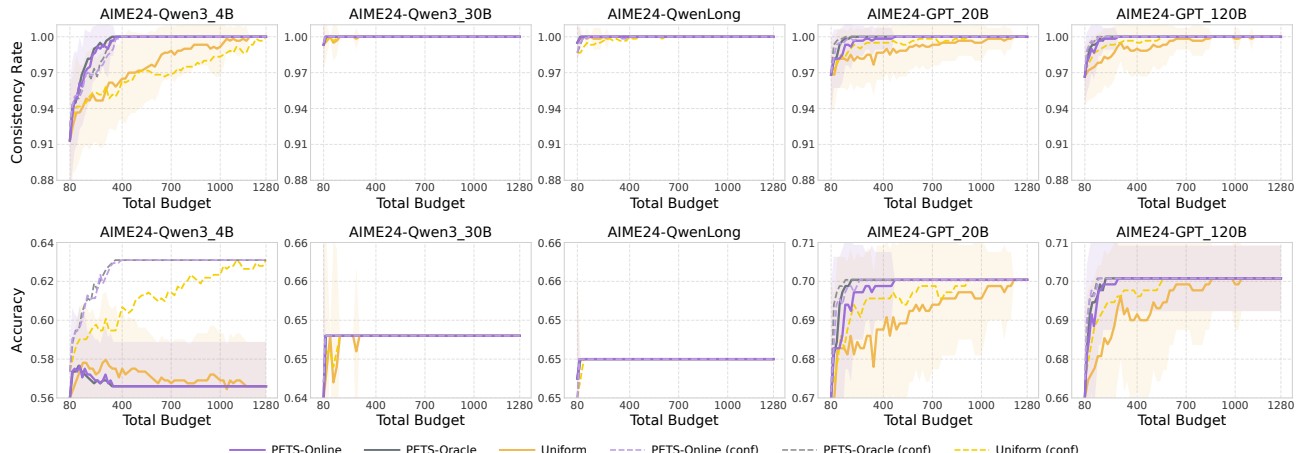

*Figure 13.* AIME 24 online

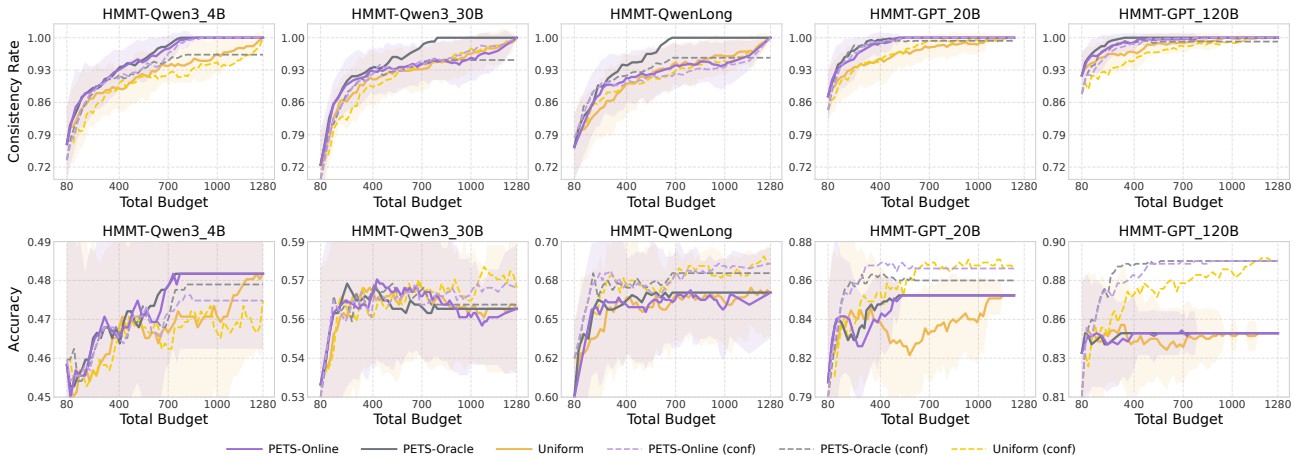

*Figure 14.* HMMT online

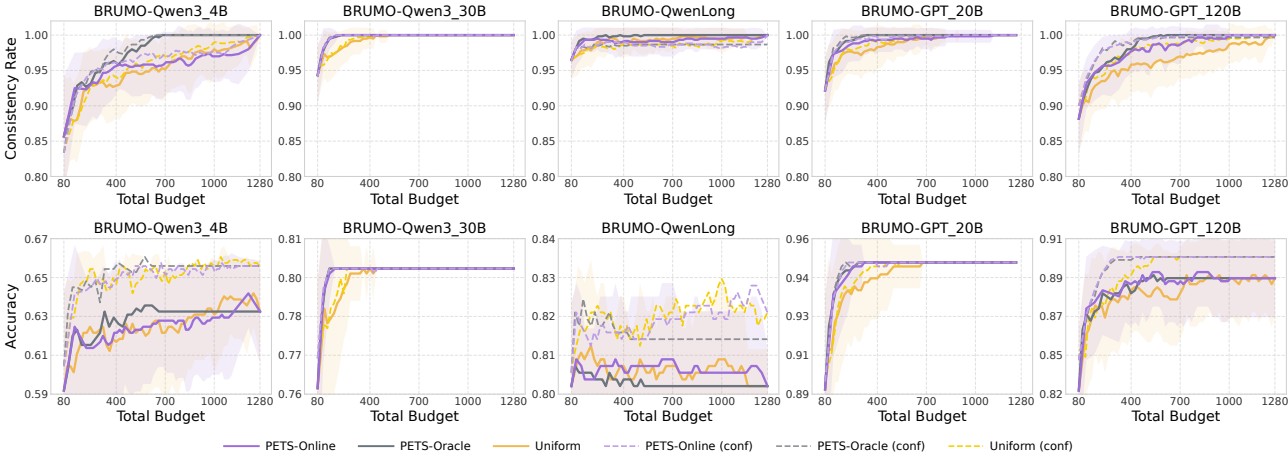

*Figure 15.* BRUMO online

*Table 8.* PETS-Online results. Results are formatted as mean (variance). Oracle case of the online setting assumes access to the latent parameter $\theta$; in the realistic online setting, $\theta$ is unavailable and must be learned from a training dataset.

| Dataset | Method | Qwen3-4B | | | Qwen3-30B | | | Qwen-Long | | | GPT-20B | | | GPT-120B | | |
| --- | --- | --- | --- | --- | --- | --- | --- | --- | --- | --- | --- | --- | --- | --- | --- | --- |
| | | #tok@con=1↓ | con@match↑ | acc@match↑ | #tok@con=1↓ | con@match↑ | acc@match↑ | #tok@con=1↓ | con@match↑ | acc@match↑ | #tok@con=1↓ | con@match↑ | acc@match↑ | #tok@con=1↓ | con@match↑ | acc@match↑ |
| GPQA | PETS-Online | 4662 (1771.56) | 1.000 (0.00) | 0.688 (0.01) | 3826 (1695.09) | 1.000 (0.00) | 0.716 (0.01) | 4852 (2352.11) | 1.000 (0.00) | 0.763 (0.01) | 6826 (1663.76) | 1.000 (0.00) | 0.759 (0.01) | 4172 (1267.37) | 1.000 (0.00) | 0.824 (0.01) |
| | PETS-Online (conf) | 4888 (2048.79) | 0.995 (0.01) | 0.696 (0.01) | 4357 (2223.79) | 0.993 (0.01) | 0.718 (0.01) | 4146 (1741.11) | 0.995 (0.01) | 0.769 (0.01) | 7269 (1543.73) | 0.991 (0.01) | 0.761 (0.01) | 5094 (1403.07) | 0.990 (0.01) | 0.816 (0.01) |
| | PETS-Oracle | 3008 (299.30) | 0.998 (0.00) | 0.687 (0.01) | 2878 (441.03) | 0.998 (0.00) | 0.715 (0.01) | 3127 (394.72) | 0.995 (0.01) | 0.763 (0.01) | 4458 (715.04) | 0.999 (0.00) | 0.759 (0.01) | 2970 (526.74) | 0.998 (0.00) | 0.823 (0.01) |
| | PETS-Oracle (conf) | 3174 (334.03) | 0.995 (0.01) | 0.697 (0.01) | 3061 (548.13) | 0.995 (0.01) | 0.719 (0.01) | 3187 (284.75) | 0.995 (0.01) | 0.769 (0.01) | 5242 (673.92) | 0.996 (0.01) | 0.761 (0.01) | 3289 (276.80) | 0.995 (0.01) | 0.815 (0.01) |
| | Uniform | 9520 (1258.23) | 0.969 (0.01) | 0.681 (0.01) | 8456 (2024.35) | 0.974 (0.02) | 0.712 (0.01) | 9195 (1768.03) | 0.983 (0.01) | 0.760 (0.01) | 9699 (1311.70) | 0.983 (0.01) | 0.761 (0.01) | 8109 (2676.36) | 0.976 (0.01) | 0.818 (0.01) |
| | Uniform (conf) | 9856 (1356.49) | 0.967 (0.02) | 0.694 (0.01) | 9257 (1947.92) | 0.964 (0.01) | 0.720 (0.01) | 9526 (1249.55) | 0.971 (0.01) | 0.764 (0.01) | 9895 (1201.26) | 0.978 (0.01) | 0.763 (0.01) | 9593 (1686.56) | 0.968 (0.01) | 0.814 (0.01) |
| AIME25 | PETS-Online | 194 (111.95) | 1.000 (0.00) | 0.750 (0.00) | 504 (493.65) | 1.000 (0.00) | 0.835 (0.02) | 218 (246.36) | 1.000 (0.00) | 0.850 (0.00) | 195 (221.00) | 1.000 (0.00) | 0.850 (0.00) | 454 (464.19) | 1.000 (0.00) | 0.917 (0.02) |
| | PETS-Online (conf) | 287 (219.04) | 0.947 (0.04) | 0.763 (0.03) | 535 (450.59) | 0.970 (0.04) | 0.825 (0.03) | 681 (492.92) | 0.955 (0.03) | 0.857 (0.03) | 330 (296.69) | 0.960 (0.04) | 0.848 (0.02) | 493 (490.02) | 0.983 (0.02) | 0.923 (0.03) |
| | PETS-Oracle | 168 (89.50) | 0.970 (0.03) | 0.750 (0.01) | 201 (101.39) | 0.978 (0.03) | 0.823 (0.04) | 126 (39.82) | 0.982 (0.02) | 0.857 (0.02) | 134 (65.99) | 0.982 (0.02) | 0.857 (0.02) | 175 (80.60) | 0.982 (0.03) | 0.908 (0.03) |
| | PETS-Oracle (conf) | 223 (102.49) | 0.932 (0.05) | 0.762 (0.03) | 226 (89.60) | 0.952 (0.05) | 0.817 (0.03) | 191 (65.17) | 0.950 (0.04) | 0.855 (0.03) | 178 (67.21) | 0.958 (0.03) | 0.847 (0.02) | 192 (86.07) | 0.963 (0.04) | 0.912 (0.04) |
| | Uniform | 290 (262.13) | 0.948 (0.03) | 0.748 (0.02) | 415 (319.94) | 0.947 (0.06) | 0.810 (0.04) | 229 (151.20) | 0.953 (0.04) | 0.847 (0.03) | 221 (190.98) | 0.955 (0.04) | 0.860 (0.02) | 416 (338.82) | 0.952 (0.05) | 0.892 (0.04) |
| | Uniform (conf) | 452 (311.86) | 0.923 (0.05) | 0.763 (0.03) | 486 (262.88) | 0.935 (0.04) | 0.812 (0.03) | 495 (301.53) | 0.925 (0.04) | 0.835 (0.03) | 349 (222.46) | 0.930 (0.03) | 0.852 (0.02) | 459 (332.33) | 0.943 (0.04) | 0.898 (0.04) |
| AIME24 | PETS-Online | 170 (82.05) | 1.000 (0.00) | 0.562 (0.03) | 81 (3.82) | 1.000 (0.00) | 0.650 (0.00) | 83 (8.68) | 1.000 (0.00) | 0.650 (0.00) | 130 (96.34) | 1.000 (0.00) | 0.700 (0.00) | 107 (37.18) | 1.000 (0.00) | 0.698 (0.01) |
| | PETS-Online (conf) | 185 (103.12) | 0.957 (0.04) | 0.600 (0.04) | 81 (3.52) | 0.998 (0.01) | 0.648 (0.01) | 86 (14.12) | 0.988 (0.02) | 0.648 (0.01) | 104 (40.90) | 0.988 (0.02) | 0.688 (0.02) | 92 (19.26) | 0.997 (0.01) | 0.695 (0.02) |
| | PETS-Oracle | 160 (60.49) | 0.992 (0.02) | 0.562 (0.03) | 82 (5.50) | 1.000 (0.00) | 0.650 (0.00) | 82 (5.38) | 1.000 (0.00) | 0.650 (0.00) | 103 (27.69) | 1.000 (0.00) | 0.700 (0.00) | 103 (28.26) | 0.998 (0.01) | 0.697 (0.01) |
| | PETS-Oracle (conf) | 176 (93.59) | 0.953 (0.04) | 0.603 (0.05) | 82 (5.16) | 0.998 (0.01) | 0.648 (0.01) | 87 (14.00) | 0.990 (0.02) | 0.648 (0.01) | 94 (17.47) | 0.992 (0.02) | 0.692 (0.02) | 95 (20.72) | 0.995 (0.02) | 0.693 (0.02) |
| | Uniform | 393 (269.74) | 0.935 (0.04) | 0.568 (0.04) | 83 (10.61) | 0.997 (0.01) | 0.647 (0.01) | 82 (6.10) | 1.000 (0.00) | 0.650 (0.00) | 237 (294.79) | 0.977 (0.03) | 0.677 (0.03) | 205 (180.65) | 0.972 (0.03) | 0.670 (0.02) |
| | Uniform (conf) | 405 (355.95) | 0.942 (0.05) | 0.588 (0.05) | 83 (8.68) | 0.997 (0.01) | 0.647 (0.01) | 101 (64.22) | 0.988 (0.02) | 0.648 (0.01) | 171 (184.40) | 0.980 (0.02) | 0.680 (0.02) | 144 (120.85) | 0.978 (0.03) | 0.677 (0.03) |
| HMMT | PETS-Online | 531 (199.60) | 1.000 (0.00) | 0.485 (0.02) | 825 (389.51) | 1.000 (0.00) | 0.562 (0.03) | 872 (397.80) | 1.000 (0.00) | 0.670 (0.03) | 248 (129.24) | 1.000 (0.00) | 0.850 (0.00) | 221 (146.53) | 1.000 (0.00) | 0.850 (0.00) |
| | PETS-Online (conf) | 615 (182.54) | 0.933 (0.06) | 0.468 (0.02) | 761 (330.68) | 0.952 (0.05) | 0.568 (0.03) | 1061 (362.50) | 0.950 (0.06) | 0.688 (0.04) | 233 (96.81) | 0.968 (0.05) | 0.855 (0.04) | 263 (134.09) | 0.955 (0.05) | 0.865 (0.03) |
| | PETS-Oracle | 498 (169.60) | 0.937 (0.03) | 0.468 (0.02) | 513 (198.68) | 0.973 (0.05) | 0.558 (0.03) | 803 (331.95) | 0.960 (0.04) | 0.665 (0.04) | 215 (101.26) | 0.982 (0.03) | 0.838 (0.03) | 166 (71.99) | 0.985 (0.03) | 0.850 (0.02) |
| | PETS-Oracle (conf) | 497 (190.79) | 0.915 (0.06) | 0.470 (0.03) | 453 (176.85) | 0.923 (0.06) | 0.565 (0.03) | 690 (364.53) | 0.927 (0.06) | 0.675 (0.04) | 205 (78.56) | 0.955 (0.05) | 0.847 (0.03) | 199 (72.56) | 0.950 (0.04) | 0.867 (0.04) |
| | Uniform | 792 (323.71) | 0.893 (0.05) | 0.462 (0.03) | 813 (345.51) | 0.925 (0.08) | 0.562 (0.04) | 803 (331.81) | 0.962 (0.06) | 0.667 (0.04) | 475 (311.66) | 0.920 (0.04) | 0.833 (0.06) | 309 (181.08) | 0.947 (0.04) | 0.848 (0.03) |
| | Uniform (conf) | 931 (315.65) | 0.885 (0.07) | 0.457 (0.04) | 786 (278.13) | 0.922 (0.11) | 0.567 (0.04) | 959 (304.29) | 0.947 (0.07) | 0.678 (0.04) | 409 (219.94) | 0.920 (0.06) | 0.832 (0.05) | 458 (280.36) | 0.913 (0.06) | 0.850 (0.04) |
| BRUMO | PETS-Online | 601 (478.46) | 1.000 (0.00) | 0.632 (0.03) | 125 (38.38) | 1.000 (0.00) | 0.800 (0.00) | 130 (89.34) | 1.000 (0.00) | 0.802 (0.01) | 213 (202.63) | 1.000 (0.00) | 0.950 (0.00) | 383 (241.37) | 1.000 (0.00) | 0.888 (0.02) |
| | PETS-Online (conf) | 494 (337.36) | 0.968 (0.06) | 0.647 (0.03) | 121 (30.34) | 0.985 (0.03) | 0.795 (0.02) | 138 (114.69) | 0.985 (0.02) | 0.815 (0.02) | 162 (59.26) | 0.993 (0.02) | 0.947 (0.01) | 241 (142.94) | 0.988 (0.02) | 0.897 (0.01) |
| | PETS-Oracle | 345 (201.30) | 0.975 (0.03) | 0.628 (0.04) | 118 (34.74) | 0.980 (0.02) | 0.792 (0.02) | 112 (33.36) | 0.993 (0.02) | 0.805 (0.02) | 141 (53.84) | 0.977 (0.03) | 0.935 (0.02) | 277 (126.84) | 0.948 (0.04) | 0.868 (0.03) |
| | PETS-Oracle (conf) | 312 (117.58) | 0.948 (0.07) | 0.645 (0.03) | 115 (28.09) | 0.972 (0.04) | 0.787 (0.02) | 109 (35.67) | 0.982 (0.02) | 0.812 (0.02) | 128 (30.64) | 0.965 (0.04) | 0.932 (0.03) | 208 (97.93) | 0.963 (0.04) | 0.880 (0.03) |
| | Uniform | 613 (415.94) | 0.940 (0.06) | 0.615 (0.04) | 172 (107.33) | 0.963 (0.04) | 0.777 (0.04) | 128 (30.64) | 0.970 (0.03) | 0.807 (0.02) | 245 (202.96) | 0.948 (0.05) | 0.917 (0.04) | 521 (344.31) | 0.915 (0.03) | 0.863 (0.03) |
| | Uniform (conf) | 519 (270.25) | 0.925 (0.08) | 0.655 (0.05) | 159 (85.26) | 0.955 (0.04) | 0.777 (0.03) | 189 (131.80) | 0.975 (0.03) | 0.813 (0.02) | 184 (98.60) | 0.917 (0.04) | 0.917 (0.04) | 353 (260.85) | 0.942 (0.04) | 0.872 (0.03) |

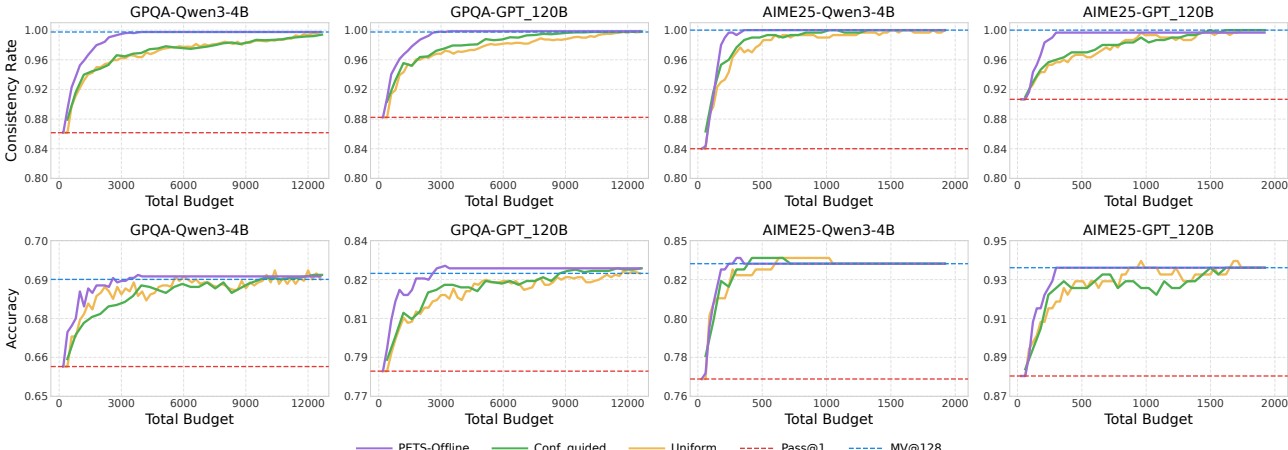

*Figure 16.* Summarization of the detailed offline comparison across datasets and models. Pass@1 denotes single-pass prediction accuracy (a lower-bound reference), while MV@128 denotes majority voting over 128 samples, used as a finite proxy of infinite-budget performance (an upper-bound reference).

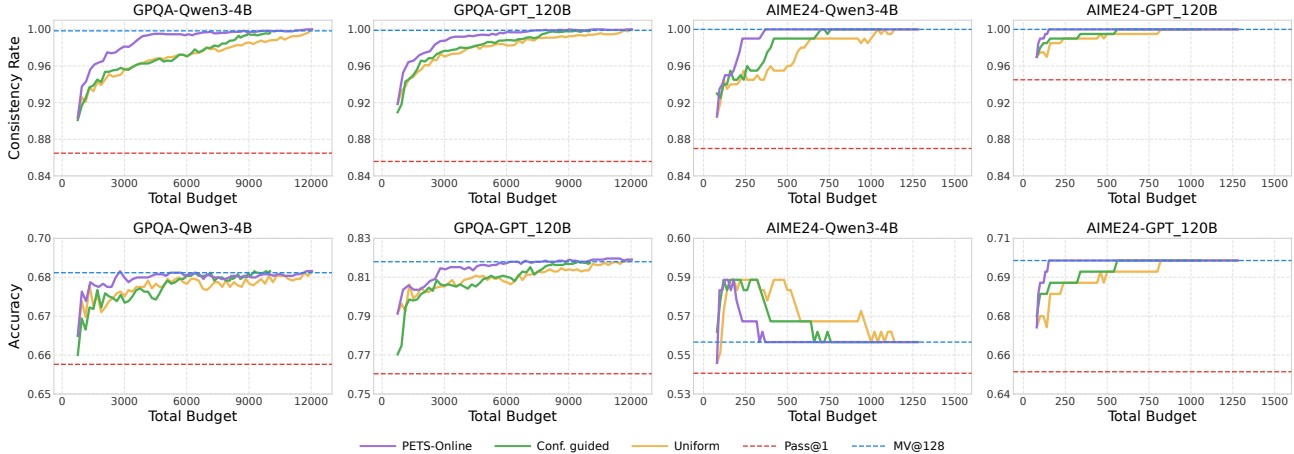

*Figure 17.* Summarization of the detailed online comparison across datasets and models. Pass@1 denotes single-pass prediction accuracy (a lower-bound reference), while MV@128 denotes majority voting over 128 samples, used as a finite proxy of infinite-budget performance (an upper-bound reference).

