# OpenReview forum: "PETS: A Principled Framework Towards Optimal Trajectory Allocation for Efficient Test-Time Self-Consistency"
_ICML.cc/2026/Conference — ICML 2026 regular_

### Official Review · Reviewer_e7fo · 2026-03-06

**Soundness:** 3
**Presentation:** 3
**Significance:** 3
**Originality:** 3
**Overall Recommendation:** 4
**Confidence:** 3

**Summary:**

This paper studies how to properly allocate trajectories to a batch of queries in both the offline and online setting. In the offline setting, the batch of queries are known up front and the goal is find an allocation that maximizes a novel objective known as the self-consistency rate across all queries in the batch. In this setting, the authors establish a connection to crowdsourcing and provide an efficiency majority-voting-based allocation algorithm with theoretical guarantees. In the online setting, the queries arrive sequentially, and allocations must determined on-the-fly. Here, they use a small training set to first estimate the difficulty distribution. Then, they greedily solve a constrained optimize problem to determine the budget for this query. Extensive experiments show that the proposed methods are significantly more efficient than the uniform allocation across various tasks.

**Compliance With Llm Reviewing Policy:**

Affirmed.

**Key Questions For Authors:**

- What is $N$ for each of the datasets you selected?
- I didn't quite understand what you meant by "we repeat this subsampling process 30 times and
report the mean performance." Are you saying that your experimental results are the average of 30 runs, where in each run, the proxy for the infinite budget majority vote for a given question is the majority vote over a subsample of 64 draws?
- What are the wall-clock run times for your method compared to the uniform?
- For the online algorithm, what does "The prior mass of each grid is esti-mated from training data by applying the same warm-up
procedure and computing empirical grid frequencies" mean? Are you using a held-out training set to compute empirical grid frequencies? If so, in the experiments, how is this held-out training set computed? How large does this held-out training set need to be? Are the trajectories needed for the held-out training set taken to be a part of the budget or are they given for free?
- On lines 255-256, you state "we use a lightweight warm-up procedure to assign each incoming question to a grid, after which the corresponding budget is allocated in one shot." Does this mean that on each round after you get question q_i, you use a few trajectories to estimate its difficulty (i.e. place it in a grid)? If so, in your experiments, how many samples do you use to estimate the difficulty? Are these samples used to estimate the difficulty of the questions included in the overall budget?
- I'm confused what "full consistency" means. For a given dataset with N questions, is full consistency when the output of PETS matches the infinite-compute majority vote proxy computed from the 64 samples for each question? If so, how does PETS compute an output? As far as I can tell, PETS just defines an allocation. Is its output just the majority vote amongst the trajectories for each question in the dataset?

**Limitations:**

Yes

**Strengths And Weaknesses:**

Strengths:
- The paper is well-motivated and easy to read
- I liked the notion of self-consistency rate, and feel that it is a natural objective to optimize for
- The authors provide both theoretical and empirical results


Weaknesses:
-  Missing algorithmic details in main text. None of the actual algorithms for either the offline or online setting are in the main-text, which left me wondering about some key details. This was especially the case for the online setting. See questions below.
- Key experimental details missing. I felt that some important experimental details were omitted.  For example, what is $N$ for the datasets that were used? See questions below.
- Minor: the notation in the main text can sometimes be vague. For example, it would helpful if the authors could be explicit about the definition of $\mu_i$ in Lines 182-183. Moreover, I do not think that $\hat{p}$ is defined in Lines 229-230 in the right column.

---

> ### Author Rebuttal · Authors · 2026-03-31
>
> We thank Reviewer e7fo for the positive assessment and the helpful questions on experimental details.
>
> **Q1: Dataset Size**
>
> **A1:** **The dataset sizes are fixed: $N=30$ for AIME24, AIME25, HMMT, and BRUMO, and $N=198$ for GPQA-Diamond.** Here $N$ denotes the number of questions in each benchmark.
>
> **Q2: Subsampling Protocol**
>
> **A2:** **For each question, we use the full pool of $128$ sampled traces to approximate the infinite-budget majority vote, and evaluate finite-budget allocation on random $64$-trace subsets.** This makes the maximum evaluation budget $64N$. We repeat the subsampling process $30$ times and report the mean performance; the variances are shown in Table 4 and Table 5 of the appendix.
>
> **Q3: Wall-clock Runtime**
>
> **A3:** **PETS-Online adds only a small routing/warm-up overhead over Uniform, while remaining far cheaper than sequential Offline OKG planning.** As shown in Table 5 of our [$\text{\color{blue}{additional results}}$](https://shorturl.at/XmTCU), the measured runtimes (seconds) are: GPQA/Qwen3-4B $4.73$ vs $<0.01$ vs $212.2$; GPQA/GPT-120B $4.68$ vs $<0.01$ vs $204.1$; AIME24/Qwen3-4B $1.55$ vs $0.08$ vs $36.0$; AIME24/GPT-120B $1.19$ vs $0.03$ vs $39.0$ (PETS-Online / Uniform / Offline OKG).
>
> **Q4: Grid Prior Estimation**
>
> **A4:** Yes. **We use a small held-out training set to estimate the prior mass of the difficulty grids.** For each training question, we repeatedly apply the same $4$-sample warm-up used at test time, map the induced count pattern to one of $5$ grids, and average the induced grid frequencies to estimate $\hat p_j$. In our experiments, we use $30$ training questions for GPQA and $10$ for AIME. If less training data are available, we reduce the number of bins to keep the estimates stable. These held-out trajectories are outside the reported test-time budget.
>
> **Q5: Warm-up Budget**
>
> **A5:** Yes. **Each incoming test question first uses $4$ warm-up traces to determine one of the $5$ grids, and those $4$ traces are counted in its total test-time budget.** After the grid is identified, we allocate the remaining budget in one shot using the precomputed grid budget.
>
> **Q6: Full Consistency**
>
> **A6:** **Full consistency means that PETS's final aggregated answer matches the MV@128 proxy on every question in the dataset.** PETS specifies the allocation; after sampling under that allocation, the final prediction is obtained by majority vote, or confidence-weighted majority vote in the weighted variant, over the allocated traces.

---

> > ### Author Rebuttal · Reviewer_e7fo · 2026-04-03
> >
> > Thanks for the response! I will maintain my positive score but strongly urge the authors to improve the presentation of the results to address my questions in the final version

---

> > > ### Author Response · Authors · 2026-04-06
> > >
> > > Thank you very much for the follow-up and for maintaining your positive score. We are very glad that our rebuttal resolved your concerns. We also appreciate your suggestion that the presentation of the results should be improved in the final version. We will make this a priority in the camera-ready paper. in particular, we will clarify the experimental protocol, explicitly define the evaluation setup and terminology, and bring key algorithmic and implementation details into a more visible place so that the results are easier to follow. Thank you again for the constructive feedback; we believe these revisions will significantly improve the clarity of the paper.

---

### Official Review · Reviewer_zAa1 · 2026-03-10

**Soundness:** 2
**Presentation:** 3
**Significance:** 3
**Originality:** 2
**Overall Recommendation:** 4
**Confidence:** 3

**Summary:**

This paper introduces PETS, a framework for optimal trajectory allocation in test-time self-consistency under limited budgets. It models reasoning traces as noisy worker labels and formulates budget allocation by leveraging theories from crowdsourcing. The method addresses both offline and online streaming settings. Experiments demonstrate budget reductions compared to uniform allocation while maintaining performance.

**Compliance With Llm Reviewing Policy:**

Affirmed.

**Final Justification:**

My main reservation remains that repeated LLM decoding lacks the worker heterogeneity of classical crowdsourcing, limiting the practical benefit of adaptive allocation when the model distribution is concentrated on incorrect answers. Nonetheless, the connection between test-time scaling and crowdsourcing is interesting, and I raised my score to 4.

**Key Questions For Authors:**

See weaknesses.

**Limitations:**

yes

**Strengths And Weaknesses:**

Strength:
1. The connection between trajectory sampling and classical crowdsourcing, modeling reasoning traces as noisy worker labels and linking allocation to budget allocation, is interesting.
2. The paper is well-organized and clearly written, with a logical progression from problem formulation to theoretical analysis and empirical evaluation.

Weakness:
1. The core algorithm relies heavily on [1]. While the adaptation to LLM test-time scaling is valuable, the underlying methodological framework remains largely identical.
2. The framework assumes that reasoning traces are i.i.d. conditioned on the question. However, unlike independent human workers in crowdsourcing, multiple samples from the same LLM may exhibit correlated errors or mode collapse. Experiments show Con.=1 yet accuracy remains relatively low (e.g., 69.7% on GPQA), suggesting traces are not independent.
3. Moreover, the objective maximizes self-consistency rate not necessarily ground-truth accuracy. If the population majority is systematically wrong, PETS lacks mechanisms to detect this. While low-entropy wrong cases receive less budget, they are confirmed with false confidence; conversely, high-entropy wrong cases may receive excessive budget. The paper does not discuss safeguards against such failure modes.
4. The experimental comparison is limited to uniform allocation baseline. It would be important to compare with recent stronger adaptive methods.
5. The terminal reward lacks an explicit mathematical form; Equation (5) is circularly defined.

[1] Chen X, Lin Q, Zhou D. Optimistic knowledge gradient policy for optimal budget allocation in crowdsourcing[C]//International conference on machine learning. PMLR, 2013: 64-72.

---

> ### Author Rebuttal · Authors · 2026-03-31
>
> We thank the reviewer for the thoughtful questions. We address them below.
>
> **Q1: Novelty Beyond OKG**
>
> **A1:** We agree that the offline case uses a Bayesian allocation skeleton close to Chen et al. (2013), but this is **not** the main contribution. Offline-PETS uses an OKG-style scheme but **changes the objective and extends it to the weighted setting**. The key offline contribution is to use our SC rate to connect test-time scaling and crowdsourcing, so existing crowdsourcing tools can be used for sample-efficient test-time scaling. **More importantly, the online case introduces a new deployment method with theoretical guarantees.**
>
> First, our target is not generic label recovery but self-consistency with the infinite-budget majority vote, the central quantity in test-time scaling. Second, we extend the framework to confidence-weighted aggregation, where both the terminal decision and allocation reward differ from standard worker-reliability weighting. Third, our online method is not a direct OKG rollout: it uses a training-estimated difficulty prior, grid discretization, and one-shot streaming allocation rather than full-batch posterior updates. We will revise the paper to separate inherited and novel parts more clearly.
>
> **Q2: i.i.d. Decoding**
>
> **A2:** **Our i.i.d. assumption is at the level of fresh decoding trials for a fixed question and fixed model, not at the level of independent human workers.** Each trace is generated by a new sampling run from the same conditional distribution
> $$
> p_\theta(r\mid q)=\prod_{t} p_\theta(r_t \mid q, r_{<t}),
> $$
> without feeding previous traces back into the context and without updating model parameters during inference. Under this black-box decoding setup, modeling repeated runs as independent draws from one conditional distribution is the standard idealization we use to analyze allocation. Importantly, the phenomenon "Con.=1 but accuracy is limited" does not by itself imply dependence: it is also compatible with an independent but highly concentrated wrong distribution. If a hard question puts most mass on one incorrect answer, repeated samples can be unanimous and still wrong. We agree that if real decoding exhibits strong correlation or mode collapse beyond this approximation, the gain from allocation can be reduced; we will state this explicitly as a modeling limitation.
>
> **Q3: SC vs. Correctness**
>
> **A3:** **PETS is an allocation method, not a truth-verification mechanism.** Its objective is to spend a limited sampling budget so that finite-budget voting matches the corresponding infinite-budget vote as efficiently as possible. This does not mean the infinite-budget vote is always correct. For low-entropy but wrong questions, PETS will not "fix" the model by allocating more traces; it will simply reach the wrong population majority faster. For high-entropy wrong questions, more budget can still be assigned because the allocation target is stability of the majority vote, not direct estimation of ground-truth correctness. We therefore agree that allocation-only methods need not protect against all systematic model errors. This is a limitation of allocation-only methods, not evidence that the objective itself is meaningless. That limitation, however, does not make self-consistency useless: **under a fixed aggregation rule and deployment protocol, higher self-consistency still typically tracks higher accuracy,** which is also the empirical pattern in our Fig. 3 and Fig. 4.
>
> **Q4: Adaptive Baselines**
>
> **A4:** **The comparison is not limited to uniform; PETS also outperforms a nontrivial confidence-guided adaptive baseline.** In the [$\text{\color{blue}{additional results}}$](https://shorturl.at/XmTCU), we further compare against confidence-guided budget allocation [1]. These results show that confidence-guided allocation already improves over uniform in both offline and online settings, while PETS still achieves stronger overall performance. We will incorporate this stronger empirical positioning in the revision and avoid wording that overstates optimality.
>
> [1] Fu, Y., Wang, X., Tian, Y., & Zhao, J. (2025). Deep think with confidence.
>
> **Q5: Eq. (5) Clarification**
>
> **A5:** **Eq. (5) is a compact decision rule under the terminal posterior, but we agree that the main-text form is too compressed.** The intent is **not circular**: Eq. (5) is dataset-level shorthand for choosing terminal labels that maximize posterior self-consistency after the final belief state $S^H$ is formed. Because the objective is separable across questions, this is equivalent to choosing each question independently as $\hat y_i \in \arg\max_m P(y_i^{W,\infty}=m \mid S_i^H)$. Appendix B.3 already makes this explicit in Eq. (12), where the sum over questions decomposes into per-question maximization, so the concern is about exposition rather than a mathematical inconsistency. We will rewrite Eq. (5) in per-question form and expand the verbal definition of the terminal reward in the main text.

---

> > ### Author Rebuttal · Reviewer_zAa1 · 2026-04-02
> >
> > In traditional crowdsourcing, independence is typically accompanied by worker heterogeneity, meaning that different annotators often have different preference structures and error distributions. This diversity can help average out individual mistakes. In contrast, repeated decoding from the same LLM, even if modeled as independent draws, still comes from a shared underlying model distribution, making the samples much more likely to exhibit similar biases and error patterns. As a result, compared with classical crowdsourcing, adaptive allocation here may not merely fail to correct systematic errors, but may even amplify them by assigning additional budget to an already wrong but internally consistent mode. This makes the cost of allocating extra budget to errors especially important to analyze in the LLM setting.
> >
> > Overall, although I still have some reservations, I find the overall connection between test-time scaling and crowdsourcing interesting, and I am willing to raise my score to 4.

---

> > > ### Author Response · Authors · 2026-04-04
> > >
> > > Thank you for raising your score and additional insightful comments! We here provide additional answers and hope that this can further ease your concerns.
> > >
> > > We agree that this is precisely the key conceptual difference between classical crowdsourcing and repeated sampling from the same LLM. **In standard crowdsourcing, labels are typically modeled as noisy observations of an underlying true label $y^*$.** Under the usual assumption that the aggregated worker distribution is centered on the truth, i.e.
> > > $$
> > > Y=y^{\*}+\\epsilon, \\epsilon\\sim \\mu\\text{ and the mode of }\\mu\\text{ is } 0.
> > > $$
> > > Then, the majority voting(or average when $\mathbb{E}[\mu]=0$) over independent samples converges to the correct answer as the number of labels increases. In this regime, additional budget reduces stochastic noise and therefore tends to improve accuracy. **For repeated decoding from a single LLM, however, even if we model the outputs for a question $x$ as i.i.d. samples from an answer distribution $p_x(y)$, the mode of this distribution need not be the correct answer.** Let
> > > $$
> > > \hat y_x = \arg\max_y p_x(y).
> > > $$
> > > Then, as the number of samples $n$ grows, the majority vote converges to $\hat y_x$. Therefore, if $\hat y_x \neq y_x^*$, repeated sampling will converge to the wrong answer with increasing confidence. In that case, the value of resampling for accuracy is actually negative rather than positive. We fully agree with the reviewer that, compared with classical crowdsourcing, adaptive allocation in the LLM setting may amplify an already wrong but internally consistent mode.
> > >
> > > In both settings, the common idea is to use repeated sampling and mode aggregation to suppress randomness; this is also the standard intuition behind test-time scaling. Our contribution is to optimize this resampling process by allocating the budget according to estimated question difficulty. **However, our method does not and cannot resolve the “wrong-mode” failure case at test time, since this is a property of the base model’s answer distribution rather than of the allocation rule.** Without changing model parameters or the decoding distribution itself, this issue is generally not removable by resampling alone. However, what we can show theoretically is the following. Let $AC_i(n)$ denote the probability that the majority vote is correct on question $i$ when using $n$ samples. Then there are two regimes:
> > >
> > > - If the correct answer is the modal answer, then $AC_i(n)$ increases with $n$ and converges to $1$.
> > >
> > > - If an incorrect answer is modal, then $AC_i(n)$ decreases with $n$ and converges to $0$.
> > >
> > > Hence, the net gain of adaptive resampling can be decomposed as
> > > $$
> > > \Delta=
> > > \sum_{i \in \mathcal G} \bigl(A_i(n_i)-A_i(m_i)\bigr)
> > > +
> > > \sum_{i \in \mathcal B} \bigl(A_i(n_i)-A_i(m_i)\bigr),
> > > $$
> > > where $\mathcal G$ is the set of questions whose modal answer is correct, and $\mathcal B$ is the set whose modal answer is incorrect, and $n_i$ is our scheduling, while $m_i$ is the baseline, like uniform sampling. The first term is positive, while the second term is non-positive. Therefore, **if most questions belong to the first regime, or more generally if the positive gains dominate the negative ones, adaptive allocation still improves overall accuracy and computational efficiency, and this is generally satisfied for most well-trained models.** We will clarify this assumption and limitation more explicitly in the revision.

---

### Official Review · Reviewer_YDRN · 2026-03-14

**Soundness:** 2
**Presentation:** 3
**Significance:** 2
**Originality:** 3
**Overall Recommendation:** 4
**Confidence:** 2

**Summary:**

This paper can be broadly categorized as the LLM test-time inference research. More specifically, there is significant computational waste in the LLM self-consistency sampling if we choose a uniform sampling budget. The existing uniform methods are neither necessary nor efficient. To resolve this, the authors propose the PETS framework based on Bayesian MDP. It considers both the offline setting and online setting and dynamically adjusts the number of sampling trajectories.
The primary contribution of this work is providing a principled theoretical framework for test-time trajectory allocation. In their empirical study, the authors demonstrate that PETS can save significant computational budget compared to uniform allocation while maintaining or even improving overall accuracy.

**Compliance With Llm Reviewing Policy:**

Affirmed.

**Final Justification:**

After the author's reponse, my concerns are partially resolved. I raised my rating accordingly based on my updated understandings.

**Key Questions For Authors:**

1. As discussed above, self-consistency (SC) and accuracy appear to represent two fundamentally different objectives. Higher SC does not necessarily translate into higher accuracy, which is also reflected in Figures 3 and 4. The authors attribute this phenomenon to a fundamental limitation of LLMs, arguing that majority voting cannot help when the model is systematically wrong. However, there may be additional factors that have not been analyzed in depth. For example, in the experiment “AIME25–Qwen3–4B”, when the budget becomes sufficiently large, both PETS-online and PETS-offline achieve very similar SC scores. However, there remains a noticeable gap in their accuracy. Similar patterns can also be observed in several cases in Figure 4. Since the same model is evaluated on the same dataset, it seems unlikely that this gap is caused purely by inherent model bias or systematic errors. This observation raises an interesting question: the model may achieve very high self-consistency in multiple distinct ways, yet the resulting accuracies can differ significantly. In other words, outputs can be nearly 100% self-consistent while still leading to substantially different correctness rates. If this is indeed the case, it may challenge the fundamental usefulness of self-consistency as a proxy objective. I would appreciate it if the authors could further discuss this phenomenon.

2. In the online setting, the algorithm relies heavily on estimating the distribution parameters within each difficulty region. The choice of the number of difficulty bins therefore introduces additional inductive bias into the algorithm. I am wondering whether there is a principled way to determine the number of bins (e.g., through theoretical guidance, validation procedures, or sensitivity analysis). Since these distribution parameters are estimated online, it would be helpful to better understand the statistical behavior of these estimates. For example, do the parameter estimates converge over time, and if so, what is the convergence rate? If the estimates are inaccurate, how does this affect the resulting budget allocation decisions? Finally, I have some concerns about the use of the term “optimal budget allocation.” The closed-form solution appears to assume that the underlying parameters are correctly estimated, which may not hold in the online setting. In practice, the allocation is based on estimated parameters rather than the true ones. Therefore, referring to it as “optimal” might be somewhat overstated. It may be more precise to characterize it as the closed-form solution for the equations.

**Limitations:**

yes

**Strengths And Weaknesses:**

Soundness:
* Pros: This paper formulates the sampling process as an optimization problem. Using MDP and Bayesian technique, this paper provides an principled approach to allocate the sampling budget. It considers both the online and offline cases. In the empirical study, it validates its superior performance in several benchmarks and demonstrates its potential in reducing the number of samples required to achieve consistency.

* Cons: Improvement of self-consistency is not equeveliant to accuracy, which is fundamentally limited by the model's power of presentation. In other words, this method couldn't resolve the systematic bias in the model. In addition, the probability modeling is oversimplified as it's only assuming the catgorical distribution and gaussian distribution for label and confidence. Though this simplifies the algorithm design, it could deviate from the real distribution. In addition, the LLM confidence is usually uncalibrated and would provide reasonable guidance on the Bayesian Optimization

Presentation
* Pros: The problem is well motivated, and test-time intelligence represents an important research direction for both industry and academia. It's also clearly organized and comprehensive with both online and offline scenarios.

* Cons: Some important elements are deferred to the appendix, for instance, the parameter estimation in online optimization is vital and it's not discussed in the main paper. I think some empirical results and large tables could be moved to the appendix as they may be redundant in terms of new information.

Significance
* Pros: Test-time intelligence represents an important research direction for both industry and academia. Their empirical results demonstrates their algorithm has a great potential in reducing the sampling complexity.

* Cons: As the method is based on budget allocation, it is likely most effective when the underlying model is sufficiently strong. In the online setting, the approach relies heavily on distribution estimation, which may be inaccurate and affect overall performance. Moreover, parameter estimation and hyperparameter choices (e.g., the number of bins) may become more challenging in dynamic environments beyond static datasets. Additional discussion on online parameter estimation would help clarify the algorithm’s practicality.

Originality
* Pros: The paper brings a new pespective in converting the resource allocation into a crowdsourcing problem. This new pespective brings more rigor in the area of inference scaling. The marginal gain is very interesting in guiding the sampling process and it moves beyond the existing heurestic methods.

* Cons: The core algorithm builds primarily on well-established methods (e.g., OKG), making the contribution appear more as an applied engineering improvement rather than a fundamental advance in ML theory. In addition, the approach still relies on majority voting (with or without weighting), mainly improving the efficiency of existing self-consistency methods. As such, the work extends an existing research line rather than introducing a fundamentally new direction.

---

> ### Author Rebuttal · Authors · 2026-03-31
>
> We thank the reviewer for the thoughtful comments. We address the questions below.
>
> **Q1: SC vs. Accuracy**
>
> **A1:** Self-consistency (SC) and ground-truth accuracy are not equivalent. **At test time, however, the model is fixed,** so its underlying capability cannot be changed by allocation. In this fixed-capability regime, self-consistency is a standard way to improve accuracy [1]: one samples multiple reasoning trajectories and aggregates them into a more reliable final answer. **This is the basic idea behind test-time scaling.** The point is not that SC equals accuracy, but that under a fixed model it is the relevant reliability target for allocation.
>
> Our motivation is that **if higher SC within a fixed setting generally helps achieve higher accuracy, how can we improve SC under the same budget?** As shown in Fig. 3, Fig. 4, and our additional results in [$\text{\color{blue}{additional results}}$](https://shorturl.at/XmTCU), larger SC budgets consistently lead to higher accuracy within each fixed setting and variant. **PETS follows this principle, but focuses on achieving high SC with fewer samples:** we optimize how a limited sampling budget is allocated across questions to achieve high SC and, empirically, higher accuracy at lower cost.
>
> [1] Wang, X., et al. (2022). Self-consistency improves chain of thought reasoning in language models.
>
> **Q2: Accuracy Gap at High SC**
>
> **A2:** **SC must be interpreted relative to a matched aggregation rule and evaluation protocol, not as an absolute accuracy guarantee across variants.** In the unweighted case, SC measures recovery of $y_i^\infty$; in the weighted case, it measures recovery of $y_i^{W,\infty}$. **Thus two curves can both approach 100% SC yet have different accuracies if their infinite-budget targets differ.** This is exactly why Fig. 3 and Fig. 4 define consistency within each matched comparison group.
>
> **The offline and online curves also come from different protocols.** Offline allocation is computed over the full batch with sequential posterior updates, whereas online allocation uses a held-out training set to estimate the prior and then makes one-shot streaming decisions. Hence similar SC values, e.g., on `AIME25-Qwen3-4B`, do not imply identical predictions or accuracy. We will revise the text to make clear that SC is meaningful only under a fixed aggregation rule and deployment setting. That is the sense in which we use SC throughout Figs. 3-4.
>
> **Q3: More Discussion on Online Estimation**
>
> **A3:** **Algorithm 1 is optimal for the discretized plug-in objective with fixed prototypes/masses**; although PETS-Online is not oracle-optimal, it is still a principled approximation.  Theorem 4.1 proves optimality only for this discretized problem. Specifically, for each arriving question $q_t$, PETS-Online first draws a warm-up set for estimation, then we obtain a soft assignment over bins. The representative parameter in each bin is estimated by a weighted plug-in fit over historical questions, and the online allocator solves the discretized plug-in objective. **Algorithm 1 is optimal for this plug-in problem, but not oracle-optimal for the true online problem with unknown parameters. We will revise the paper to make it explicit.**
>
> This also clarifies the two main error sources: (i) **discretization bias**, from approximating the continuous difficulty space by $K$ bins, and (ii) **estimation error**, from using $\widehat p_j(t)$ and $\widehat\theta_t^j$ instead of the true quantities. A useful decomposition is
> $$\mathrm{Regret}\sim\mathrm{Disc}(K)+\sum_{j=1}^K |\hat p_j(t)-p_j|+\sum_{j=1}^K p_j\sup_B |SC(\hat\theta_t^j;B)-SC(\theta^j;B)|$$. We will add a more detailed analysis in the future version.
>
>  **Tables 3/4 in our [$\text{\color{blue}{additional results}}$](https://shorturl.at/XmTCU) also show limited $K$ sensitivity.** Empirically, PETS-Online stays close to PETS-Oracle, suggesting limited error in our experiments. Appendix D supports asymptotic stabilization of allocation proportions, but the paper should not imply a full guarantee for the online estimator.
>
> **Q4: Relation to OKG Algorithm**
>
> **A4:** **PETS is inspired by OKG in the offline case, but it is not a direct OKG application.** In the **offline setting**, we cast trajectory allocation as a Bayesian decision problem in the crowdsourcing spirit of OKG and extend it to the **weighted** setting, which OKG does not handle.
>
> In the **online setting**, the method is **structurally different**: it uses a training-estimated difficulty prior, grid discretization, and **one-shot streaming** allocation without full-batch posterior updates. As discussed in Appendix D.3, this gives a different statistical viewpoint from the offline Bayesian OKG-style formulation. We will revise the paper to make this distinction more explicit.
>
> **Q5: Paper Presentation**
>
> **A5:** We thank the reviewer for the advice. We will carefully revise the manuscript to include more vital content in the main paper.

---

> > ### Author Rebuttal · Reviewer_YDRN · 2026-04-05
> >
> > Thanks for the author's detailed response.
> >
> > However, I still have some concerns about the the gap between SC and accuracy. I agree that in the test time, the inherent capability of the model is fixed and we are trying to optimize its consistency given certain budget. However, as you may notice in Figure 3, different methods may eventually converge to a similar level of consistency (e.g. 100%) as budget increases, however, their accuracy in the same setup could still vary. This may indicate that SC may not be a good surrogate metric, as our ultimate goal is to optimize the model's accuracy or other actual performance metric.
> >
> > Thus, I would keep my current score as a boarderline score.

---

> > > ### Author Response · Authors · 2026-04-05
> > >
> > > We thank Reviewer YDRN for the helpful follow-up.
> > >
> > > **Fig. 3 contains two matched comparison groups in the offline setting:** (i) the unweighted setting, i.e., **PETS-Offline vs. Uniform (solid purple vs. solid yellow),** and (ii) the confidence-weighted setting, i.e., **PETS-Offline (conf) vs. Uniform (conf) (dashed purple vs. dashed yellow).** These two settings **use different aggregation rules of reasoning traces** and therefore may have different asymptotic accuracies. Hence, **the accuracy variance across solid and dashed curves comes from the difference between the unweighted and confidence-weighted setups,** rather than from SC failing within a single setup.
> > >
> > > **Within each fixed setting, as the total budget increases, SC increases, and accuracy also increases, and both eventually converge.** **PETS reaches that convergence faster,** instead of changing the covnergence target. This holds in both matched groups: in the unweighted setting, PETS-Offline converges faster than Uniform, and in the confidence-weighted setting, PETS-Offline (conf) converges faster than Uniform (conf).
> > >
> > > Under a fixed model and a fixed aggregation rule, SC is the relevant allocation objective. In that regime, increasing the budget improves both SC and accuracy, and PETS achieves this convergence more efficiently than Uniform in both settings.

---

### Decision · Program_Chairs · 2026-04-30

**Decision:**

Accept (regular)

**Comment:**

Thank you for your submission to the ICML 2026. We have now received reviews for your manuscript. Although everyone has pointed out some merits, they also have raised some concerns, such as experimental setup and theoretical consideration. During the rebuttal period, the authors’ feedback has helped on clarifying the reviewers’ concerns. Thus, the average score (4) is both above the average levels. Based on the current reviews and closed-door reviewer discussions, every reviewer seems to be fine with an acceptance.